# EFFIBENCH: Benchmarking the Efficiency of Automatically Generated Code

**Dong Huang**[*]
The University of Hong Kong
dhuang@cs.hku.hk

**Yuhao Qing**[*]
The University of Hong Kong
yhqing@cs.hku.hk

**Weiyi Shang**
University of Waterloo
wshang@uwaterloo.ca

**Heming Cui**
The University of Hong Kong
Shanghai AI Laboratory
heming@cs.hku.hk

**Jie M. Zhang**[†]
King's College London
jie.zhang@kcl.ac.uk

## Abstract

Code generation models have increasingly become integral to aiding software development. Although current research has thoroughly examined the correctness of the code produced by code generation models, a vital aspect that plays a pivotal role in green computing and sustainability efforts — the efficiency of the generated code — has often been neglected. This paper presents EFFIBENCH, a benchmark with 1,000 efficiency-critical coding problems to assess the efficiency of code generated by code generation models. EFFIBENCH contains a diverse set of LeetCode coding problems. Each problem is paired with an executable human-written canonical solution, which obtains the SOTA efficiency on the LeetCode solution leaderboard. With EFFIBENCH, we empirically examine the ability of 42 large language models (35 open-source and 7 closed-source) in generating efficient code. Our evaluation results demonstrate that the efficiency of the code generated by LLMs is generally worse than the efficiency of human-written canonical solutions. For example, GPT-4 generated code has an average **3.12** times execution time that of the human-written canonical solutions. In the most extreme cases, the execution time and total memory usage of GPT-4 generated code are **13.89** and **43.92** times that of the canonical solutions. The source code of EffiBench is released on `https://github.com/huangd1999/EffiBench`. We also provide the LeaderBoard in `https://huggingface.co/spaces/EffiBench/effibench-leaderboard`.

## 1 Introduction

Large language models (LLMs), such as GPT-4 [42] and Copilot [36], have become increasingly popular for assisting software developers with various tasks such as program repair [18, 26], automated testing [29, 14], and code translation [44, 3]. LLMs generate code based on instructions and offer intelligent recommendations, boosting developers' productivity. Various benchmarks have been proposed to evaluate the **correctness** of code generation. Notable examples include HumanEval [12], APPS [21], BigCodeBench [62], and DS-1000 [28], which cover basic programming, competition-level, and data science tasks. These benchmarks have been widely used to assess the code generation capabilities of LLMs.

---

[*]Equal Contribution.
[†]Corresponding Author.

38th Conference on Neural Information Processing Systems (NeurIPS 2024) Track on Datasets and Benchmarks.

**Task Description**

Write a function to merge two sorted arrays into a single sorted array.
**Input:** Two arrays, array1 and array2, each sorted in non-decreasing order.
**Output:** A single array that combines the elements of array1 and array2, sorted in non-decreasing order.
**Example:** Given array1 = [2, 7, 11, 15], array2 = [5, 8, 15, 17], return [2, 5, 7, 8, 11, 15, 15, 17].

**VS Code Copilot v1.156.683**

```python
def mergeArrays(array1, array2):
    # Manually concatenating the two
    ↪    arrays
    merged_array = []
    for element in array1:
        merged_array.append(element)
    for element in array2:
        merged_array.append(element)

    n = len(merged_array)
    for i in range(n):
        for j in range(0, n-i-1):
            if merged_array[j] >
    ↪    merged_array[j+1]:
                merged_array[j],
    ↪    merged_array[j+1] =
    ↪    merged_array[j+1],
    ↪    merged_array[j]

    return merged_array
```

**GPT-4**

```python
def mergeArrays(array1, array2):
    # Merging two arrays efficiently
    result = []
    i, j = 0, 0
    while i < len(array1) and j <
    ↪    len(array2):
        if array1[i] < array2[j]:
            result.append(array1[i])
            i += 1
        else:
            result.append(array2[j])
            j += 1
    # Append remaining elements of array1
    while i < len(array1):
        result.append(array1[i])
        i += 1
    # Append remaining elements of array2
    while j < len(array2):
        result.append(array2[j])
        j += 1
    return result
```

Figure 1: Example codes with distinct time complexity generated by Copilot and GPT-4, respectively. Code accessed on January 15, 2024.

Despite advancements in ensuring code correctness, there remains a significant gap in the literature regarding the efficiency of code produced by LLMs [40, 50]. The importance of efficiency cannot be understated, as it directly impacts the speed of execution and the utilization of memory, which is especially important in resource-constrained environments such as mobile devices or embedded systems [46]. **Efficiency** of code is crucial for building scalable and sustainable software to meet the growing demands of the digital world. Furthermore, efficient code plays a pivotal role in green computing and sustainability efforts. By optimizing algorithms and reducing computational overhead, we can significantly lower energy consumption and carbon footprint. This is particularly relevant as the global demand for digital services increases.

The efficiency of two correctly generated code snippets for the same task can vary significantly. Consider the example in Figure 1, where Copilot and GPT-4 are tasked with merging two sorted arrays. Copilot generates a function that concatenates the arrays and then applies a basic Bubble Sort algorithm. While functionally correct, this approach suffers from sub-optimal time complexity of $O((n + m)^2)$ and space complexity of $O(n + m)$, where $n$ and $m$ are the array lengths. In contrast, GPT-4 generates a function that efficiently merges the arrays by systematically comparing and appending elements from each array in a single pass. This method achieves a time complexity of $O(n + m)$, exhibiting a linear relationship with the combined lengths of the arrays. Its space complexity remains $O(n + m)$. The disparity in efficiency highlighted in Figure 1 underscores the critical need to benchmark code generation from the perspective of code efficiency.

While being intuitive, using existing code generation benchmarks like HumanEval [12] and MBPP [7] to assess code efficiency has several limitations. These efforts primarily focus on correctness, often featuring simple tasks solvable with short code snippets. This simplicity can lead to indistinguishable efficiency across different LLMs, making it difficult to discern meaningful differences in their performance. Furthermore, most tasks are not inherently efficiency-critical, making any observed efficiency discrepancies less significant. Finally, these benchmarks lack comprehensive and diverse

test cases that can thoroughly evaluate code efficiency under varying and substantial computational loads. Consequently, they are inadequate for assessing the efficiency of code generation.

This paper introduces EFFIBENCH, a benchmark specifically designed for evaluating the efficiency of the code that is automatically generated. EFFIBENCH comprises 1,000 efficiency-critical code generation problems selected from LeetCode. Each coding problem is paired with an executable manually-written canonical solution which has been awarded the highest rating on LeetCode for its optimal time and space efficiency. We also develop a test case generator to produce a vast number of test cases for each problem to allow for an in-depth and comprehensive analysis of the code efficiency. Moreover, EFFIBENCH integrates a diverse set of efficiency metrics, such as execution time, maximum memory usage, and total memory usage during execution.

We conduct a comprehensive study to evaluate the efficiency of code generated by 42 LLMs. Our findings reveal that among both open- and closed-source LLMs, StarCoder2-15B [34] and GPT-4 consistently produced the most efficient code. Nevertheless, even these top performers still lag behind the efficiency of human-written canonical solutions. For instance, GPT-4 generated code exhibits an average execution time that is 3.12 times that of the human-written canonical solutions. In the most extreme cases, the execution time and total memory usage of GPT-4 code are **13.89** and **43.92** times that of the canonical solutions, respectively. Furthermore, our analysis reveals that a high pass@1 score (indicating the LLM's ability to generate correct code on the first attempt) does not necessarily translate to more efficient code. For example, GPT-4-turbo-preview has a higher pass@1 score than GPT-4, but lower code efficiency.

To conclude, this paper makes the following contributions:

- We introduce EFFIBENCH, the first benchmark specifically designed to assess the efficiency of code generated by LLMs.
- We conduct an extensive evaluation of 42 LLMs on EFFIBENCH, revealing that even state-of-the-art LLMs (e.g. GPT-4) exhibit significant inefficiencies compared to optimal human-written solutions.
- We release an efficiency testing framework[3], which enables evaluating the efficiency across various code generation benchmarks (See Appendix A.9).

## 2 Related Work

### 2.1 LLMs for Code

The burgeoning interest in LLMs for code has coincided with the profusion of openly available code repositories and the pressing need to enhance the productivity of software developers. Initial models predominantly focused on code generation tasks have included AlphaCode [31], CodeGen [39], CodeT5+ [52], InCoder [17], StarCoder [30], SantaCoder [5] and DeepSeek Coder [13], all of which were trained on code. Contrastingly, models such as Codex [12], Astraios [63], and CodeLLaMA [45] represent a subsequent stride, having been fine-tuned from foundation models [10, 49]. The evolution continued as LLMs leveraged instruction-like datasets derived from GPT [41, 42] for fine-tuning. Among these, WizardCoder [35] and Phi-3 [2] are notable examples. Across various coding applications, these code LLMs have set new standards of excellence, showcasing their prowess in domains including program repair [18, 26], automated testing [29, 14, 22, 24, 23], code translation [44, 3], type prediction [37, 54], and code summarization [20, 4].

### 2.2 Code Generation Benchmarks

Code generation [7, 12, 61, 55, 59] has emerged as a vital domain for evaluating LLMs, where models generate code snippets based on natural language descriptions, often given in the form of docstrings. Recent works try to improve HumanEval and MBPP from different perspectives. For example, HumanEval+ [32] enhances HumanEval with improved test cases, remedying the issue of mistakenly accepted faulty solutions. Meanwhile, ReCode [51] takes a different approach by altering function names and docstrings within the HumanEval structure. Expanding the scope

---

[3]We also make Github Repo public and then researchers can create issues in Github to evaluate the efficiency. Or they can directly use the docker and our public Hugging Face Server for efficiency calculation.

Table 1: Statistics of EFFIBENCH with different algorithms.

| Algorithm | Greedy | DP | Backtracking | Divide and Conquer | DFS | BFS | Binary Search | Two Pointers | Sliding Window | Bit Manipulation | Sorting | Total/Avg. |
|---|---|---|---|---|---|---|---|---|---|---|---|---|
| Number of problems | 243 | 277 | 48 | 21 | 108 | 86 | 148 | 105 | 70 | 102 | 238 | 1000 |
| Number of Easy problems | 32 | 8 | 1 | 4 | 18 | 8 | 23 | 39 | 9 | 26 | 63 | 171 |
| Number of Medium problems | 170 | 151 | 37 | 8 | 72 | 52 | 75 | 59 | 47 | 58 | 133 | 589 |
| Number of Hard problems | 41 | 118 | 10 | 9 | 18 | 26 | 50 | 7 | 14 | 18 | 42 | 240 |
| Avg. length of problem description | 224.8 | 216.4 | 162.0 | 205.1 | 218.9 | 239.7 | 216.4 | 198.6 | 188.7 | 195.0 | 220.7 | 212.0 |
| Avg. lines of Canonical Solution | 12.6 | 15.1 | 19.3 | 18.2 | 20.8 | 22.7 | 14.4 | 13.0 | 14.6 | 12.8 | 12.0 | 14.6 |

beyond Python, HumanEval-X [60], MultiPLe [11], and MBXP [6] extend the HumanEval and MBPP benchmarks to incorporate a variety of programming languages. The universe of code generation benchmarks widens further when we consider the specialized needs of data science. DS-1000 [28], ARCADE [56], NumpyEval [57], and PandasEval [25] focus on the generation of code within this context. Beyond mere code creation, there are benchmarks like APIBench [43], MTPB [38], RepoBench [33], ODEX [53], SWE-Bench [27], GoogleCodeRepo [47], RepoEval [58], and Cocomic-Data [15], which ratchet up the complexity by evaluating a model's prowess in utilizing APIs or completing broader software engineering tasks. Recent studies [46, 40] have indicated that code generated by LLMs tends to be less efficient in terms of execution time and memory usage compared to canonical solutions. To bridge this gap, our benchmark EFFIBENCH is specifically designed to evaluate the efficiency of code generation[4].

## 3 Benchmark Construction

### 3.1 Efficiency-critical Problem Collection

**Coding problem collection**    Inspired by the common practice [9, 19, 8] of using LeetCode problems to evaluate human developers' abilities in writing efficient algorithms, we collect the coding problems that appear on LeetCode. Specifically, we collect all problems tagged with "LeetCode" on the HuggingFace platform. We remove duplicate problems with identical problem IDs (each project has a unique ID in LeetCode). We also remove problems whose interview frequencies are lower than 40% at LeetCode. In the end, we obtain 2,605 problems as initial problem candidates.

**Efficiency-critical problem filtering**    This step selects efficiency-critical problems from the initial 2,605 problem candidates. The problems collected from HuggingFace are not tagged with algorithm topics. Therefore, we map each problem in LeetCode and label the problem with the "Topic" tag provided by LeetCode. We then choose typical algorithms (Table 1) that are introduced in common algorithm textbooks [48], which are also the most widely covered in Leetcode. This yields 1,146 problems altogether.

### 3.2 Canonical Solution Construction

For each coding problem, EFFIBENCH provides an executable canonical solution to serve as a baseline to calculate the normalised efficiency. Drawing inspiration from DS-1000 [28], which collects canonical solutions based on the most starred responses on Stack Overflow, we begin with collecting the top-starred solutions for each problem from the LeetCode Discussion Forum. For each collected solution, we need to guarantee that they are executable in a non-Leetcode environment. To this end, we manually fix the solutions that need to import extra classes such as TreeNode and ListNode as well as extra packages such as List and Bisect. We also remove the solutions that require specialized packages implemented only by LeetCode. In the end, we managed to map executable canonical solutions for 1,000 coding problems, which then be regarded as our final efficiency dataset.

### 3.3 Test Case Generation

It is essential to have adequate and diverse test cases to evaluate a program's efficiency across various scenarios. Since directly generating test cases with LLMs (e.g., GPT-3.5) requires large token overhead and has a low accuracy (See Appendix A.26), we develop a test case generator for each coding problem as an integral part of our benchmark construction. In particular, we require GPT-3.5-turbo to produce the test case generator, which is prompted to generate massive test cases with different input sizes, data distribution, and edge cases. Users can decide how many tests they

---

[4]A parallel work, Mercury [16], is also used to measure the efficiency of LLM-generated code.

would like to generate for each problem. We also provide 100 tests within EFFIBENCH for users to use directly, which also serve as the tests in our evaluation in this paper (Results with 10 tests and 1,000 tests are shown in Appendix Table 24).

## 3.4 Efficiency Metrics

Efficiency metrics are crucial for benchmarking code generation models automatically. Following LeetCode, we design automatic efficiency metrics from two aspects: execution time and memory usage. Specifically, we use the following metrics: Execution Time (ET), Normalized Execution Time (NET), Max Memory Usage (MU), Normalized Max Memory Usage (NMU), Total Memory Usage (TMU), and Normalized Total Memory Usage (NTMU) to measure the overall capability of a code generation model in generating efficient code.

**Execution Time (ET)**   Execution time (ET) measures the average time taken for code execution. Mathematically, ET is defined as:

$$ET = \frac{1}{N} \sum^{N} T_{\text{code}}$$

where $ET$ is the execution time metric, $T_{\text{code}}$ is the execution time of the code (with all the test cases), and $N$ is the number of codes generated by code generation models used for evaluation.

**Normalized Execution Time (NET)**   Normalized Execution Time (NET)[5] measures the execution time required by generated code relative to that of a canonical solution. We define NET as:

$$NET = \frac{1}{N} \sum^{N} \frac{T_{\text{code}}}{T_{\text{canonical}}}$$

where $T_{\text{code}}$ is the execution time of the generated code and $T_{\text{canonical}}$ is the execution time of the canonical solution. A NET value greater than 1 indicates that the generated code is slower than the canonical solution, while a value less than 1 suggests the generated code is faster.

**Max Memory Usage (MU)**   Max Memory Usage (MU) measures the average max memory consumption during code execution. Mathematically, MU is defined as:

$$MU = \frac{1}{N} \sum^{N} M_{\text{code}}$$

where $MU$ is the memory usage metric, $M_{\text{code}}$ is the max memory consumption of the generated code among all the test cases, and $N$ is the number of code instances generated by code generation models used for evaluation. This metric is critical to assess the resource efficiency of generated code, particularly in environments with limited maximum memory capacity.

**Normalized Max Memory Usage (NMU)**   Normalized Max Memory Usage (NMU) quantifies how the max memory efficiency of the generated code compares to the canonical solution. We define NMU as:

$$NMU = \frac{1}{N} \sum^{N} \frac{M_{\text{code}}}{M_{\text{canonical}}}$$

where $NMU$ is the normalized max memory usage metric, $M_{\text{code}}$ is the max memory usage of the generated code, and $M_{\text{canonical}}$ is the max memory usage of the canonical solution. An NMU value less than 1 indicates that the generated code is more memory-efficient than the canonical solution, whereas a value greater than 1 suggests it is less efficient in terms of memory usage. This metric provides a relative measure of the memory optimization in the generated code in comparison to a standard baseline.

---

[5]To demonstrate code-level efficiency, we evaluate the normalized efficiency metrics in task level, rather than total LLM-generated code / total canonical solutions. For the second calculation strategy, we also provide the scripts in our Github Repo.

**Total Memory Usage (TMU)**    Total Memory Usage (TMU) assesses the efficiency of memory usage throughout the execution of code, taking into account both the magnitude and duration of memory utilization. To calculate TMU, first, monitor and record the memory usage at discrete time intervals during the execution, resulting in a memory usage profile $M(t)$, where $t$ represents time. Then, compute the area under the curve of $M(t)$ over the total execution time, $T_{\text{total}}$, using numerical integration methods such as the trapezoidal rule:

$$TMU = \frac{1}{N} \sum^{N} \int_0^{T_{\text{total}}} M(t) \, dt$$

A lower TMU value indicates higher memory efficiency, reflecting an optimized balance between the amount of memory used and the duration of its usage.

**Normalized Total Memory Usage (NTMU)**    The Normalized Total Memory Usage (NTMU) offers a comparison of the dynamic memory efficiency between the generated code and the canonical solution. To determine NTMU, calculate the TMU for both the generated code and the canonical solution. Normalize the TMU of the generated code by dividing it by the TMU of the canonical solution:

$$NTMU = \frac{1}{N} \sum^{N} \frac{TMU_{\text{code}}}{TMU_{\text{canonical}}}$$

where $TMU_{\text{code}}$ is the TMU of the generated code and $TMU_{\text{canonical}}$ is the TMU of the canonical solution. An NTMU value less than 1 signifies that the generated code manages dynamic memory more efficiently compared to the canonical solution, while a value greater than 1 indicates less efficient management of dynamic memory. This metric provides insight into the relative use of dynamic memory of generated code compared to an established benchmark.

## 4    Benchmark Statistics

We provide the detailed statistics of the dataset in Table 1. The coding problems in EFFIBENCH have three difficulty levels (171 easy-level, 589 medium-level, and 240 hard-level problems), where the difficulty of each problem is defined by LeetCode [1]. The table lists the number of problems for each algorithm. Specifically, EFFIBENCH contains 243 problems for the greedy algorithm, 277 for dynamic programming (DP), 48 for backtracking, 21 for divide and conquer, 108 for depth-first search (DFS), 86 for breadth-first search (BFS), 148 for binary search, 105 for two pointers, 70 for sliding window, 102 for bit manipulation and 238 for sorting algorithm. The sum of problems in different algorithms can be larger than the number of total problems because one problem in our dataset may belong to two algorithm classes. On average, a problem description in EFFIBENCH contains 212.0 words. The canonical solutions, which represent the baseline code against which the generated code is compared, have 14.6 lines on average.

We provide a comparison of EFFIBENCH and other code generation datasets in Table 2. Specifically, we compare EFFIBENCH with the five most widely used code-related datasets (i.e., HumanEval, MBPP, APPS, DSP, and DS-1000). Different from the previous dataset that focuses on analyzing whether the code passes all test cases, EFFIBENCH also analyzes the efficiency during the code execution procedure. Although EFFIBENCH is primarily designed to assess the efficiency of generated code, it can also serve to evaluate code correctness, akin to other code generation datasets.

## 5    Evaluation

By default, the experiments are conducted in an edge server with an Intel Xeon Platinum 8336C CPU with 128 cores, 8 * NVIDIA A100-SXM GPUs, and a total memory capacity of 2.0TiB. We set the timeout for each code execution as 10 (s). The main goal of our work is to provide a benchmark that evaluates the efficiency of LLM-generated code within an identical environment, and we do expect that with different environments, the absolute values of the efficiency metrics would be different. We report results with different environments in Table 26, where our evaluation results demonstrate that despite the differences in absolute values, the ranking of LLMs is rather stable (p-value$>>0.05$ based on Kruskal-Wallis H tests). Besides, to provide a more reliable evaluation framework, we have also provided a server in the Hugging Face Space, where users can directly upload the code

Table 2: Comparison of EFFIBENCH to other code generation benchmarks. In addition to test cases, EFFIBENCH provides efficiency metrics and analysis for code generation models.

| Dataset | Number of Problems | Evaluation Support | Avg. Test Cases | Avg. Lines of Canonical Solution | Data Source | Assessment |
|---------|-------------------|-------------------|-----------------|----------------------------------|-------------|------------|
| HumanEval | 164 | Test Cases | 7.7 | 6.3 | Hand-Written | Correctness |
| MBPP | 974 | Test Cases | 3.0 | 6.7 | Crowd-sourced | Correctness |
| APPS | 10000 | Test Cases | 13.2 | 18.0 | Competitions | Correctness |
| DSP | 1119 | Test Cases | 2.1 | 4.5 | Notebooks | Correctness |
| DS-1000 | 1000 | Test Cases | 1.6 | 3.6 | StackOverflow | Correctness |
| **EFFIBENCH** (Ours) | 1000 | Test Cases + Efficiency metrics and analysis | Self-defined 100 by default | 14.6 | LeetCode | Efficiency and Correctness |

generation JSON file and then the server will execute the code locally and report the efficiency results with the same environment in the future.

**Models:** We evaluate both open- and closed-source LLMs in code generation. For open-source models, we evaluate EFFIBENCH with CodeLlama-hf family (i.e., 7B, 13b, 34b, and 70B), CodeLlama-Instruct-hf family (i.e., 7B, 13b, 34b, and 70B), deepseek-coder-instruct (i.e., 1.3B and 6.7B) and base models (i.e., 6.7B and 33B), Phind-CodeLlama-34B (i.e., v1 and v2), starcoder, starcoderbase, and starcoder2 (i.e., 3B, 7B, and 15B), WizardCoder (i.e., 13B and 15B), XwinCoder (i.e., 13B and 34B), Yi models (34B, 34B-Chat, and 200K version), and five widely proposed SOTA models, i.e., Magicoder-6.7B, Mistral-7B, octocoder, Artigenz-6.7B, CodeFuse-33B, and codegemma-7b[6] since these open-source models have obtained SOTA pass@1 in the HumanEval and MBPP datasets. For closed-source models, we evaluated EFFIBENCH with GPT-3.5, GPT-4 [42], and claude-3, since we observe that these models obtain high pass@1 in code generation datasets (e.g., HumanEval [12], MBPP [7]). For GPT-3.5 models, we experiment with GPT-3.5-turbo-0301, GPT-3.5-turbo-0613, and GPT-3.5-turbo-1106 which represent three different versions of the GPT-3.5. For GPT-4 models, we experiment with GPT-4-turbo and GPT-4 (GPT-4-0613). For the claude-3 model, we evaluate the sonnet and haiku versions. For each LLM, we first collect the code that is correctly generated for each coding problem (i.e., they can pass all test cases provided by the dataset), then execute these correct code and calculate the efficiency metrics (See Section 3.4).

**Prompt:** Our prompt follows the MBPP code generation prompt, where the prompt first provides the task description and then provides a few examples with input and output pairs. Each example has an explanation of the rationality of the output. The prompt also has the assertion part, which intends to constrain the function signature with the input and output format.

## 5.1 End2End Results

**Open-source models** The evaluation results of open-source models are illustrated in Table 3. Our evaluation results demonstrate that **all open-source models' generated code requires more overhead than the human-written canonical solutions**. For example, StarCoder2-15B, the most efficient open-source model in terms of NET, NMU, and NTMU, on average still needs 2.59x execution time, 1.71x max memory usage (i.e., memory peak), and 4.83x total memory usage during the code execution compared with the canonical solutions. We suspect that this is because human-written canonical solutions, while optimal, are in the minority within the training data of these LLMs. Consequently, the LLMs tend to learn non-optimal solutions, which are more frequently distributed in the training data. In addition, our results demonstrate that open-source LLMs with lower pass@1 tend to have better efficiency. The key reason is that these LLMs can only generate correct code on relatively simple problems, which makes it easier to achieve efficiency compared to more complex and challenging problems (see Table 27-29).

**Closed-source models** The evaluation results of closed-source models are demonstrated in the bottom part of Table 3. Our results illustrate that similar to open-source models, all closed-source models generated code still need more overhead than the canonical solution on average. Despite GPT-4 generated code obtaining the most efficient results for closed-source models, its generated code still needs on average 3.12x execution time and 6.36x total memory usage during the code execution compared with the canonical solution. In the worst case, the execution time is almost 14x that of the canonical solution. In addition, **although consistent training can improve the**

---

[6]The model names are extracted from Hugging Face model card.

Table 3: Code efficiency of widely-studied LLMs reported by EFFIBENCH. In addition to the mean values of the basic metrics introduced in Section 3.4, we also report the maximum normalised execution time/memory among all the generated correct code (e.g., Column "max NET") and the ratio of problems with normalised metric value larger than 5 (e.g., Column "NET>5") in the correct code. The most efficient result for each metric is highlighted in grey.

| Model | max NET | NET | NET>5 | ET (s) | max NMU | NMU | NMU>5 | MU (Mb) | max NTMU | NTMU | NTMU>5 | TMU (Mb*s) | Pass@1 |
|---|---|---|---|---|---|---|---|---|---|---|---|---|---|
| **Open-source models** | | | | | | | | | | | | | |
| CodeLlama-7b-hf | 3.25 | 2.95 | 0.0 | 0.31 | 2.05 | 1.98 | 0.0 | 48.59 | 6.80 | 6.03 | 100.0 | 9.99 | 1.1 |
| CodeLlama-13b-hf | 3.21 | 2.71 | 0.0 | 0.40 | 2.05 | 1.85 | 0.0 | 104.42 | 6.53 | 5.32 | 81.8 | 43.83 | 1.1 |
| CodeLlama-34b-hf | 4.46 | 2.98 | 0.0 | 0.34 | 2.06 | 1.92 | 0.0 | 55.38 | 9.17 | 6.01 | 92.9 | 13.41 | 8.4 |
| CodeLlama-70b-hf | 13.92 | 3.19 | 4.4 | 0.42 | 2.06 | 1.90 | 0.0 | 62.41 | 32.04 | 6.47 | 87.8 | 22.27 | 9.0 |
| CodeLlama-7b-Instruct-hf | 17.26 | 3.44 | 4.2 | 0.46 | 3.59 | 1.94 | 0.0 | 77.87 | 56.61 | 7.65 | 87.5 | 32.14 | 4.8 |
| CodeLlama-13b-Instruct-hf | 4.46 | 2.93 | 0.0 | 0.35 | 2.48 | 1.92 | 0.0 | 65.96 | 10.22 | 5.94 | 91.6 | 18.74 | 8.4 |
| CodeLlama-34b-Instruct-hf | 13.66 | 3.04 | 0.9 | 0.37 | 2.56 | 1.93 | 0.0 | 61.31 | 31.46 | 6.16 | 87.4 | 18.53 | 11.1 |
| CodeLlama-70b-Instruct-hf | 14.60 | 3.07 | 1.4 | 0.38 | 2.06 | 1.93 | 0.0 | 54.04 | 33.69 | 6.27 | 90.3 | 18.27 | 7.2 |
| deepseek-coder-1.3b-instruct | 3.63 | 2.82 | 0.0 | 0.33 | 2.03 | 1.91 | 0.0 | 57.73 | 8.13 | 5.69 | 88.9 | 13.11 | 4.5 |
| deepseek-coder-6.7b-instruct | 5.59 | 2.89 | 1.4 | 0.38 | 2.57 | 1.90 | 0.0 | 73.73 | 13.81 | 5.86 | 88.4 | 26.84 | 6.9 |
| deepseek-coder-6.7b-base | 12.25 | 2.98 | 1.2 | 0.37 | 2.14 | 1.91 | 0.0 | 62.78 | 23.39 | 6.01 | 89.7 | 19.55 | 16.5 |
| deepseek-coder-33b-base | 19.54 | 3.14 | 1.3 | 0.38 | 37.39 | 2.08 | 0.4 | 60.30 | 604.13 | 8.76 | 91.9 | 22.05 | 23.5 |
| OpenCodeInterpreter-DS-1.3B | 3.93 | 2.89 | 0.0 | 0.35 | 2.05 | 1.91 | 0.0 | 68.25 | 8.44 | 5.82 | 87.0 | 21.88 | 5.5 |
| OpenCodeInterpreter-DS-6.7B | 6.03 | 2.95 | 1.5 | 0.37 | 2.37 | 1.91 | 0.0 | 63.41 | 14.14 | 5.96 | 87.9 | 19.17 | 13.2 |
| OpenCodeInterpreter-DS-33B | 26.06 | 3.15 | 1.7 | 0.37 | 2.43 | 1.91 | 0.0 | 59.37 | 66.25 | 6.48 | 88.2 | 18.34 | 23.7 |
| Phind-CodeLlama-34B-v1 | 3.57 | 2.91 | 0.0 | 0.36 | 2.06 | 1.90 | 0.0 | 67.63 | 7.76 | 5.83 | 88.0 | 22.61 | 11.7 |
| Phind-CodeLlama-34B-v2 | 53.08 | 3.28 | 1.0 | 0.42 | 2.60 | 1.89 | 0.0 | 70.53 | 139.88 | 6.80 | 86.4 | 26.24 | 19.1 |
| starcoder | 3.34 | 2.84 | 0.0 | 0.33 | 2.06 | 1.91 | 0.0 | 65.23 | 6.88 | 5.69 | 85.3 | 17.67 | 3.4 |
| starcoder2-3b | 3.13 | 2.90 | 0.0 | 0.31 | 2.04 | 1.94 | 0.0 | 51.58 | 6.61 | 5.87 | 92.3 | 10.55 | 1.3 |
| starcoder2-7b | 5.19 | 3.02 | 6.7 | 0.32 | 2.46 | 1.98 | 0.0 | 48.55 | 12.69 | 6.29 | 100.0 | 10.63 | 1.5 |
| starcoder2-15b | 3.20 | **2.59** | **0.0** | 0.43 | **2.01** | **1.71** | **0.0** | 122.52 | 6.59 | **4.83** | **57.1** | 47.39 | 0.7 |
| starcoderbase | 3.34 | 2.80 | 0.0 | 0.35 | 2.05 | 1.87 | 0.0 | 74.94 | 7.09 | 5.56 | 80.0 | 21.87 | 2.0 |
| WizardCoder-13B | 16.48 | 3.13 | 2.9 | 0.46 | 3.57 | 1.90 | 0.0 | 80.77 | 53.63 | 6.76 | 76.5 | 30.74 | 3.4 |
| WizardCoder-15B | 4.07 | 2.84 | 0.0 | 0.35 | 2.06 | 1.91 | 0.0 | 72.72 | 9.51 | 5.73 | 83.3 | 20.63 | 3.0 |
| XwinCoder-13B | 4.16 | 2.94 | 0.0 | 0.33 | 2.05 | 1.95 | 0.0 | 57.70 | 8.95 | 5.99 | 92.8 | 14.40 | 8.4 |
| XwinCoder-34B | 6.32 | 2.98 | 0.5 | 0.34 | 2.42 | 1.92 | 0.0 | 57.92 | 17.70 | 6.03 | 87.5 | 14.31 | 18.4 |
| Yi-34B-200K | 3.17 | 2.91 | 0.0 | 0.31 | 2.06 | 1.96 | 0.0 | 49.88 | 6.78 | 5.94 | 91.7 | 10.23 | 3.6 |
| Yi-34B-Chat | 3.15 | 2.77 | 0.0 | 0.34 | 2.05 | 1.89 | 0.0 | 68.99 | 6.69 | 5.52 | 89.3 | 19.09 | 2.8 |
| Yi-34B | 3.38 | 2.81 | 0.0 | 0.37 | 2.05 | 1.89 | 0.0 | 83.42 | 7.13 | 5.62 | 88.5 | 26.71 | 2.6 |
| Artigenz-Coder-DS-6.7B | 27.78 | 3.22 | 1.6 | 0.39 | 2.48 | 1.91 | 0.0 | 62.13 | 70.28 | 6.65 | 90.9 | 19.72 | 36.4 |
| CodeFuse-DeepSeek-33B | 6.10 | 3.07 | 0.3 | 0.36 | 2.06 | 1.91 | 0.0 | 58.30 | 15.19 | 6.21 | 87.6 | 16.45 | 29.2 |
| codegemma-7b | 8.09 | 3.02 | 0.8 | 0.34 | 2.06 | 1.93 | 0.0 | 55.68 | 20.96 | 6.15 | 92.2 | 13.78 | 12.8 |
| Magicoder-S-DS-6.7B | 6.73 | 2.99 | 0.6 | 0.35 | 2.61 | 1.91 | 0.0 | 60.12 | 14.24 | 6.05 | 89.0 | 16.84 | 36.3 |
| Mistral-7B-codealpaca-lora | 3.82 | 2.85 | 0.0 | 0.31 | 2.36 | 1.95 | 0.0 | 51.51 | 9.20 | 5.81 | 88.5 | 10.50 | 2.6 |
| octocoder | **2.99** | 2.67 | 0.0 | 0.32 | 2.02 | 1.84 | 0.0 | 58.98 | **6.20** | 5.07 | 75.0 | 11.52 | 0.4 |
| **Closed-source models** | | | | | | | | | | | | | |
| gpt-3.5-turbo-0301 | 27.70 | 3.18 | 1.4 | 0.39 | **2.05** | **1.91** | 0.0 | 60.53 | 70.62 | 6.50 | **89.1** | 19.06 | 42.3 |
| gpt-3.5-turbo-0613 | 46.70 | 3.22 | 0.9 | 0.39 | 2.64 | 1.92 | 0.0 | 59.82 | 161.12 | 6.71 | 89.9 | 19.11 | 46.4 |
| gpt-3.5-turbo-1106 | 68.71 | 3.40 | 1.6 | 0.40 | 9.12 | 1.94 | 0.2 | 59.34 | 182.63 | 7.24 | 90.9 | 19.39 | 49.3 |
| gpt-4 | **13.89** | **3.12** | 1.0 | **0.37** | 2.25 | 1.92 | **0.0** | 58.85 | **43.92** | **6.36** | 91.1 | 17.69 | 50.8 |
| gpt-4-turbo-preview | 27.00 | 3.19 | 1.2 | 0.38 | 9.13 | 1.93 | 0.2 | **57.06** | 68.48 | 6.57 | 91.1 | **16.92** | 65.4 |
| claude-3-haiku | 28.75 | 3.28 | **0.7** | 0.39 | 2.05 | 1.91 | 0.0 | 59.15 | 72.87 | 6.71 | 90.0 | 17.99 | 42.9 |
| claude-3-sonnet | 17.43 | 3.22 | 0.9 | 0.40 | 2.06 | 1.91 | 0.0 | 60.22 | 50.78 | 6.57 | 90.5 | 23.29 | 43.2 |

Table 4: Efficiency results of closed-source LLMs with 210 problems correctly addressed by all models in the Table. Although GPT-3.5-turbo models have the same ET (i.e., 0.37s), the NET is not the same since the task level NET does not have the same distribution (e.g., the max NET of the 0301 model is 16.24x while it only requires 4.05x in 0613 model).

| Model | max NET | NET | NET>5 | ET (s) | max NMU | NMU | NMU>5 | MU (Mb) | max NTMU | NTMU | NTMU>5 | TMU (Mb*s) |
|---|---|---|---|---|---|---|---|---|---|---|---|---|
| gpt-3.5-turbo-0301 | 16.24 | 3.10 | 0.5 | 0.37 | **2.05** | 1.90 | 0.0 | 66.91 | 46.95 | 6.32 | 88.6 | 20.89 |
| gpt-3.5-turbo-0613 | 4.05 | **3.05** | 0.0 | 0.37 | 2.64 | 1.90 | 0.0 | 66.99 | 10.21 | 6.18 | 89.5 | 20.92 |
| gpt-3.5-turbo-1106 | 6.12 | 3.07 | 0.5 | 0.37 | 2.06 | 1.90 | 0.0 | 66.94 | 15.53 | 6.22 | 89.0 | **20.78** |
| gpt-4 | **4.04** | 3.06 | **0.0** | **0.37** | 2.06 | **1.90** | **0.0** | 66.91 | 9.22 | **6.17** | 89.0 | 21.17 |
| gpt-4-turbo-preview | 4.09 | 3.10 | 0.0 | 0.37 | 2.05 | 1.90 | 0.0 | 66.92 | **8.92** | 6.28 | 89.0 | 20.78 |
| claude-3-haiku | 11.06 | 3.27 | 0.5 | 0.39 | 2.05 | 1.90 | 0.0 | **66.90** | 29.68 | 6.68 | 89.0 | 22.52 |
| claude-3-sonnet | 17.43 | 3.20 | 0.5 | 0.38 | 2.06 | 1.90 | 0.0 | 66.93 | 50.78 | 6.55 | 89.0 | 21.52 |

**correctness of LLM-generated code, the efficiency of LLM-generated code may not improve**. For example, the pass@1 for GPT-3.5-turbo increases from 42.3% to 49.3% when the model version is updated from 0301 to the 1106 version, the execution time of the code generated by GPT-3.5-turbo increases from 3.18x to 3.40x.

**Consistency of different metrics**: When we compare the benchmarking results from different efficiency metrics, we can observe that the rankings of different LLMs from the basic metrics (highlighted in bold in the head row) maintain a general consistency. For example, in closed-source models, GPT-4 obtains the most efficient results in the majority of metrics. Yet, for other metrics where GPT-4 does not get the highest efficiency, the code generated by GPT-4 is also close to the most efficient LLM-generated ones. This consistency across metrics reinforces their credibility in assessing a model's capability to generate efficient code.

**Correctness**: Although EFFIBENCH is designed to focus on benchmarking efficiency of LLM-generated code, it can also be adapted to benchmark code correctness, as shown by pass@1 in the last column of Table 3. For open-sourced LLMs, our results demonstrate that they have low pass@1:

Table 5: Efficiency results for different algorithm subsets with closed-source LLMs.

| Model | max NET | NET | NET>5 | ET (s) | max NMU | NMU | NMU>5 | MU (Mb) | max NTMU | NTMU | NTMU>5 | TMU (Mb*s) | Pass@1 |
|---|---|---|---|---|---|---|---|---|---|---|---|---|---|
| GPT-3.5-turbo-0301 | | | | | | | | | | | | | |
| greedy | 3.63 | 3.02 | 0.0 | 0.35 | 2.03 | 1.92 | 0.0 | 59.42 | 7.51 | 6.14 | 90.7 | 16.75 | 39.9 |
| dynamic_programming | 27.70 | 3.64 | 4.5 | 0.46 | 2.05 | 1.93 | 0.0 | 55.25 | 70.62 | 7.73 | 89.3 | 21.44 | 40.4 |
| backtracking | 14.99 | 3.44 | 4.5 | 0.56 | 2.03 | 1.82 | 0.0 | 83.45 | 34.36 | 6.90 | 72.7 | 38.40 | 45.8 |
| divide_and_conquer | 3.53 | 3.00 | 0.0 | 0.34 | 2.02 | 1.89 | 0.0 | 53.42 | 7.00 | 5.96 | 87.5 | 11.41 | 38.1 |
| dfs | 3.47 | 2.91 | 0.0 | 0.35 | 2.05 | 1.81 | 0.0 | 59.62 | 6.82 | 5.68 | 85.2 | 13.60 | 25.0 |
| bfs | 6.35 | 3.17 | 4.2 | 0.41 | 2.05 | 1.90 | 0.0 | 55.10 | 13.56 | 6.43 | 91.7 | 15.99 | 27.9 |
| binary_search | 3.61 | 2.92 | 0.0 | 0.38 | 2.05 | 1.87 | 0.0 | 79.97 | 7.39 | 5.83 | 85.7 | 27.09 | 42.6 |
| two_pointers | 3.61 | 3.04 | 0.0 | 0.36 | 2.04 | 1.94 | 0.0 | 70.22 | 7.37 | 6.24 | 94.2 | 25.77 | 49.5 |
| sliding_window | 3.87 | 3.04 | 0.0 | 0.36 | 2.05 | 1.94 | 0.0 | 67.21 | 8.22 | 6.20 | 91.4 | 23.55 | 50.0 |
| bit_manipulation | 3.59 | 3.03 | 0.0 | 0.35 | 2.02 | 1.94 | 0.0 | 62.42 | 7.61 | 6.15 | 89.6 | 19.40 | 47.1 |
| sorting | 3.76 | 2.99 | 0.0 | 0.36 | 2.05 | 1.88 | 0.0 | 67.09 | 8.05 | 6.01 | 87.9 | 21.95 | 41.6 |
| GPT-4 | | | | | | | | | | | | | |
| greedy | 5.83 | 3.08 | 0.8 | 0.35 | 2.04 | 1.93 | 0.0 | 57.15 | 15.28 | 6.32 | 92.7 | 15.74 | 50.6 |
| dynamic_programming | 4.53 | 3.11 | 0.0 | 0.36 | 2.25 | 1.94 | 0.0 | 53.97 | 10.16 | 6.31 | 91.3 | 15.44 | 49.8 |
| backtracking | 4.53 | 3.01 | 0.0 | 0.44 | 2.03 | 1.84 | 0.0 | 81.67 | 10.16 | 5.89 | 77.3 | 32.23 | 45.8 |
| divide_and_conquer | 3.68 | 3.04 | 0.0 | 0.34 | 2.02 | 1.90 | 0.0 | 53.16 | 7.94 | 6.15 | 87.5 | 11.72 | 38.1 |
| dfs | 3.82 | 3.05 | 0.0 | 0.35 | 2.06 | 1.88 | 0.0 | 57.57 | 7.72 | 6.09 | 93.9 | 13.32 | 30.6 |
| bfs | 11.22 | 3.38 | 5.6 | 0.45 | 2.06 | 1.87 | 0.0 | 55.58 | 25.19 | 6.85 | 91.7 | 19.23 | 41.9 |
| binary_search | 3.69 | 2.96 | 0.0 | 0.38 | 2.04 | 1.88 | 0.0 | 75.09 | 7.78 | 5.92 | 89.3 | 25.24 | 50.7 |
| two_pointers | 3.94 | 3.09 | 0.0 | 0.36 | 2.04 | 1.94 | 0.0 | 66.90 | 8.90 | 6.36 | 95.2 | 23.65 | 59.0 |
| sliding_window | 8.46 | 3.23 | 2.5 | 0.39 | 2.06 | 1.92 | 0.0 | 66.36 | 17.85 | 6.60 | 95.0 | 25.41 | 57.1 |
| bit_manipulation | 4.53 | 3.12 | 0.0 | 0.36 | 2.03 | 1.95 | 0.0 | 60.22 | 10.16 | 6.39 | 92.6 | 18.60 | 52.9 |
| sorting | 13.89 | 3.11 | 1.5 | 0.38 | 2.25 | 1.89 | 0.0 | 63.62 | 43.92 | 6.40 | 90.0 | 21.09 | 54.6 |
| Claude-3-sonnet | | | | | | | | | | | | | |
| greedy | 3.75 | 3.13 | 0.0 | 0.36 | 2.03 | 1.93 | 0.0 | 58.47 | 7.90 | 6.39 | 90.3 | 16.68 | 42.4 |
| dynamic_programming | 16.34 | 3.42 | 1.8 | 0.47 | 2.04 | 1.94 | 0.0 | 54.95 | 37.83 | 6.96 | 92.0 | 35.81 | 40.8 |
| backtracking | 17.43 | 4.92 | 13.3 | 0.75 | 2.04 | 1.89 | 0.0 | 89.79 | 50.78 | 11.28 | 86.7 | 53.91 | 31.2 |
| divide_and_conquer | 3.56 | 3.03 | 0.0 | 0.36 | 2.02 | 1.88 | 0.0 | 53.44 | 7.18 | 6.01 | 75.0 | 12.62 | 57.1 |
| dfs | 3.61 | 3.03 | 0.0 | 0.36 | 2.05 | 1.81 | 0.0 | 59.20 | 7.53 | 5.94 | 86.2 | 13.98 | 26.9 |
| bfs | 6.24 | 3.08 | 3.4 | 0.42 | 2.05 | 1.84 | 0.0 | 59.57 | 13.17 | 6.06 | 86.2 | 16.36 | 33.7 |
| binary_search | 3.61 | 2.99 | 0.0 | 0.40 | 2.04 | 1.87 | 0.0 | 80.89 | 7.60 | 5.98 | 83.6 | 28.93 | 41.2 |
| two_pointers | 3.61 | 3.18 | 0.0 | 0.38 | 2.05 | 1.94 | 0.0 | 70.62 | 7.53 | 6.54 | 94.1 | 27.10 | 48.6 |
| sliding_window | 3.69 | 3.13 | 0.0 | 0.36 | 2.06 | 1.95 | 0.0 | 64.09 | 7.77 | 6.41 | 95.2 | 22.12 | 60.0 |
| bit_manipulation | 17.43 | 3.51 | 2.4 | 0.40 | 2.02 | 1.95 | 0.0 | 63.19 | 50.78 | 7.56 | 92.9 | 22.32 | 41.2 |
| sorting | 4.98 | 3.10 | 0.0 | 0.37 | 2.04 | 1.89 | 0.0 | 64.34 | 11.81 | 6.27 | 89.2 | 20.90 | 50.4 |

many of their pass@1 are lower than 10% (i.e., 23 out of 35 models), which indicates that open-source models still need to put a lot of effort into improving code generation correctness. For closed-sourced LLMs, GPT-4-turbo-preview has the highest pass@1 of 65.4%.

## 5.2 Results with Identical Coding Problems

In Table 3, we directly calculate the efficiency of the correct code for each model. However, different LLMs may have different correctness for the same coding problem. As a result, the results for different LLMs in Table 3 are based on different coding problems. In this section, we mitigate such threats by by analyzing the efficiency results with identical coding problems. In other words, we focus on analyzing problems correctly addressed by all LLMs. Since open-source LLMs do not have overlap for the tasks that are correctly generated, we only report results on closed-source LLMs only. The evaluation results are shown in Table 4, which contains 210 problems that have been correctly addressed by all closed-source LLMs. The evaluation results demonstrate that the results of each metric are slightly different from those shown in Table 3. Overall, **GPT models outperform Claude models in code efficiency, with GPT-4 achieving the highest efficiency as measured by most efficiency metrics**.

## 5.3 Results for Different Algorithms

As shown in Table 1, EFFIBENCH is constructed with 11 different algorithms[7]. In this section, we explore whether the LLMs have different code efficiency across different algorithm subsets. Table 7 reports the results of three closed-source LLMs for different algorithm subsets. Our results demonstrate that LLMs have different code efficiency for different algorithm subsets. For example, GPT-3.5-turbo-0301 is less efficient for dynamic programming (DP), which requires 7.73x total memory usage during the code execution procedure. In contrast, GPT-3.5-turbo-0301 demonstrated higher efficiency in the DFS and binary search subset, which only requires 5.68x and 5.83x NTMU compared with the canonical solution. We indicate that the observed differences come from the availability of training data. Specifically, models tend to perform better on tasks for which their training corpus contains abundant and varied examples with efficient solutions.

---

[7]Note that the task is classified as a specific algorithm but the code generated by LLMs may consider addressing the task with other algorithms.

Table 6: Evaluation results of Top-10 inefficient code generated by GPT-3.5-turbo-0301. We manually analyze the algorithm of each code.

| Metrics | Greedy | DP | Backtracking | Divide and Conquer | DFS | BFS | Binary Search | Two Pointers | Sliding Window | Bit Manipulation | Sorting |
|---|---|---|---|---|---|---|---|---|---|---|---|
| NET | 0 | 1 | 2 | 0 | 0 | 1 | 0 | 1 | 2 | 3 | 0 |
| NMU | 1 | 1 | 1 | 0 | 0 | 1 | 0 | 1 | 1 | 2 | 2 |
| NTMU | 3 | 4 | 1 | 0 | 0 | 1 | 0 | 0 | 0 | 1 | 0 |

## 5.4 Worst Case Analysis

In this section, we conduct a study to analyze the inefficient code generated by GPT-3.5-turbo-0301 (similar to the analysis in Section 5.3). Specifically, we collect the 10 most inefficient pieces of code for NET, NMU, and NTMU metrics and then manually analyze the implementation algorithm used by each code. The evaluation results are demonstrated in Table 6. The evaluation results demonstrate that the majority of the inefficient pieces of code are associated with DP and backtracking algorithms, with these categories showing the highest occurrences across the metrics. In particular, DP and backtracking algorithms show the highest counts in NTMU, indicating that these algorithms tend to generate code with higher memory consumption inefficiency, which highlights the areas where GPT-3.5-turbo-0301 struggles the most, suggesting a need for further optimization in generating code for complex algorithmic tasks.

To further understand the reasons for inefficiency in the LLM-generated code, we conduct a case comparison of GPT-3.5-turbo-0301 generated code and canonical solution in DP subset to analyze why LLM-generated code is inefficient. As shown in Figure 2, we can observe that the key reason for GPT-3.5-turbo-0301 being less efficient than the *canonical_solution* is due to the code generated by GPT-3.5-turbo-0301 first generating a 2-dimensional matrix which requires large overhead for memory usage when the parameters $n$ and $k$ are very large. However, the *canonical_solution* generates two lists, which significantly reduces the memory usage for the code. GPT-3.5-Turbo-0301 implements a straightforward dynamic programming approach with a complete matrix to keep track of results for every possible pair of $n$ and $k$, while the canonical solution optimizes by maintaining a rolling sum, which helps to reduce the space complexity from $O(n \times k)$ to $O(k)$, leading to a more memory-efficient implementation. This optimization in the canonical solution results in a significant performance improvement. Specifically, GPT-3.5-turbo-0301 generated code has 70.62x memory usage during the code execution compared with *canonical_solution*.

## 6 Conclusion and Future work

In this paper, we introduce EFFIBENCH, a benchmark designed to evaluate the efficiency of code generated by various code generation models. EFFIBENCH encompasses 1,000 problems and consists of 11 distinct algorithmic subsets. Unlike previous benchmarks that primarily emphasize the correctness of code generation, EFFIBENCH extends the evaluation criteria to include both execution time analysis and memory usage analysis. We also provide the evaluation server in Hugging Face to allow researchers to evaluate their methods with the same hardware and software. By incorporating these metrics and the Hugging Face server, EFFIBENCH aims to inspire the research community's focus towards not only the correctness but also the efficiency and sustainability of code generated by code generation models. In the future, we will consider extending EFFIBENCH with other programming languages (e.g., C++, Java, JS, and Go).

## 7 ACKNOWLEDGMENT

The work is supported in part by National Key R&D Program of China (2022ZD0160201), HK RGC RIF (R7030-22), HK ITF (GHP/169/20SZ), a Huawei Flagship Research Grant in 2023, HK RGC GRF (Ref: 17208223 & 17204424), and the HKU-CAS Joint Laboratory for Intelligent System Software.

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

# A Appendix

## A.1 Limiations

While EFFIBENCH represents a significant step towards evaluating code efficiency in code generation models, it currently has several limitations:

**Language Focus**: The benchmark is currently limited to Python and does not encompass other programming languages. This restricts the scope of the evaluation and prevents a comprehensive understanding of efficiency across different language paradigms.

**Dataset Scope**: EFFIBENCH focuses solely on LeetCode problems, which primarily involve algorithmic challenges. This excludes real-world applications and other coding scenarios that might necessitate different efficiency considerations.

**Environment Dependency**: The efficiency results obtained using EFFIBENCH may vary across different hardware and software environments. This highlights the need for standardized testing environments to ensure consistent and reliable comparisons between models. To address this limitation, we provide the **request link** in our Hugging Face Leaderboard for researchers to evaluate their pre-trained LLMs generated code efficiency, which uses the same environment for efficiency testing. In the future, we will also set up an efficiency testing server in Hugging Face Space for researchers to automatically get the efficiency metrics for LLM-generated code.

## A.2 Improvement Strategies

To address the limitations of EFFIBENCH, we propose several improvement strategies as follows:

**Broadening Language Coverage**: Recognizing the importance of a diverse range of programming languages, we aim to expand the benchmark beyond Python in the future. This allows for a more comprehensive evaluation of code efficiency across different language paradigms, ultimately providing a more holistic understanding of the performance of code generation models.

**Enhancing Dataset Diversity**: To ensure that EFFIBENCH is representative of a wide array of coding scenarios, we plan to incorporate more diverse datasets into our evaluation framework. While LeetCode problems offer valuable insights into algorithmic efficiency, we understand the need to consider real-world applications and other coding contexts. As a starting step, we have provided an efficiency testing framework that can be used with other datasets, such as HumanEval [12] and MBPP [7]. Moving forward, we will continue to seek out and integrate datasets that can enrich our understanding of code efficiency.

**Standardizing Testing Environments**: To address the variability in efficiency results due to different hardware and software environments, we are committed to establishing more standardized testing conditions. We have already taken a step in this direction by providing a request link in our Hugging Face Leaderboard for researchers to evaluate their LLMs generated code efficiency, which ensures that the same environment is used for testing. We also plan to set up an efficiency testing server, potentially hosted on Hugging Face Space, where developers can automatically obtain efficiency metrics for their LLM-generated code, which not only promotes consistency and reliability in our results but also makes the testing process more convenient and accessible for our users.

## A.3 Broader Impacts

We list the potential positive societal impacts as follows:

**Improved Software Efficiency** By benchmarking and improving the efficiency of code generated by LLMs, we can develop software that runs faster, consumes less memory and processing power. This can lead to more responsive applications, reduced operational costs, and a better user experience.

**Environmental Sustainability** More efficient code can contribute to reduced energy consumption, which is beneficial for the environment. This aligns with global efforts to reduce carbon emissions and promote sustainability.

**Enhanced Developer Productivity** LLMs can significantly augment developer productivity by generating code snippets based on coding instructions and offering intelligent recommendations. This can free up developers' time to focus on more complex tasks.

**Scalable Software Development**  Efficient code is crucial for building scalable software to meet the growing demands of the digital world. By improving the efficiency of code generated by LLMs, we can develop software that can handle larger volumes of data and users.

On the other hand, we summarize the potential negative societal impacts as follows:

**Job Displacement**  The increased use of LLMs in code generation could potentially lead to job displacement for some software developers in the future, particularly those involved in more routine coding tasks.

**Over-reliance on AI**  Developers may become overly reliant on LLMs, which could lead to a lack of understanding of the generated code and potential security or functionality issues.

**Security Risks**  If not properly managed, the use of LLMs could introduce security risks. For example, LLMs might generate code with vulnerabilities that could be exploited by malicious actors.

**Quality Concerns**  While LLMs can generate efficient code, the quality of the code in terms of readability, maintainability, and adherence to coding standards may not always meet the desired levels. This could lead to difficulties in code maintenance and development in the long term.

## A.4  Efficiency Metrics

**Execution Time (ET)**  Execution time (ET) measures the average time taken for code execution. Mathematically, ET is defined as:

$$ET = \frac{1}{N} \sum^{N} T_{\text{code}}$$

where $ET$ is the execution time metric, $T_{\text{code}}$ is the execution time of the code (with all the test cases), and $N$ is the number of codes generated by code generation models used for evaluation.

**Normalized Execution Time (NET)**  Normalized Execution Time (NET)[8] measures the execution time required by generated code relative to that of a canonical solution. We define NET as:

$$NET = \frac{1}{N} \sum^{N} \frac{T_{\text{code}}}{T_{\text{canonical}}}$$

where $T_{\text{code}}$ is the execution time of the generated code and $T_{\text{canonical}}$ is the execution time of the canonical solution. A NET value greater than 1 indicates that the generated code is slower than the canonical solution, while a value less than 1 suggests the generated code is faster.

**Max Memory Usage (MU)**  Max Memory Usage (MU) measures the average max memory consumption during code execution. Mathematically, MU is defined as:

$$MU = \frac{1}{N} \sum^{N} M_{\text{code}}$$

where $MU$ is the memory usage metric, $M_{\text{code}}$ is the max memory consumption of the generated code among all the test cases, and $N$ is the number of code instances generated by code generation models used for evaluation. This metric is critical to assess the resource efficiency of generated code, particularly in environments with limited maximum memory capacity.

**Normalized Max Memory Usage (NMU)**  Normalized Max Memory Usage (NMU) quantifies how the max memory efficiency of the generated code compares to the canonical solution. We define NMU as:

$$NMU = \frac{1}{N} \sum^{N} \frac{M_{\text{code}}}{M_{\text{canonical}}}$$

where $NMU$ is the normalized max memory usage metric, $M_{\text{code}}$ is the max memory usage of the generated code, and $M_{\text{canonical}}$ is the max memory usage of the canonical solution. An NMU value

---

[8]To demonstrate code-level efficiency, we evaluate the normalized efficiency metrics in task level, rather than total LLM-generated code / total canonical solutions. For the second calculation strategy, we also provide the scripts in our Github Repo.

less than 1 indicates that the generated code is more memory-efficient than the canonical solution, whereas a value greater than 1 suggests it is less efficient in terms of memory usage. This metric provides a relative measure of the memory optimization in the generated code in comparison to a standard baseline.

**Total Memory Usage (TMU)** Total Memory Usage (TMU) assesses the efficiency of memory usage throughout the execution of code, taking into account both the magnitude and duration of memory utilization. To calculate TMU, first, monitor and record the memory usage at discrete time intervals during the execution, resulting in a memory usage profile $M(t)$, where $t$ represents time. Then, compute the area under the curve of $M(t)$ over the total execution time, $T_{\text{total}}$, using numerical integration methods such as the trapezoidal rule:

$$TMU = \frac{1}{N} \sum^{N} \int_0^{T_{\text{total}}} M(t) \, dt$$

A lower TMU value indicates higher memory efficiency, reflecting an optimized balance between the amount of memory used and the duration of its usage.

**Normalized Total Memory Usage (NTMU)** The Normalized Total Memory Usage (NTMU) offers a comparison of the dynamic memory efficiency between the generated code and the canonical solution. To determine NTMU, calculate the TMU for both the generated code and the canonical solution. Normalize the TMU of the generated code by dividing it by the TMU of the canonical solution:

$$NTMU = \frac{1}{N} \sum^{N} \frac{TMU_{\text{code}}}{TMU_{\text{canonical}}}$$

where $TMU_{\text{code}}$ is the TMU of the generated code and $TMU_{\text{canonical}}$ is the TMU of the canonical solution. An NTMU value less than 1 signifies that the generated code manages dynamic memory more efficiently compared to the canonical solution, while a value greater than 1 indicates less efficient management of dynamic memory. This metric provides insight into the relative use of dynamic memory of generated code compared to an established benchmark.

### A.5   Model

We study both open- and closed-source LLMs in code generation. For open-source models, we evaluate[9] EFFIBENCH with CodeLlama-hf family (i.e., 7B, 13b, 34b, and 70B), CodeLlama-Instruct-hf family (i.e., 7B, 13b, 34b, and 70B), deepseek-coder-instruct (i.e., 1.3B and 6.7B) and base models (i.e., 6.7B and 33B), Phind-CodeLlama-34B (i.e., v1 and v2), starcoder, starcoderbase, and starcoder2 (i.e., 3B, 7B, and 15B), WizardCoder (i.e., 13B and 15B), XwinCoder (i.e., 13B and 34B), Yi models (34B, 34B-Chat, and 200K version), and five widely proposed SOTA models, i.e., Magicoder-6.7B, Mistral-7B, octocoder, Artigenz-6.7B, CodeFuse-33B, and codegemma-7b[10] since these open-source models have obtained SOTA pass@1 in the HumanEval and MBPP datasets. For closed-source models, we evaluated EFFIBENCH with GPT-3.5, GPT-4 [42], and claude-3, since we observe that these models obtain high pass@1 in code generation datasets (e.g., HumanEval [12], MBPP [7]). For GPT-3.5 models, we experiment with GPT-3.5-turbo-0301, GPT-3.5-turbo-0613, and GPT-3.5-turbo-1106 which represent three different versions of the GPT-3.5. For GPT-4 models, we experiment with GPT-4-turbo and GPT-4 (GPT-4-0613). For the claude-3 model, we evaluate the sonnet and haiku versions. For each LLM, we first collect the code that is correctly generated for each coding problem (i.e., they can pass all test cases provided by the dataset), then execute these correct code and calculate the efficiency metrics (See Section 3.4).

### A.6   Generalizability for other Benchmarks

Since one of our contributions is that we provide an efficiency evaluation framework, in this section we provide the generalizability of our framework on other benchmarks. Specifically, we evaluate

---

[9]The full evaluated model lists can be seen in our Hugging Face leaderboard.

[10]The model names are extracted from Hugging Face model card.

Table 7: Efficiency results for different algorithm subsets with closed-source LLMs.

| Model | max NET | NET | NET>5 | ET (s) | max NMU | NMU | NMU>5 | MU (Mb) | max NTMU | NTMU | NTMU>5 | TMU (Mb*s) | Pass@1 |
|---|---|---|---|---|---|---|---|---|---|---|---|---|---|
| GPT-3.5-turbo-0301 | | | | | | | | | | | | | |
| greedy | 3.63 | 3.02 | 0.0 | 0.35 | 2.03 | 1.92 | 0.0 | 59.42 | 7.51 | 6.14 | 90.7 | 16.75 | 39.9 |
| dynamic_programming | 27.70 | 3.64 | 4.5 | 0.46 | 2.05 | 1.93 | 0.0 | 55.25 | 70.62 | 7.73 | 89.3 | 21.44 | 40.4 |
| backtracking | 14.99 | 3.44 | 4.5 | 0.56 | 2.03 | 1.82 | 0.0 | 83.45 | 34.36 | 6.90 | 72.7 | 38.40 | 45.8 |
| divide_and_conquer | 3.53 | 3.00 | 0.0 | 0.34 | 2.02 | 1.89 | 0.0 | 53.42 | 7.00 | 5.96 | 87.5 | 11.41 | 38.1 |
| dfs | 3.47 | 2.91 | 0.0 | 0.35 | 2.05 | 1.81 | 0.0 | 59.62 | 6.82 | 5.68 | 85.2 | 13.60 | 25.0 |
| bfs | 6.35 | 3.17 | 4.2 | 0.41 | 2.05 | 1.90 | 0.0 | 55.10 | 13.56 | 6.43 | 91.7 | 15.99 | 27.9 |
| binary_search | 3.61 | 2.92 | 0.0 | 0.38 | 2.05 | 1.87 | 0.0 | 79.97 | 7.39 | 5.83 | 85.7 | 27.09 | 42.6 |
| two_pointers | 3.61 | 3.04 | 0.0 | 0.36 | 2.04 | 1.94 | 0.0 | 70.22 | 7.37 | 6.24 | 94.2 | 25.77 | 49.5 |
| sliding_window | 3.87 | 3.04 | 0.0 | 0.36 | 2.05 | 1.94 | 0.0 | 67.21 | 8.22 | 6.20 | 91.4 | 23.55 | 50.0 |
| bit_manipulation | 3.59 | 3.03 | 0.0 | 0.35 | 2.02 | 1.94 | 0.0 | 62.42 | 7.61 | 6.15 | 89.6 | 19.40 | 47.1 |
| sorting | 3.76 | 2.99 | 0.0 | 0.36 | 2.05 | 1.88 | 0.0 | 67.09 | 8.05 | 6.01 | 87.9 | 21.95 | 41.6 |
| GPT-4 | | | | | | | | | | | | | |
| greedy | 5.83 | 3.08 | 0.8 | 0.35 | 2.04 | 1.93 | 0.0 | 57.15 | 15.28 | 6.32 | 92.7 | 15.74 | 50.6 |
| dynamic_programming | 4.53 | 3.11 | 0.0 | 0.36 | 2.25 | 1.94 | 0.0 | 53.97 | 10.16 | 6.31 | 91.3 | 15.44 | 49.8 |
| backtracking | 4.53 | 3.01 | 0.0 | 0.44 | 2.03 | 1.84 | 0.0 | 81.67 | 10.16 | 5.89 | 77.3 | 32.23 | 45.8 |
| divide_and_conquer | 3.68 | 3.04 | 0.0 | 0.34 | 2.02 | 1.90 | 0.0 | 53.16 | 7.94 | 6.15 | 87.5 | 11.72 | 38.1 |
| dfs | 3.82 | 3.05 | 0.0 | 0.35 | 2.06 | 1.88 | 0.0 | 57.57 | 7.72 | 6.09 | 93.9 | 13.32 | 30.6 |
| bfs | 11.22 | 3.38 | 5.6 | 0.45 | 2.06 | 1.87 | 0.0 | 55.58 | 25.19 | 6.85 | 91.7 | 19.23 | 41.9 |
| binary_search | 3.69 | 2.96 | 0.0 | 0.38 | 2.04 | 1.88 | 0.0 | 75.09 | 7.78 | 5.92 | 89.3 | 25.24 | 50.7 |
| two_pointers | 3.94 | 3.09 | 0.0 | 0.36 | 2.04 | 1.94 | 0.0 | 66.90 | 8.90 | 6.36 | 95.2 | 23.65 | 59.0 |
| sliding_window | 8.46 | 3.23 | 2.5 | 0.39 | 2.06 | 1.92 | 0.0 | 66.36 | 17.85 | 6.60 | 95.0 | 25.41 | 57.1 |
| bit_manipulation | 4.53 | 3.12 | 0.0 | 0.36 | 2.03 | 1.95 | 0.0 | 60.22 | 10.16 | 6.39 | 92.6 | 18.60 | 52.9 |
| sorting | 13.89 | 3.11 | 1.5 | 0.38 | 2.25 | 1.89 | 0.0 | 63.62 | 43.92 | 6.40 | 90.0 | 21.09 | 54.6 |
| Claude-3-sonnet | | | | | | | | | | | | | |
| greedy | 3.75 | 3.13 | 0.0 | 0.36 | 2.03 | 1.93 | 0.0 | 58.47 | 7.90 | 6.39 | 90.3 | 16.68 | 42.4 |
| dynamic_programming | 16.34 | 3.42 | 1.8 | 0.47 | 2.04 | 1.94 | 0.0 | 54.95 | 37.83 | 6.96 | 92.0 | 35.81 | 40.8 |
| backtracking | 17.43 | 4.92 | 13.3 | 0.75 | 2.04 | 1.89 | 0.0 | 89.79 | 50.78 | 11.28 | 86.7 | 53.91 | 31.2 |
| divide_and_conquer | 3.56 | 3.03 | 0.0 | 0.36 | 2.02 | 1.88 | 0.0 | 53.44 | 7.18 | 6.01 | 75.0 | 12.62 | 57.1 |
| dfs | 3.61 | 3.03 | 0.0 | 0.36 | 2.05 | 1.81 | 0.0 | 59.20 | 7.53 | 5.94 | 86.2 | 13.98 | 26.9 |
| bfs | 6.24 | 3.08 | 3.4 | 0.42 | 2.05 | 1.84 | 0.0 | 59.57 | 13.17 | 6.06 | 86.2 | 16.36 | 33.7 |
| binary_search | 3.61 | 2.99 | 0.0 | 0.40 | 2.04 | 1.87 | 0.0 | 80.89 | 7.60 | 5.98 | 83.6 | 28.93 | 41.2 |
| two_pointers | 3.61 | 3.18 | 0.0 | 0.38 | 2.05 | 1.94 | 0.0 | 70.62 | 7.53 | 6.54 | 94.1 | 27.10 | 48.6 |
| sliding_window | 3.69 | 3.13 | 0.0 | 0.36 | 2.06 | 1.95 | 0.0 | 64.09 | 7.77 | 6.41 | 95.2 | 22.12 | 60.0 |
| bit_manipulation | 17.43 | 3.51 | 2.4 | 0.40 | 2.02 | 1.95 | 0.0 | 63.19 | 50.78 | 7.56 | 92.9 | 22.32 | 41.2 |
| sorting | 4.98 | 3.10 | 0.0 | 0.37 | 2.04 | 1.89 | 0.0 | 64.34 | 11.81 | 6.27 | 89.2 | 20.90 | 50.4 |

Table 8: Efficiency results of different models on HumanEvalPlus and MBPPPlus dataset.

| Model | HumanEvalPlus | | | | | | MBPPPlus | | | | | |
|---|---|---|---|---|---|---|---|---|---|---|---|---|
| | ET (s) | NET | MU (Mb) | NMU | TMU (Mb*s) | NTMU | ET (s) | NET | MU (Mb) | NMU | TMU (Mb*s) | NTMU |
| OpenCodeInterpreter-DS-1.3B | 0.20 | 0.86 | 57.24 | 1.00 | 6.63 | 0.84 | 0.28 | 0.94 | 59.01 | 1.01 | 11.73 | 0.98 |
| OpenCodeInterpreter-DS-6.7B | 0.21 | 0.98 | 58.83 | 1.06 | 6.79 | 0.99 | 0.26 | 1.06 | 58.39 | 1.00 | 9.25 | 1.08 |
| OpenCodeInterpreter-DS-33B | 0.21 | 0.95 | 59.90 | 1.05 | 7.05 | 0.94 | 0.44 | 1.59 | 58.72 | 1.00 | 20.19 | 1.86 |
| deepseek-coder-1.3b-instruct | 0.23 | 0.90 | 62.80 | 1.00 | 7.85 | 0.87 | 0.63 | 1.68 | 354.01 | 6.05 | 1463.46 | 89.12 |
| deepseek-coder-6.7b-instruct | 0.22 | 0.76 | 59.57 | 1.00 | 7.34 | 0.77 | 0.76 | 3.62 | 58.44 | 1.00 | 39.11 | 5.69 |
| deepseek-coder-33b-instruct | 0.21 | 0.95 | 63.52 | 0.99 | 7.18 | 0.95 | 0.58 | 2.33 | 53.48 | 0.91 | 28.74 | 3.16 |
| CodeLlama-7b-Instruct-hf | 0.20 | 0.71 | 57.39 | 0.91 | 7.08 | 0.70 | 0.45 | 2.04 | 56.96 | 0.97 | 13.26 | 1.79 |
| CodeLlama-13b-Instruct-hf | 0.23 | 0.95 | 58.13 | 0.96 | 7.97 | 0.94 | 0.53 | 2.11 | 55.37 | 0.95 | 21.75 | 2.34 |
| CodeLlama-34b-Instruct-hf | 0.24 | 0.95 | 61.79 | 1.01 | 8.45 | 0.96 | 0.42 | 1.18 | 69.80 | 1.19 | 84.01 | 5.47 |
| CodeLlama-70b-Instruct-hf | 0.21 | 0.93 | 60.19 | 1.01 | 6.76 | 1.01 | 0.23 | 1.06 | 58.13 | 0.98 | 7.65 | 1.05 |
| XwinCoder-13B | 0.27 | 1.08 | 61.14 | 1.04 | 9.25 | 1.09 | 0.50 | 1.96 | 58.38 | 1.00 | 23.88 | 2.50 |
| XwinCoder-34B | 0.25 | 1.07 | 60.75 | 1.05 | 8.46 | 1.08 | 0.38 | 1.44 | 58.27 | 1.00 | 14.77 | 1.48 |
| WizardCoder-7B | 0.21 | 0.91 | 58.59 | 1.01 | 6.63 | 0.89 | 0.22 | 1.05 | 58.44 | 0.99 | 7.19 | 1.03 |
| WizardCoder-13B | 0.21 | 0.81 | 60.59 | 1.00 | 7.22 | 0.79 | 0.62 | 1.35 | 57.74 | 0.99 | 30.66 | 1.43 |
| WizardCoder-34B | 0.22 | 0.79 | 58.13 | 1.00 | 7.10 | 0.78 | 0.68 | 2.43 | 56.75 | 0.97 | 34.06 | 3.14 |
| starcoder2-3b | 0.24 | 1.02 | 62.45 | 1.00 | 7.73 | 0.89 | 0.17 | 0.83 | 45.82 | 0.79 | 5.10 | 0.77 |
| starcoder2-7b | 0.21 | 0.89 | 62.53 | 1.00 | 7.41 | 0.85 | 1.72 | 8.63 | 25.61 | 0.44 | 40.42 | 6.22 |

the efficiency of LLM-generated code on HumanEvalPlus and MBPPPlus[11] [32]. The evaluation results are demonstrated on Table 10. We can observe that EFFIBENCH's framework can integrate with other benchmarks and then be used to evaluate the efficiency of LLM-generated code. Besides, we can also observe that the efficiency of LLM-generated code in this benchmark is close to the canonical solutions and sometimes even better than the canonical solutions. For example, the NET of OpenCodeInterpreter-DS-1.3B is 0.86, which is even lower than the canonical solutions. We can also observe that this behavior also exists in the MBPPPlus, while different from these benchmarks, we can observe that most of the code generated by LLMs is less efficient than the canonical solutions in EFFIBENCH.

## A.7 Efficiency metrics distribution

As shown in Table 3, we report the ratio of correct code with 5x efficiency metrics (i.e., NET, NMU, and NTMU) in total correct code generated by LLMs. In this section, we further analyze

---

[11]HumanEval and MBPP datasets have a limited number of test cases (fewer than 10) for each task, which can lead to highly random efficiency testing results due to the rapid execution of the code. To mitigate the impact of randomness, we utilize the test cases provided by EvalPlus to ensure sufficient testing time.

<table>
<tr><td>

**GPT-3.5-Turbo-0301**

```python
class Solution:
    def kInversePairs(self, n: int, k: int) ->
    ↪    int:
        MOD = 10**9 + 7
        # Initialization of a 2D matrix with
        ↪    (n+1)x(k+1) dimensions
        # Memory-intensive: Utilizes a matrix
        ↪    for storing all subproblem results
        dp = [[0 for _ in range(k+1)] for _ in
        ↪    range(n+1)]
        for i in range(n+1):
            dp[i][0] = 1  # Base case: one way
            ↪    to have zero inverse pairs
        for i in range(1, n+1):
            for j in range(1, k+1):
                # Dynamic programming state
                ↪    transition
                dp[i][j] = (dp[i-1][j] +
                ↪    dp[i][j-1]) % MOD
                if j-i >= 0:
                    # Adjustment to avoid
                    ↪    overcounting,
                    ↪    demonstrates the
                    ↪    complexity of state
                    ↪    management
                    dp[i][j] = (dp[i][j] -
                    ↪    dp[i-1][j-i] + MOD) %
                    ↪    MOD
        return dp[n][k] % MOD
```

</td><td>

**Canonica Solution**

```python
class Solution:
    def kInversePairs(self, n: int, k: int) ->
    ↪    int:
        mod = 10**9 + 7
        # f array represents current count of
        ↪    inverse pairs at index k
        # Space optimization: Only one array of
        ↪    size k+1 is used
        f = [1] + [0] * k
        # s is a prefix sum array to optimize
        ↪    the range sum calculation
        # Efficient rolling sum reduces space
        ↪    complexity from O(n*k) to O(k)
        s = [0] * (k + 2)
        for i in range(1, n + 1):
            for j in range(1, k + 1):
                # Utilizing prefix sum to
                ↪    calculate range sums
                ↪    efficiently
                f[j] = (s[j + 1] - s[max(0, j -
                ↪    (i - 1))]) % mod
            for j in range(1, k + 2):
                # Update prefix sums after each
                ↪    iteration
                s[j] = (s[j - 1] + f[j - 1]) %
                ↪    mod
        return f[k]
```

</td></tr>
</table>

Figure 2: A case illustration of GPT-3.5-turbo-0301 and *canonica_solution*. GPT-3.5-turbo-0301 generated code requires 70.62x memory usage compared with *canonical_solution*. GPT-3.5-turbo-0301 generated code employs a 2-dimensional matrix to manage state transitions, leading to substantial memory overhead, particularly evident when the parameters $n$ and $k$ are large. In contrast, the *canonical_solution* optimizes memory usage by utilizing a rolling sum technique and a single-dimensional dynamic array, significantly reducing the space complexity from $O(n \times k)$ to $O(k)$.

Table 9: Efficiency results of 7 different LLMs generated code. In this table, we focus on three normalized metrics (i.e., NET, NMU, and NTMU). For each metric, we consider four different scenarios. For example, For NET, we report the min NET, the ratio of NET<1 in corrected code, the ratio of NET>=1 in corrected code, and max NET values.

| Model | min NET | NET <1 | NET >1 | max NET | min NMU | NMU <1 | NMU >1 | max NMU | min NTMU | NTMU <1 | NTMU >1 | max NTMU |
|---|---|---|---|---|---|---|---|---|---|---|---|---|
| gpt-3.5-turbo-0301 | 1.09 | 0.00 | 100.00 | 27.70 | 0.82 | 2.13 | 97.9 | 2.1 | 0.98 | 0.47 | 99.5 | 47.0 |
| gpt-3.5-turbo-0613 | 1.10 | 0.00 | 100.00 | 46.70 | 0.82 | 1.72 | 98.3 | 2.6 | 0.99 | 0.22 | 99.8 | 68.9 |
| gpt-3.5-turbo-1106 | 1.11 | 0.00 | 100.00 | 68.71 | 0.82 | 1.83 | 98.2 | 9.1 | 1.01 | 0.20 | 99.8 | 68.8 |
| gpt-4 | 1.10 | 0.00 | 100.00 | 13.89 | 0.82 | 1.57 | 98.4 | 2.2 | 1.01 | 0.00 | 100.0 | 15.3 |
| gpt-4-turbo-preview | 0.90 | 0.15 | 99.85 | 27.00 | 0.82 | 1.38 | 98.6 | 9.1 | 0.66 | 0.46 | 99.5 | 68.5 |
| claude-3-haiku | 0.94 | 0.23 | 99.77 | 28.75 | 0.82 | 1.86 | 98.1 | 2.1 | 0.68 | 0.23 | 99.8 | 72.9 |
| claude-3-sonnet | 0.98 | 0.23 | 99.77 | 17.43 | 0.50 | 1.62 | 98.4 | 2.1 | 0.94 | 0.46 | 99.5 | 24.0 |

the distribution of normalized efficiency metrics, i.e., whether there are cases where LLMs yield more efficient code than the canonical solutions. The evaluation results are demonstrated in Table 11, where we evaluated 7 LLMs based on following the setup of Table 4. We can observe that for all evaluated LLMs, there are only a small of code generated by LLMs in Table 11 are more efficient than the canonical solutions, while most of the code is less efficient. For example, we can observe that only 0.23% code in Claude-3-sonnet generated correct code is more efficient than the canonical solution, while 99.77% code's NET is large or equal to the canonical solution generated code. We suspect that the overall inefficiency of the code produced by LLMs when compared to canonical solutions may be attributed to the distribution of the training data. Typically, these datasets prioritize the correctness of code and collect code from repositories like GitHub where code is often correct but not necessarily optimized for efficiency. Focusing primarily on correctness without adequate attention to efficiency could result in neglecting efficiency in the code generated by LLMs.

Figure 3: Example problems synthesized (few-shot) by GPT-4-0613. The prompt is shown in purple, and the model response is shown in blue. The prompt also typically contains several few-shot examples in the same format, which are not shown here.

### A.8  Case illustration for worst case

As shown in Table 7, we can observe that most of the three most inefficient pieces of code are implemented by DP, backtracking, and BFS. In this section, we provide the comparison of GPT-3.5-turbo-0301 generated code and canonical solution to analyze why LLM-generated code is inefficient[12].

**BFS** We provide the worst-case illustration for BFS in Figure 4. We can observe that the code completed by GPT-3.5-turbo-0301 is less efficient in terms of memory usage compared to our *canonical_solution*. Specifically, GPT-3.5-turbo-0301's code employs a standard BFS with a list-based queue, alongside a set for tracking visited states and deadends. The space complexity for this solution includes O(N) for deadends and visited states, and potentially $O(10^4)$ for the queue, as it may store all possible lock combinations in the worst-case scenario. The breadth of the search linearly expands with the number of steps, as each step introduces multiple neighbors into the queue. Conversely, *canonical_solution* adopts a more sophisticated approach with a two-way BFS, utilizing two dictionaries for tracking the search from both ends and two deques for managing the queues. The space complexity remains O(N) for deadends, similar to Solution 1, but each dictionary and deque can grow up to $O(10^4)$ in the worst-case scenario. However, the two-way BFS approach

---

[12]We demonstrate DP example in Figure 2.

**GPT-3.5-Turbo-0301**

```python
class Solution:
    def openLock(self, deadends: List[str],
      target: str) -> int:
        # Convert deadends into a set for
          O(1) lookup times
        deadends = set(deadends)
        head = '0000'
        # Immediate check to avoid
          unnecessary processing
        if head in deadends:
            return -1

        # Helper function to generate all
          possible next states from a
          given state
        def _gen_neighbors(num):
            for i in range(4):
                x = int(num[i])
                # Generate neighbors by
                  incrementing or
                  decrementing each wheel
                  digit
                for d in [-1, 1]:
                    y = (x + d) % 10
                    yield num[:i] + str(y)
                      + num[i+1:]

        # Initialize BFS with the starting
          point
        stack = [head]
        visited = set(stack)  # Track
          visited states to prevent
          re-processing
        steps = 0
        while len(stack) > 0:
            size = len(stack)
            for i in range(size):
                # Inefficient pop operation
                  due to list usage
                node = stack.pop(0)
                # Check if the target has
                  been reached
                if node == target:
                    return steps
                # Explore all neighboring
                  states
                for neighbor in
                  _gen_neighbors(node):
                    if neighbor in deadends
                      or neighbor in
                      visited:
                        continue
                    # Add new state to
                      visited and queue
                      for further
                      exploration
                    visited.add(neighbor)
                    stack.append(neighbor)
            # Increment the number of steps
              after processing each
              level
            steps += 1
        return -1  # If no solution is
          found, return -1
```

**Canonica Solution**

```python
class Solution:
    def openLock(self, deadends: List[str], target:
      str) -> int:
        # Function to generate all possible next
          states for a given state
        def next(s):
            res = []
            s = list(s)
            for i in range(4):
                c = s[i]
                # Decrement the wheel value
                s[i] = '9' if c == '0' else
                  str(int(c) - 1)
                res.append(''.join(s))
                # Increment the wheel value
                s[i] = '0' if c == '9' else
                  str(int(c) + 1)
                res.append(''.join(s))
                # Restore original wheel value
                s[i] = c
            return res

        # Function to expand the search frontier in
          one direction
        def extend(m1, m2, q):
            for _ in range(len(q)):
                p = q.popleft()  # Efficient pop from
                  deque
                step = m1[p]
                for t in next(p):
                    if t in s or t in m1:
                        continue
                    # Check if paths meet; if so,
                      return the combined steps
                    if t in m2:
                        return step + 1 + m2[t]  #
                          Early termination when
                          paths intersect
                    # Record steps to reach new state
                      and add to the queue
                    m1[t] = step + 1
                    q.append(t)
            return -1

        # Main function to perform bidirectional BFS
        def bfs():
            # Initial setups for BFS: maps and queues
              for both directions
            m1, m2 = {"0000": 0}, {target: 0}
            q1, q2 = deque(['0000']), deque([target])
            while q1 and q2:
                # Alternate between expanding the
                  front from start and target
                t = extend(m1, m2, q1) if len(q1) <=
                  len(q2) else extend(m2, m1, q2)
                if t != -1:
                    return t  # Return the total
                      steps if a meeting point is
                      found
            return -1

        if target == '0000':
            return 0
        s = set(deadends)
        if '0000' in s:
            return -1
        return bfs()  # Start the bidirectional BFS
          process
```

Figure 4: A case illustration of GPT-3.5-turbo-0301 and *canonical_solution*. The left code is completed by GPT-3.5-turbo-0301, which requires 50.1 MB*seconds, while the right result is our *canonical_solution*, which requires 7.5 MB*seconds. The key advantage of the *canonical_solution* is its use of bidirectional BFS, which significantly speeds up the search space reduction, resulting in a more efficient computation.



**GPT-3.5-Turbo-0301**

```python
class Solution:
    def restoreIpAddresses(self, s: str) ->
    ↪  List[str]:
        result = []
        n = len(s)

        # Generate all possible segment splits
        ↪  through iterative nested loops
        for i in range(1, min(4, n - 2) + 1):
            for j in range(i + 1, min(i + 4, n -
            ↪  1) + 1):
                for k in range(j + 1, min(j + 4,
                ↪  n) + 1):
                    s1 = s[:i]
                    s2 = s[i:j]
                    s3 = s[j:k]
                    s4 = s[k:]
                    # Delayed validation results
                    ↪  in more recursive stack
                    ↪  consumption
                    if self.isValid(s1) and
                    ↪  self.isValid(s2) and
                    ↪  self.isValid(s3) and
                    ↪  self.isValid(s4):
                        result.append(s1 + "." +
                        ↪  s2 + "." + s3 + "."
                        ↪  + s4)

        return result

    def isValid(self, s: str) -> bool:
        # Perform checks after generating all
        ↪  combinations, less efficient in
        ↪  pruning
        if len(s) == 0 or len(s) > 3 or (s[0] ==
        ↪  '0' and len(s) > 1) or int(s) > 255:
            return False
        return True
```





**Canonica Solution**

```python
class Solution:
    def restoreIpAddresses(self, s: str) ->
    ↪  List[str]:
        def check(i: int, j: int) -> int:
            # Validate the segment early;
            ↪  disallow leading zeros unless
            ↪  the segment is '0'
            if s[i] == "0" and i != j:
                return False
            return 0 <= int(s[i : j + 1]) <= 255

        def dfs(i: int):
            # Check for successful completion:
            ↪  correct path found
            if i >= n and len(t) == 4:
                ans.append(".".join(t))
                return
            # Early termination to prevent
            ↪  unnecessary recursion
            if i >= n or len(t) >= 4:
                return
            # Dynamically manage segment
            ↪  additions and pruning
            for j in range(i, min(i + 3, n)):
                if check(i, j):
                    t.append(s[i : j + 1])
                    dfs(j + 1)
                    t.pop()  # Efficient
                    ↪  backtracking by removing
                    ↪  last segment

        n = len(s)
        ans = []
        t = []  # Temporary list to manage IP
        ↪  segments
        dfs(0)
        return ans
```



Figure 5: A side-by-side case illustration of GPT-3.5-turbo-0301 and *canonical_solution* in backtracking implementations. The left code by GPT-3.5-turbo-0301 employs a less efficient recursive method, leading to high memory usage by exhaustively checking every possible segment combination. In contrast, the *canonical_solution* on the right optimizes memory usage through effective backtracking that prunes invalid paths early and dynamically manages segments with a list $t$, significantly reducing memory overhead. This results in the GPT-3.5-turbo-0301 code requiring 34.36 times more memory during execution compared to the *canonical_solution*.

significantly condenses the search breadth by converging from both ends, reducing the overall memory consumption.

**Backtracking** We provide the worst-case illustration for Backtracking in Figure 6. We can observe that GPT-3.5-turbo-0301 implementation requires substantially higher memory usage due to its less optimized recursive exploration strategy. This version systematically checks every possible combination of segments that could form an IP address by recursively calling the validation and appending results for each possible segment split. This approach accumulates a significant memory overhead as every recursive call consumes stack space and each path's state is saved until the recursion unwinds. Conversely, the canonical solution leverages a more refined backtracking mechanism that strategically prunes invalid paths earlier through its *check* function and reduces unnecessary recursive depth by verifying conditions upfront. Additionally, the canonical method uses a dynamic list $t$ to store temporary segments, effectively managing memory by adding and removing segments as needed without redundantly holding onto unsuccessful paths, leading to a drastically reduced memory footprint during execution. This optimization in the canonical solution translates into a significant performance improvement. Specifically, GPT-3.5-turbo-0301 generated code has 34.36x memory usage during the code execution compared with *canonical_solution*.

Table 10: Efficiency results of different models on HumanEvalPlus and MBPPPlus dataset.

| Model | HumanEvalPlus | | | | | | MBPPPlus | | | | | |
|---|---|---|---|---|---|---|---|---|---|---|---|---|
| | ET (s) | NET | MU (Mb) | NMU | TMU (Mb*s) | NTMU | ET (s) | NET | MU (Mb) | NMU | TMU (Mb*s) | NTMU |
| OpenCodeInterpreter-DS-1.3B | 0.20 | 0.86 | 57.24 | 1.00 | 6.63 | 0.84 | 0.28 | 0.94 | 59.01 | 1.01 | 11.73 | 0.98 |
| OpenCodeInterpreter-DS-6.7B | 0.21 | 0.98 | 58.83 | 1.06 | 6.79 | 0.99 | 0.26 | 1.06 | 58.39 | 1.00 | 9.25 | 1.08 |
| OpenCodeInterpreter-DS-33B | 0.21 | 0.95 | 59.90 | 1.05 | 7.05 | 0.94 | 0.44 | 1.59 | 58.72 | 1.00 | 20.19 | 1.86 |
| deepseek-coder-1.3b-instruct | 0.23 | 0.90 | 62.80 | 1.00 | 7.85 | 0.87 | 0.63 | 1.68 | 354.01 | 6.05 | 1463.46 | 89.12 |
| deepseek-coder-6.7b-instruct | 0.22 | 0.76 | 59.57 | 1.00 | 7.34 | 0.77 | 0.76 | 3.62 | 58.44 | 1.00 | 39.11 | 5.69 |
| deepseek-coder-33b-instruct | 0.21 | 0.95 | 63.52 | 0.99 | 7.18 | 0.95 | 0.58 | 2.33 | 53.48 | 0.91 | 28.74 | 3.16 |
| CodeLlama-7b-Instruct-hf | 0.20 | 0.71 | 57.39 | 0.91 | 7.08 | 0.70 | 0.45 | 2.04 | 56.96 | 0.97 | 13.26 | 1.79 |
| CodeLlama-13b-Instruct-hf | 0.23 | 0.95 | 58.13 | 0.96 | 7.97 | 0.94 | 0.53 | 2.11 | 55.37 | 0.95 | 21.75 | 2.34 |
| CodeLlama-34b-Instruct-hf | 0.24 | 0.95 | 61.79 | 1.01 | 8.45 | 0.96 | 0.42 | 1.18 | 69.80 | 1.19 | 84.01 | 5.47 |
| CodeLlama-70b-Instruct-hf | 0.21 | 0.93 | 60.19 | 1.01 | 6.76 | 1.01 | 0.23 | 1.06 | 58.13 | 0.98 | 7.65 | 1.05 |
| XwinCoder-13B | 0.27 | 1.08 | 61.14 | 1.04 | 9.25 | 1.09 | 0.50 | 1.96 | 58.38 | 1.00 | 23.88 | 2.50 |
| XwinCoder-34B | 0.25 | 1.07 | 60.75 | 1.05 | 8.46 | 1.08 | 0.38 | 1.44 | 58.27 | 1.00 | 14.77 | 1.48 |
| WizardCoder-7B | 0.21 | 0.91 | 58.59 | 1.01 | 6.63 | 0.89 | 0.22 | 1.05 | 58.44 | 0.99 | 7.19 | 1.03 |
| WizardCoder-13B | 0.21 | 0.81 | 60.59 | 1.00 | 7.22 | 0.79 | 0.62 | 1.35 | 57.74 | 0.99 | 30.66 | 1.43 |
| WizardCoder-34B | 0.22 | 0.79 | 58.13 | 1.00 | 7.10 | 0.78 | 0.68 | 2.43 | 56.75 | 0.97 | 34.06 | 3.14 |
| starcoder2-3b | 0.24 | 1.02 | 62.45 | 1.00 | 7.73 | 0.89 | 0.17 | 0.83 | 45.82 | 0.79 | 5.10 | 0.77 |
| starcoder2-7b | 0.21 | 0.89 | 62.53 | 1.00 | 7.41 | 0.85 | 1.72 | 8.63 | 25.61 | 0.44 | 40.42 | 6.22 |

## A.9 Generalizability for other Benchmarks

Since one of our contributions is that we provide an efficiency evaluation framework, which raises one question about whether we can use the framework of EFFIBENCH to measure the efficiency of LLM-generated code for other benchmarks. In this section, we provide the generalizability of our framework on other benchmarks. Specifically, we evaluate the efficiency of LLM-generated code on HumanEval+ and MBPPP+[13] [32]. The evaluation results are demonstrated on Table 10. The evaluation results demonstrate that EFFIBENCH's framework can integrate with other benchmarks and then be used to evaluate the efficiency of LLM-generated code. In addition, our results also demonstrate that the efficiency of LLM-generated code in these two datasets is close to the canonical solutions and sometimes even better than the canonical solutions. For example, the NET of OpenCodeInterpreter-DS-1.3B is 0.86 in the HumanEval+ dataset, which is even lower than the canonical solutions.

## A.10 Efficiency metrics distribution

As demonstrated in Table 3, the efficiency of LLM-generated code are lower than the efficiency of the dataset provided canonical solution. To measure the ratio of the inefficient code generated by LLMs in the total LLM-generated code, we provide the ratio of the code higher / lower than the efficiency of the canonical solution provided by the dataset. The evaluation results are demonstrated in Table 11, where we evaluated 7 LLMs based on following the setup of Table 4. The evaluation results demonstrate that for all evaluated LLMs, there are only a small of code generated by LLMs in Table 11 are more efficient than the canonical solutions, while most of the code is less efficient. For example, only 0.23% code in Claude-3-sonnet generated correct code is more efficient than the canonical solution, while 99.77% code's NET is large or equal to the canonical solution generated code. We suspect that the overall inefficiency of the code produced by LLMs when compared to canonical solutions may be attributed to the distribution of the training data. Typically, these datasets prioritize the correctness of code and collect code from repositories like GitHub where code is often correct but not necessarily optimized for efficiency. Focusing primarily on correctness without adequate attention to efficiency could result in neglecting efficiency in the code generated by LLMs.

## A.11 Case study for efficient solution

## A.12 Calculating the normalized metrics with task level

In Section 3.4, we define the normalized efficiency metrics at the dataset level. For example, NET is defined as:

$$NET = \frac{1}{N} \sum^{N} \frac{T_{\text{code}}}{T_{\text{canonical}}}$$

---

[13]HumanEval and MBPP datasets have a limited number of test cases (fewer than 10) for each task, which can lead to highly random efficiency testing results due to the rapid execution of the code. To mitigate the impact of randomness, we utilize the test cases provided by EvalPlus to ensure sufficient testing time.

Table 11: Efficiency results of 7 different LLMs generated code. In this table, we focus on three normalized metrics (i.e., NET, NMU, and NTMU). For each metric, we consider four different scenarios. For example, For NET, we report the min NET, the ratio of NET<1 in corrected code, the ratio of NET>=1 in corrected code, and max NET values.

| Model | min NET | NET <1 | NET >1 | max NET | min NMU | NMU <1 | NMU >1 | max NMU | min NTMU | NTMU <1 | NTMU >1 | max NTMU |
|---|---|---|---|---|---|---|---|---|---|---|---|---|
| gpt-3.5-turbo-0301 | 1.09 | 0.00 | 100.00 | 27.70 | 0.82 | 2.13 | 97.9 | 2.1 | 0.98 | 0.47 | 99.5 | 47.0 |
| gpt-3.5-turbo-0613 | 1.10 | 0.00 | 100.00 | 46.70 | 0.82 | 1.72 | 98.3 | 2.6 | 0.99 | 0.22 | 99.8 | 68.9 |
| gpt-3.5-turbo-1106 | 1.11 | 0.00 | 100.00 | 68.71 | 0.82 | 1.83 | 98.2 | 9.1 | 1.01 | 0.20 | 99.8 | 68.8 |
| gpt-4 | 1.10 | 0.00 | 100.00 | 13.89 | 0.82 | 1.57 | 98.4 | 2.2 | 1.01 | 0.00 | 100.0 | 15.3 |
| gpt-4-turbo-preview | 0.90 | 0.15 | 99.85 | 27.00 | 0.82 | 1.38 | 98.6 | 9.1 | 0.66 | 0.46 | 99.5 | 68.5 |
| claude-3-haiku | 0.94 | 0.23 | 99.77 | 28.75 | 0.82 | 1.86 | 98.1 | 2.1 | 0.68 | 0.23 | 99.8 | 72.9 |
| claude-3-sonnet | 0.98 | 0.23 | 99.77 | 17.43 | 0.50 | 1.62 | 98.4 | 2.1 | 0.94 | 0.46 | 99.5 | 24.0 |

**Claude-3-Sonnet**

```python
class Solution:
    def removeStones(self, stones:
↪   List[List[int]]) -> int:
        def find(x):
            if x != root[x]:
                root[x] = find(root[x])
            return root[x]

        def union(x, y):
            rootX = find(x)
            rootY = find(y)
            if rootX != rootY:
                root[rootX] = rootY

        root = {}
        for i, (x, y) in enumerate(stones):
            root[i] = i

        # Union stones sharing the same row or
↪       column
        for i in range(len(stones)):
            for j in range(i+1, len(stones)):
                if stones[i][0] == stones[j][0]
↪               or stones[i][1] ==
↪               stones[j][1]:
                    union(i, j)

        # Count the number of distinct islands
        islands = set()
        for i in range(len(stones)):
            islands.add(find(i))

        return len(stones) - len(islands)
```

**Canonica Solution**

```python
class Solution:
    def removeStones(self, stones:
↪   List[List[int]]) -> int:
        def find(x):
            if p[x] != x:
                p[x] = find(p[x])
            return p[x]

        n = 10010
        p = list(range(n << 1))
        for x, y in stones:
            p[find(x)] = find(y + n)

        s = {find(x) for x, _ in stones}
        return len(stones) - len(s)
```

Figure 6: Case example for Claude-3-sonnet generated code which is more efficient than the canonical solution for MU.

In this section, we further discuss the normalized efficiency metrics for LLM-generated code at the dataset level. For example, we set NET* as the dataset-level normalized execution time metric. The NET* is defined as: where $T_{code}$ is the execution time of the generated code, and $T_{canonical}$ is the execution time of the canonical solution.

$$NET = \frac{\sum^{N} T_{code}}{\sum^{N} T_{canonical}}$$

We follow the setup of Table 4 to evaluate the efficiency of LLM-generated code in 9 open- and closed-source models. The evaluation results are demonstrated in Table 23. We can observe that with the dataset-level normalized metric calculation, the efficiency of LLM-generated code is closer to the canonical solution. For example, GPT-3.5-turbo-0301 generated code required execution time decreases from 3.18x to 2.92x compared to the canonical solution. The key reason is that the dataset-level normalization aggregates the performance across all tasks, potentially masking significant variations in efficiency on individual tasks. While the dataset-level normalized metric, such as NET*, provides a broad overview of the model's performance, it can obscure important details about how well the model handles specific tasks. For example, this dataset-level calculation ignores

Table 12: Evaluation results of different LLMs efficiency results for EFFIBENCH. We use * to represent the results with the new calculation type.

| Model | ET | NET | NET* | MU | NMU | NMU* | TMU | NTMU | NTMU* |
|-------|-----|-----|------|-----|-----|------|-----|------|-------|
| gpt-3.5-turbo-0301 | 0.39 | 3.18 | 2.92 | 60.53 | 1.91 | 1.61 | 19.06 | 6.50 | 2.52 |
| gpt-3.5-turbo-0613 | 0.39 | 3.22 | 2.96 | 59.82 | 1.92 | 1.64 | 19.11 | 6.71 | 2.68 |
| gpt-3.5-turbo-1106 | 0.40 | 3.40 | 3.15 | 59.34 | 1.94 | 1.66 | 19.39 | 7.24 | 2.85 |
| gpt-4 | 0.37 | 3.12 | 2.88 | 58.85 | 1.92 | 1.66 | 17.69 | 6.36 | 2.69 |
| gpt-4-turbo-preview | 0.38 | 3.19 | 3.02 | 57.06 | 1.93 | 1.71 | 16.92 | 6.57 | 3.02 |
| claude-3-haiku | 0.39 | 3.28 | 3.00 | 59.15 | 1.91 | 1.64 | 17.99 | 6.71 | 2.66 |
| claude-3-sonnet | 0.40 | 3.22 | 3.05 | 60.22 | 1.91 | 1.62 | 23.29 | 6.57 | 3.13 |

Table 13: Evaluation result of GPT-3.5-turbo-0301 with the different number of tests for EFFIBENCH. "10" means the evaluation results are obtained with 10 tests.

| number of tests | max NET | NET | NET>5 | ET (s) | max NMU | NMU | NMU>5 | MU (Mb) | max NTMU | NTMU | NTMU>5 | TMU (Mb*s) |
|-----------------|---------|-----|-------|--------|---------|-----|-------|---------|----------|------|--------|-----------|
| 10 | 4.13 | 2.36 | 0.0 | 0.27 | 2.01 | 1.83 | 0.0 | 49.00 | 8.84 | 4.75 | 41.9 | 8.84 |
| 100 | 27.70 | 3.18 | 1.4 | 0.39 | 2.05 | 1.91 | 0.0 | 60.53 | 70.62 | 6.50 | 89.1 | 19.06 |
| 1000 | 66.68 | 3.95 | 4.6 | 0.56 | 11.91 | 2.84 | 5.0 | 162.11 | 436.11 | 10.08 | 66.6 | 340.51 |

the metrics evaluated in Table 11. This aggregation can lead to a situation where poor performance on a few tasks is averaged out by better performance on others, giving a potentially misleading impression of overall efficiency.

## A.13 Efficiency distribution for the normalized metrics

As shown in Table 11, we report the efficiency distribution for normalized metrics of the LLM-generated code. In this section, we further break down the efficiency distribution of GPT-3.5-turbo-0301 generated code. Specifically, for each normalized metric, we collect all GPT-3.5-turbo-0301 generated code's efficiency metric. Then we divide them into 100 buckets. Then, we report the accumulated figures in Figure 8. We can observe that most of the GPT-3.5-turbo-0301 generated code is less efficient than the canonical solution (i.e., value = 1).

## A.14 Efficiency of Code with different number of tests

Our experiments in Table 3 only consider 100 tests for each problem, which inspires us to consider how different numbers of tests affect the efficiency of code generated by code generation models. To answer this question, we investigate how does different number of tests affects the efficiency score for each metric. The evaluation results are shown in Table 24, where we can observe that once we increase the tests from 10 to 1,000, the efficiency score for NET, NMU, and NTMU increase for GPT-3.5-turbo-0301. For example, the GPT-3.5-turbo-0301's NTMU increases from 4.75 to 10.08. We indicate that the key reason is once we increase the number of tests, more edge cases would be covered (e.g., more length, data distribution). However, since the tests for the efficiency experiments, the overhead such as memory usage increases largely. For example, when we increase the tests from 100 to 1,000, the TMU increases from 8.84 MB*s to 340.51 MB*s, which requires more computation resources for experiments. So in our experiments and Leaderboard, we focus on studying the LLM-generated code efficiency in 100 tests.

## A.15 Randomness

**Seed**    We also evaluated the efficiency of the code generated by GPT-3.5-turbo-0301 five times in the same environments to ensure the reliability of our results. As demonstrated in Table 25, performance metrics such as ET, MU, and TMU show remarkable consistency across different executions. Specifically, the standard deviations (std) for these metrics are exceptionally low, demonstrating minimal variability and highlighting the stability of the code execution in our testing environment. For example, the mean of the ET is 0.39 (s), while the std of the ET is 0 for the five times results. This consistent performance underpins the robustness of our experimental approach, providing a solid foundation for further analysis of the model's operational characteristics.

**Environment**    We also provide an analysis of the efficiency of the code generated by closed-source models in different local environments. The results are shown in Table 26, where we can observe that

Table 14: Evaluation result of GPT-3.5-turbo-0301 with five different executions. The mean and standard deviation (std) values are reported to two decimal places.

| number of tests | max NET | NET | NET>5 | ET (s) | max NMU | NMU | NMU>5 | MU (Mb) | max NTMU | NTMU | NTMU>5 | TMU (Mb*s) |
|---|---|---|---|---|---|---|---|---|---|---|---|---|
| 0 | 27.70 | 3.18 | 1.4 | 0.39 | 2.05 | 1.91 | 0.0 | 60.53 | 70.62 | 6.50 | 89.1 | 19.06 |
| 1 | 27.70 | 3.17 | 1.4 | 0.39 | 2.06 | 1.91 | 0.0 | 60.55 | 70.50 | 6.48 | 89.1 | 19.07 |
| 2 | 27.76 | 3.17 | 1.4 | 0.38 | 2.06 | 1.91 | 0.0 | 60.55 | 70.41 | 6.52 | 89.1 | 19.21 |
| 3 | 27.42 | 3.18 | 1.4 | 0.39 | 2.05 | 1.91 | 0.0 | 60.54 | 70.70 | 6.70 | 89.2 | 18.95 |
| 4 | 27.78 | 3.18 | 1.4 | 0.39 | 2.05 | 1.91 | 0.0 | 60.53 | 70.48 | 6.41 | 89.1 | 19.05 |
| Mean | 27.67 | 3.18 | 1.4 | 0.39 | 2.05 | 1.91 | 0.0 | 60.54 | 70.54 | 6.52 | 89.1 | 19.07 |
| Std | 0.13 | 0.00 | 0.0 | 0.00 | 0.00 | 0.00 | 0.0 | 0.01 | 0.10 | 0.10 | 0.0 | 0.09 |

Table 15: Evaluation result of closed-source models for different environments. Both the canonical solution and LLM-generated code were executed in the same environments.

| number of tests | max NET | NET | NET>5 | ET (s) | max NMU | NMU | NMU>5 | MU (Mb) | max NTMU | NTMU | NTMU>5 | TMU (Mb*s) |
|---|---|---|---|---|---|---|---|---|---|---|---|---|
| 8336C CPU\|Python 3.11.2 | | | | | | | | | | | | |
| gpt-3.5-turbo-0301 | 27.70 | 3.18 | 1.4 | 0.39 | 2.05 | 1.91 | 0.0 | 60.53 | 70.62 | 6.50 | 89.1 | 19.06 |
| gpt-3.5-turbo-0613 | 46.70 | 3.22 | 0.9 | 0.39 | 2.64 | 1.92 | 0.0 | 59.82 | 161.12 | 6.71 | 89.9 | 19.11 |
| gpt-3.5-turbo-1106 | 68.71 | 3.40 | 1.6 | 0.40 | 9.12 | 1.94 | 0.2 | 59.34 | 182.63 | 7.24 | 90.9 | 19.39 |
| gpt-4 | 13.89 | 3.12 | 1.0 | 0.37 | 2.25 | 1.92 | 0.0 | 58.85 | 43.92 | 6.36 | 91.1 | 17.69 |
| gpt-4-turbo-preview | 27.00 | 3.19 | 1.2 | 0.38 | 9.13 | 1.93 | 0.2 | 57.06 | 68.48 | 6.57 | 91.1 | 16.92 |
| claude-3-haiku | 28.75 | 3.28 | 0.7 | 0.39 | 2.05 | 1.91 | 0.0 | 59.15 | 72.87 | 6.71 | 90.0 | 17.99 |
| claude-3-sonnet | 17.43 | 3.22 | 0.9 | 0.40 | 2.06 | 1.91 | 0.0 | 60.22 | 50.78 | 6.57 | 90.5 | 23.29 |
| 8336C CPU\|Python 3.10.14 | | | | | | | | | | | | |
| GPT-3.5-turbo-0301 | 22.92 | 2.77 | 1.7 | 0.34 | 2.07 | 1.91 | 0.0 | 60.65 | 58.28 | 5.69 | 73.7 | 17.07 |
| gpt-3.5-turbo-0613 | 38.48 | 2.78 | 0.9 | 0.33 | 2.64 | 1.92 | 0.0 | 59.92 | 133.49 | 5.80 | 73.9 | 16.54 |
| gpt-3.5-turbo-1106 | 53.63 | 2.84 | 1.2 | 0.34 | 9.03 | 1.94 | 0.2 | 59.43 | 142.42 | 6.04 | 68.3 | 16.38 |
| gpt-4 | 10.01 | 2.71 | 1.6 | 0.32 | 2.33 | 1.92 | 0.0 | 58.96 | 31.21 | 5.57 | 70.1 | 15.05 |
| gpt-4-turbo-preview | 23.00 | 2.80 | 1.2 | 0.33 | 9.03 | 1.94 | 0.2 | 57.17 | 58.01 | 5.81 | 71.4 | 14.91 |
| claude-3-haiku | 22.38 | 2.76 | 0.7 | 0.33 | 2.06 | 1.91 | 0.0 | 59.22 | 56.46 | 5.66 | 75.5 | 15.15 |
| claude-3-sonnet | 14.97 | 2.70 | 0.7 | 0.33 | 2.06 | 1.92 | 0.0 | 60.30 | 43.86 | 5.51 | 73.3 | 19.29 |
| 8336C CPU\|Python 3.9.19 | | | | | | | | | | | | |
| GPT-3.5-turbo-0301 | 22.62 | 2.40 | 1.2 | 0.29 | 2.06 | 1.91 | 0.0 | 60.64 | 57.11 | 4.89 | 29.6 | 14.25 |
| gpt-3.5-turbo-0613 | 39.71 | 2.80 | 0.9 | 0.33 | 2.64 | 1.92 | 0.0 | 59.90 | 137.14 | 5.85 | 73.7 | 16.65 |
| gpt-3.5-turbo-1106 | 53.73 | 2.89 | 1.2 | 0.34 | 9.03 | 1.94 | 0.2 | 59.45 | 142.72 | 6.16 | 72.8 | 16.69 |
| gpt-4 | 10.04 | 2.67 | 0.6 | 0.32 | 2.33 | 1.92 | 0.0 | 58.96 | 31.11 | 5.44 | 72.6 | 15.20 |
| gpt-4-turbo-preview | 22.25 | 2.78 | 1.4 | 0.33 | 9.04 | 1.94 | 0.2 | 57.16 | 56.62 | 5.73 | 72.9 | 14.84 |
| claude-3-haiku | 21.55 | 2.79 | 1.4 | 0.33 | 2.06 | 1.92 | 0.0 | 59.25 | 54.78 | 5.74 | 75.3 | 15.30 |
| claude-3-sonnet | 14.31 | 2.48 | 0.9 | 0.31 | 2.06 | 1.92 | 0.0 | 60.32 | 40.41 | 5.03 | 41.1 | 18.37 |
| 8336C CPU\|Python 3.8.19 | | | | | | | | | | | | |
| GPT-3.5-turbo-0301 | 19.04 | 2.36 | 1.2 | 0.29 | 2.08 | 1.92 | 0.0 | 60.94 | 48.50 | 4.84 | 24.6 | 14.21 |
| gpt-3.5-turbo-0613 | 36.77 | 2.29 | 0.6 | 0.27 | 2.64 | 1.92 | 0.0 | 59.92 | 126.82 | 4.76 | 10.8 | 13.53 |
| gpt-3.5-turbo-1106 | 53.30 | 2.93 | 1.4 | 0.35 | 9.04 | 1.94 | 0.2 | 59.43 | 141.23 | 6.23 | 74.4 | 16.88 |
| gpt-4 | 9.04 | 2.69 | 0.8 | 0.32 | 2.33 | 1.92 | 0.0 | 58.96 | 27.48 | 5.49 | 74.6 | 15.33 |
| gpt-4-turbo-preview | 22.08 | 2.71 | 0.8 | 0.32 | 9.03 | 1.94 | 0.2 | 57.17 | 56.10 | 5.59 | 71.2 | 14.60 |
| claude-3-haiku | 22.10 | 2.77 | 0.7 | 0.33 | 2.06 | 1.92 | 0.0 | 59.25 | 55.93 | 5.73 | 72.3 | 15.39 |
| claude-3-sonnet | 15.91 | 2.76 | 0.7 | 0.34 | 2.06 | 1.92 | 0.0 | 60.29 | 46.86 | 5.68 | 74.2 | 20.17 |
| 4216 CPU \|Python 3.11.2 | | | | | | | | | | | | |
| GPT-3.5-turbo-0301 | 19.42 | 2.32 | 1.2 | 0.28 | 2.07 | 1.92 | 0.0 | 60.94 | 49.48 | 4.74 | 16.1 | 13.95 |
| gpt-3.5-turbo-0613 | 39.10 | 2.44 | 0.6 | 0.29 | 2.65 | 1.92 | 0.0 | 59.94 | 134.30 | 5.08 | 26.7 | 14.10 |
| gpt-3.5-turbo-1106 | 53.92 | 2.91 | 1.4 | 0.35 | 9.04 | 1.94 | 0.2 | 59.44 | 143.13 | 6.20 | 74.0 | 16.63 |
| gpt-4 | 9.62 | 2.70 | 0.8 | 0.32 | 2.33 | 1.92 | 0.0 | 58.94 | 30.08 | 5.52 | 72.6 | 15.25 |
| gpt-4-turbo-preview | 22.71 | 2.76 | 1.1 | 0.33 | 9.03 | 1.94 | 0.2 | 57.17 | 57.82 | 5.68 | 72.0 | 14.87 |
| claude-3-haiku | 23.36 | 2.71 | 0.7 | 0.32 | 2.06 | 1.92 | 0.0 | 59.26 | 58.57 | 5.56 | 69.5 | 15.09 |
| claude-3-sonnet | 15.35 | 2.68 | 0.9 | 0.33 | 2.06 | 1.92 | 0.0 | 60.31 | 44.58 | 5.50 | 68.9 | 19.25 |
| 4116 CPU \|Python 3.11.2 | | | | | | | | | | | | |
| GPT-3.5-turbo-0301 | 19.08 | 2.27 | 1.2 | 0.28 | 2.06 | 1.91 | 0.0 | 60.65 | 48.41 | 4.63 | 11.8 | 13.82 |
| gpt-3.5-turbo-0613 | 38.45 | 2.82 | 1.7 | 0.34 | 2.64 | 1.92 | 0.0 | 59.92 | 132.92 | 5.91 | 70.0 | 16.70 |
| gpt-3.5-turbo-1106 | 54.70 | 2.97 | 1.6 | 0.35 | 9.04 | 1.94 | 0.2 | 59.41 | 145.30 | 6.32 | 75.0 | 16.82 |
| gpt-4 | 9.68 | 2.70 | 1.2 | 0.32 | 2.33 | 1.92 | 0.0 | 58.94 | 29.62 | 5.47 | 72.8 | 15.20 |
| gpt-4-turbo-preview | 22.23 | 2.76 | 0.9 | 0.33 | 9.04 | 1.94 | 0.2 | 57.17 | 59.36 | 5.68 | 74.3 | 14.85 |
| claude-3-haiku | 23.12 | 2.70 | 0.5 | 0.32 | 2.06 | 1.92 | 0.0 | 59.27 | 58.72 | 5.54 | 70.2 | 15.02 |
| claude-3-sonnet | 16.98 | 2.66 | 0.9 | 0.33 | 2.06 | 1.92 | 0.0 | 60.29 | 48.34 | 5.45 | 65.0 | 19.28 |

in different environments, the efficiency changed slightly, which pushes us to consider how can we avoid the bias for different users to use EFFIBENCH to quantify the efficiency of their pre-trained code generation models. To avoid this problem, we provide two different solutions that can maintain the same code execution environment. First, we provide **Request efficiency evaluation form in our Leaderboard and Github**, by filling the request we will evaluate the efficiency of the request pre-trained code generation model and then report it to the user. Second, we also provide a server in the Hugging Face Space where users can directly upload the code generation JSON file and then the server will execute the code locally and then report the efficiency results. The testing time in the server only requires less than one minute for each model (See Appendix A.24).

Table 16: Overhead result of closed-source models efficiency testing time.

| model | time |
|---|---|
| gpt-3.5-turbo-0301 | 32s |
| gpt-3.5-turbo-0613 | 34s |
| gpt-3.5-turbo-1106 | 35s |
| gpt-4 | 37s |
| gpt-4-turbo-preview | 34s |
| claude-3-haiku | 17s |
| claude-3-sonnet | 24s |

Table 17: Efficiency results for different algorithm subsets with GPT-4-turbo-0613.

| Model | max NET | NET | NET>5 | ET (s) | max NMU | NMU | NMU>5 | MU (Mb) | max NTMU | NTMU | NTMU>5 | TMU (Mb*s) | Pass@1 |
|---|---|---|---|---|---|---|---|---|---|---|---|---|---|
| greedy | 3.62 | 3.05 | 0.0 | 0.35 | 2.04 | 1.93 | 0.0 | 58.44 | 7.82 | 6.23 | 92.0 | 16.97 | 41.2 |
| dynamic_programming | 27.10 | 3.40 | 2.3 | 0.42 | 2.64 | 1.94 | 0.0 | 54.25 | 68.94 | 7.10 | 90.6 | 19.63 | 46.2 |
| backtracking | 16.27 | 3.61 | 4.2 | 0.57 | 2.04 | 1.85 | 0.0 | 78.83 | 37.56 | 7.38 | 79.2 | 38.25 | 50.0 |
| divide_and_conquer | 3.59 | 3.21 | 0.0 | 0.35 | 2.03 | 1.95 | 0.0 | 49.39 | 7.67 | 6.64 | 100.0 | 11.61 | 52.4 |
| dfs | 3.52 | 2.96 | 0.0 | 0.37 | 2.06 | 1.84 | 0.0 | 66.01 | 7.31 | 5.88 | 86.7 | 16.26 | 27.8 |
| bfs | 3.41 | 2.92 | 0.0 | 0.36 | 2.06 | 1.84 | 0.0 | 63.07 | 7.04 | 5.75 | 81.2 | 14.98 | 37.2 |
| binary_search | 3.54 | 2.92 | 0.0 | 0.38 | 2.04 | 1.87 | 0.0 | 79.10 | 7.62 | 5.83 | 87.5 | 27.21 | 43.2 |
| two_pointers | 3.58 | 3.08 | 0.0 | 0.37 | 2.04 | 1.94 | 0.0 | 68.99 | 7.52 | 6.33 | 92.9 | 25.72 | 53.3 |
| sliding_window | 3.60 | 3.07 | 0.0 | 0.35 | 2.05 | 1.95 | 0.0 | 64.08 | 7.71 | 6.29 | 95.2 | 21.68 | 60.0 |
| bit_manipulation | 46.70 | 3.97 | 2.0 | 0.46 | 2.18 | 1.96 | 0.0 | 60.87 | 161.12 | 9.42 | 94.1 | 25.09 | 50.0 |
| sorting | 5.58 | 3.03 | 0.9 | 0.36 | 2.04 | 1.89 | 0.0 | 65.15 | 13.79 | 6.12 | 88.3 | 21.50 | 46.6 |

## A.16 Overhead

The overhead of the efficiency evaluation is important as if the overhead of the evaluation is very long, the validity of the results will be questionable. To address this concern, we provide the overhead report for the closed-source models in Table 30. We can observe that the overhead required by each model for efficiency testing is lower than 1 minute. For example, the source code generated by GPT-3.5-turbo-0301 only requires 32 (s) to finish the efficiency testing.

## A.17 Discussion on Time and Space Complexity

In our experiment, we aim to quantify the efficiency of code generated by code generation models with our efficiency metrics. While time and space complexity are conventional metrics in software development for assessing code efficiency, we opted not to rely solely on these for several reasons. Firstly, identical time and space complexity annotations do not guarantee equivalent performance across different implementations. For instance, two algorithms with time complexities expressed as $T(2n)$ and $T(n)$ might both be classified under the same complexity order $O(n)$. However, their practical execution times and resource utilization can vary significantly, underscoring the limitations of using complexity classes as the sole measure of efficiency. Secondly, accurately determining the time and space complexity of a given piece of code typically requires manual analysis and labeling. This process is inherently subjective and prone to human error, making it less suitable for automated, large-scale evaluation of code generation models. The necessity for manual intervention contradicts our goal of automating the efficiency evaluation process as much as possible. Thirdly, although there are models designed to predict the time and space complexity of code, these predictions are often sub-optimal and can be inaccurate[14]. Relying on such models for critical evaluations might introduce significant errors, leading to misleading conclusions about a code generation model's efficiency. Given these considerations, we chose to focus on direct measurements of execution time and memory usage through our specified metrics. These measurements provide a more accurate, objective, and practical assessment of the generated code's efficiency, reflecting real-world performance more closely than theoretical complexity classes. This approach allows for a nuanced analysis of the models' output, enabling a comprehensive evaluation of their practical utility in software development scenarios.

Table 18: Efficiency results for different algorithm subsets with GPT-3.5-turbo-1106

| Model | max NET | NET | NET>5 | ET (s) | max NMU | NMU | NMU>5 | MU (Mb) | max NTMU | NTMU | NTMU>5 | TMU (Mb*s) | Pass@1 |
|---|---|---|---|---|---|---|---|---|---|---|---|---|---|
| greedy | 6.12 | 3.10 | 0.9 | 0.35 | 2.04 | 1.94 | 0.0 | 57.27 | 15.53 | 6.37 | 91.4 | 15.98 | 47.7 |
| dynamic_programming | 68.71 | 3.95 | 3.0 | 0.48 | 9.12 | 1.99 | 0.7 | 55.24 | 182.63 | 8.68 | 91.8 | 21.89 | 48.4 |
| backtracking | 5.38 | 3.19 | 4.8 | 0.46 | 9.12 | 2.27 | 4.8 | 86.95 | 29.17 | 7.34 | 90.5 | 37.27 | 43.8 |
| divide_and_conquer | 3.99 | 3.08 | 0.0 | 0.35 | 2.02 | 1.91 | 0.0 | 51.97 | 8.68 | 6.21 | 92.3 | 11.69 | 61.9 |
| dfs | 3.47 | 2.86 | 0.0 | 0.35 | 2.06 | 1.83 | 0.0 | 63.32 | 7.09 | 5.63 | 85.7 | 14.98 | 32.4 |
| bfs | 6.82 | 3.02 | 3.2 | 0.41 | 2.06 | 1.87 | 0.0 | 62.10 | 14.60 | 6.01 | 83.9 | 17.41 | 36.0 |
| binary_search | 6.13 | 3.02 | 1.4 | 0.38 | 2.05 | 1.89 | 0.0 | 76.83 | 15.97 | 6.13 | 91.3 | 26.15 | 46.6 |
| two_pointers | 3.58 | 3.11 | 0.0 | 0.37 | 2.04 | 1.94 | 0.0 | 67.87 | 7.51 | 6.38 | 93.2 | 24.76 | 56.2 |
| sliding_window | 3.60 | 3.09 | 0.0 | 0.36 | 2.05 | 1.95 | 0.0 | 65.28 | 7.58 | 6.33 | 92.3 | 22.81 | 55.7 |
| bit_manipulation | 47.10 | 4.01 | 2.0 | 0.46 | 2.20 | 1.95 | 0.0 | 61.25 | 163.96 | 9.59 | 92.0 | 25.37 | 49.0 |
| sorting | 6.12 | 3.06 | 0.8 | 0.36 | 2.04 | 1.90 | 0.0 | 63.49 | 15.53 | 6.21 | 88.8 | 20.10 | 52.5 |

Table 19: Efficiency results for different algorithm subsets with GPT-4.

| Model | max NET | NET | NET>5 | ET (s) | max NMU | NMU | NMU>5 | MU (Mb) | max NTMU | NTMU | NTMU>5 | TMU (Mb*s) | Pass@1 |
|---|---|---|---|---|---|---|---|---|---|---|---|---|---|
| greedy | 5.83 | 3.08 | 0.8 | 0.35 | 2.04 | 1.93 | 0.0 | 57.15 | 15.28 | 6.32 | 92.7 | 15.74 | 50.6 |
| dynamic_programming | 4.53 | 3.11 | 0.0 | 0.36 | 2.25 | 1.94 | 0.0 | 53.97 | 10.16 | 6.31 | 91.3 | 15.44 | 49.8 |
| backtracking | 4.53 | 3.01 | 0.0 | 0.44 | 2.03 | 1.84 | 0.0 | 81.67 | 10.16 | 5.89 | 77.3 | 32.23 | 45.8 |
| divide_and_conquer | 3.68 | 3.04 | 0.0 | 0.34 | 2.02 | 1.90 | 0.0 | 53.16 | 7.94 | 6.15 | 87.5 | 11.72 | 38.1 |
| dfs | 3.82 | 3.05 | 0.0 | 0.35 | 2.06 | 1.88 | 0.0 | 57.57 | 7.72 | 6.09 | 93.9 | 13.32 | 30.6 |
| bfs | 11.22 | 3.38 | 5.6 | 0.45 | 2.06 | 1.87 | 0.0 | 55.58 | 25.19 | 6.85 | 91.7 | 19.23 | 41.9 |
| binary_search | 3.69 | 2.96 | 0.0 | 0.38 | 2.04 | 1.88 | 0.0 | 75.09 | 7.78 | 5.92 | 89.3 | 25.24 | 50.7 |
| two_pointers | 3.94 | 3.09 | 0.0 | 0.36 | 2.04 | 1.94 | 0.0 | 66.90 | 8.90 | 6.36 | 95.2 | 23.65 | 59.0 |
| sliding_window | 8.46 | 3.23 | 2.5 | 0.39 | 2.06 | 1.92 | 0.0 | 66.90 | 17.85 | 6.60 | 95.0 | 25.41 | 57.1 |
| bit_manipulation | 4.53 | 3.12 | 0.0 | 0.36 | 2.03 | 1.95 | 0.0 | 60.22 | 10.16 | 6.39 | 92.6 | 18.60 | 52.9 |
| sorting | 13.89 | 3.11 | 1.5 | 0.38 | 2.25 | 1.89 | 0.0 | 63.62 | 43.92 | 6.40 | 90.0 | 21.09 | 54.6 |

Table 20: Efficiency results for different algorithm subsets with GPT-4-turbo-preview.

| Model | max NET | NET | NET>5 | ET (s) | max NMU | NMU | NMU>5 | MU (Mb) | max NTMU | NTMU | NTMU>5 | TMU (Mb*s) | Pass@1 |
|---|---|---|---|---|---|---|---|---|---|---|---|---|---|
| greedy | 3.94 | 3.06 | 0.0 | 0.34 | 2.03 | 1.94 | 0.0 | 55.02 | 8.92 | 6.29 | 92.1 | 14.27 | 67.5 |
| dynamic_programming | 27.00 | 3.42 | 2.6 | 0.41 | 9.13 | 1.98 | 0.5 | 53.49 | 68.48 | 7.26 | 92.6 | 17.01 | 68.2 |
| backtracking | 5.03 | 3.17 | 3.7 | 0.47 | 9.13 | 2.15 | 3.7 | 79.98 | 27.42 | 6.98 | 81.5 | 33.79 | 56.2 |
| divide_and_conquer | 3.52 | 3.06 | 0.0 | 0.36 | 2.03 | 1.89 | 0.0 | 52.62 | 7.63 | 6.16 | 87.5 | 12.52 | 76.2 |
| dfs | 4.09 | 3.05 | 0.0 | 0.36 | 2.05 | 1.85 | 0.0 | 56.57 | 8.89 | 6.04 | 90.0 | 13.26 | 37.0 |
| bfs | 6.42 | 3.09 | 2.6 | 0.41 | 2.05 | 1.86 | 0.0 | 57.33 | 13.66 | 6.15 | 84.6 | 15.53 | 45.3 |
| binary_search | 3.77 | 3.00 | 0.0 | 0.37 | 2.04 | 1.90 | 0.0 | 69.90 | 8.01 | 6.06 | 90.4 | 22.14 | 63.5 |
| two_pointers | 3.74 | 3.10 | 0.0 | 0.36 | 2.04 | 1.95 | 0.0 | 65.53 | 8.30 | 6.37 | 94.0 | 22.56 | 63.8 |
| sliding_window | 8.54 | 3.20 | 1.8 | 0.38 | 2.05 | 1.93 | 0.0 | 61.46 | 17.89 | 6.56 | 92.7 | 21.28 | 78.6 |
| bit_manipulation | 19.79 | 3.39 | 1.5 | 0.43 | 2.03 | 1.95 | 0.0 | 58.06 | 44.72 | 7.02 | 93.9 | 20.90 | 64.7 |
| sorting | 10.40 | 3.07 | 0.6 | 0.37 | 2.04 | 1.89 | 0.0 | 61.40 | 19.63 | 6.20 | 88.1 | 19.34 | 66.8 |

Table 21: Efficiency results for different algorithm subsets with Claude-3-haiku.

| Model | max NET | NET | NET>5 | ET (s) | max NMU | NMU | NMU>5 | MU (Mb) | max NTMU | NTMU | NTMU>5 | TMU (Mb*s) | Pass@1 |
|---|---|---|---|---|---|---|---|---|---|---|---|---|---|
| greedy | 4.09 | 3.22 | 0.0 | 0.36 | 2.03 | 1.94 | 0.0 | 53.85 | 8.71 | 6.62 | 91.8 | 13.99 | 40.3 |
| dynamic_programming | 28.75 | 3.47 | 0.9 | 0.40 | 2.02 | 1.94 | 0.0 | 52.55 | 72.87 | 7.23 | 92.2 | 15.82 | 41.9 |
| backtracking | 4.65 | 3.06 | 0.0 | 0.46 | 2.03 | 1.84 | 0.0 | 90.79 | 10.07 | 6.04 | 75.0 | 39.13 | 33.3 |
| divide_and_conquer | 3.90 | 3.31 | 0.0 | 0.35 | 2.03 | 1.94 | 0.0 | 49.75 | 7.78 | 6.77 | 100.0 | 11.71 | 42.9 |
| dfs | 4.22 | 3.02 | 0.0 | 0.39 | 2.05 | 1.77 | 0.0 | 69.36 | 8.36 | 5.84 | 80.0 | 18.01 | 23.1 |
| bfs | 6.69 | 3.12 | 3.6 | 0.46 | 2.05 | 1.81 | 0.0 | 67.97 | 14.20 | 6.14 | 78.6 | 20.31 | 32.6 |
| binary_search | 4.27 | 3.12 | 0.0 | 0.40 | 2.04 | 1.87 | 0.0 | 78.61 | 9.30 | 6.28 | 87.7 | 29.13 | 43.9 |
| two_pointers | 4.27 | 3.26 | 0.0 | 0.38 | 2.04 | 1.94 | 0.0 | 62.44 | 9.30 | 6.69 | 92.0 | 22.39 | 47.6 |
| sliding_window | 3.90 | 3.20 | 0.0 | 0.38 | 2.05 | 1.94 | 0.0 | 66.27 | 7.89 | 6.57 | 91.9 | 25.85 | 52.9 |
| bit_manipulation | 4.60 | 3.21 | 0.0 | 0.37 | 2.03 | 1.95 | 0.0 | 62.77 | 10.08 | 6.54 | 90.7 | 20.82 | 42.2 |
| sorting | 11.06 | 3.25 | 0.9 | 0.38 | 2.04 | 1.90 | 0.0 | 60.54 | 29.68 | 6.68 | 90.3 | 19.93 | 47.5 |

Table 22: Efficiency results for different algorithm subsets with Claude-3-sonnet.

| Model | max NET | NET | NET>5 | ET (s) | max NMU | NMU | NMU>5 | MU (Mb) | max NTMU | NTMU | NTMU>5 | TMU (Mb*s) | Pass@1 |
|---|---|---|---|---|---|---|---|---|---|---|---|---|---|
| greedy | 3.75 | 3.13 | 0.0 | 0.36 | 2.03 | 1.93 | 0.0 | 58.47 | 7.90 | 6.39 | 90.3 | 16.68 | 42.4 |
| dynamic_programming | 16.34 | 3.42 | 1.8 | 0.47 | 2.04 | 1.94 | 0.0 | 54.95 | 37.83 | 6.96 | 92.0 | 35.81 | 40.8 |
| backtracking | 17.43 | 4.92 | 13.3 | 0.75 | 2.04 | 1.89 | 0.0 | 89.79 | 50.78 | 11.28 | 86.7 | 53.91 | 31.2 |
| divide_and_conquer | 3.56 | 3.03 | 0.0 | 0.36 | 2.02 | 1.88 | 0.0 | 53.44 | 7.18 | 6.01 | 75.0 | 12.62 | 57.1 |
| dfs | 3.61 | 3.03 | 0.0 | 0.36 | 2.05 | 1.81 | 0.0 | 59.20 | 7.53 | 5.94 | 86.2 | 13.98 | 26.9 |
| bfs | 6.24 | 3.08 | 3.4 | 0.42 | 2.05 | 1.84 | 0.0 | 59.57 | 13.17 | 6.06 | 86.2 | 16.36 | 33.7 |
| binary_search | 3.61 | 2.99 | 0.0 | 0.40 | 2.04 | 1.87 | 0.0 | 80.89 | 7.60 | 5.98 | 83.6 | 28.93 | 41.2 |
| two_pointers | 3.61 | 3.18 | 0.0 | 0.38 | 2.05 | 1.94 | 0.0 | 70.62 | 7.53 | 6.54 | 94.1 | 27.10 | 48.6 |
| sliding_window | 3.69 | 3.13 | 0.0 | 0.36 | 2.06 | 1.95 | 0.0 | 64.09 | 7.77 | 6.41 | 95.2 | 22.12 | 60.0 |
| bit_manipulation | 17.43 | 3.51 | 2.4 | 0.40 | 2.02 | 1.95 | 0.0 | 63.19 | 50.78 | 7.56 | 92.9 | 22.32 | 41.2 |
| sorting | 4.98 | 3.10 | 0.0 | 0.37 | 2.04 | 1.89 | 0.0 | 64.34 | 11.81 | 6.27 | 89.2 | 20.90 | 50.4 |

Table 23: Evaluation results of different LLMs efficiency results for EffiBench. We use "*" to represent the results with the new calculation type.

| Model | ET | NET | NET* | MU | NMU | NMU* | TMU | NTMU | NTMU* |
|---|---|---|---|---|---|---|---|---|---|
| gpt-3.5-turbo-0301 | 0.39 | 3.18 | 2.92 | 60.53 | 1.91 | 1.61 | 19.06 | 6.50 | 2.52 |
| gpt-3.5-turbo-0613 | 0.39 | 3.22 | 2.96 | 59.82 | 1.92 | 1.64 | 19.11 | 6.71 | 2.68 |
| gpt-3.5-turbo-1106 | 0.40 | 3.40 | 3.15 | 59.34 | 1.94 | 1.66 | 19.39 | 7.24 | 2.85 |
| gpt-4 | 0.37 | 3.12 | 2.88 | 58.85 | 1.92 | 1.66 | 17.69 | 6.36 | 2.69 |
| gpt-4-turbo-preview | 0.38 | 3.19 | 3.02 | 57.06 | 1.93 | 1.71 | 16.92 | 6.57 | 3.02 |
| claude-3-haiku | 0.39 | 3.28 | 3.00 | 59.15 | 1.91 | 1.64 | 17.99 | 6.71 | 2.66 |
| claude-3-sonnet | 0.40 | 3.22 | 3.05 | 60.22 | 1.91 | 1.62 | 23.29 | 6.57 | 3.13 |

## A.18 Algorithm subsets

## A.19 Calculating the normalized metrics with task level

In Section 3.4, we define the normalized efficiency metrics at the dataset level. For example, NET is defined as:

$$NET = \frac{1}{N} \sum^{N} \frac{T_{\text{code}}}{T_{\text{canonical}}}$$

. In this section, we further discuss the normalized efficiency metrics for LLM-generated code at the dataset level. For example, we set NET* as the dataset-level normalized execution time metric. The NET* is defined as: where $T_{\text{code}}$ is the execution time of the generated code, and $T_{\text{canonical}}$ is the execution time of the canonical solution.

$$NET = \frac{\sum^{N} T_{\text{code}}}{\sum^{N} T_{\text{canonical}}}$$

We follow the setup of Table 4 to evaluate the efficiency of LLM-generated code in 9 open- and closed-source models. The evaluation results are demonstrated in Table 23. We can observe that with the dataset-level normalized metric calculation, the efficiency of LLM-generated code is closer to the canonical solution. For example, GPT-3.5-turbo-0301 generated code required execution time decreases from 3.18x to 2.92x compared to the canonical solution. The key reason is that the dataset-level normalization aggregates the performance across all tasks, potentially masking significant variations in efficiency on individual tasks. While the dataset-level normalized metric, such as NET*, provides a broad overview of the model's performance, it can obscure important details about how well the model handles specific tasks. For example, this dataset-level calculation ignores the metrics evaluated in Table 11. This aggregation can lead to a situation where poor performance on a few tasks is averaged out by better performance on others, giving a potentially misleading impression of overall efficiency.

## A.20 Efficiency distribution for the normalized metrics

As shown in Table 11, we report the efficiency distribution for normalized metrics of the LLM-generated code. In this section, we further break down the efficiency distribution of GPT-3.5-turbo-0301 generated code. Specifically, for each normalized metric, we collect all GPT-3.5-turbo-0301 generated code's efficiency metric. Then we divide them into 100 buckets. Then, we report the accumulated figures in Figure 8. We can observe that most of the GPT-3.5-turbo-0301 generated code is less efficient than the canonical solution (i.e., value = 1).

## A.21 Efficiency of Code with different number of tests

Our experiments in Table 3 only consider 100 tests for each problem, which inspires us to consider how different numbers of tests affect the efficiency of code generated by code generation models. To answer this question, we investigate how does different number of tests affects the efficiency score for each metric. The evaluation results are shown in Table 24, where we can observe that once we increase the tests from 10 to 1,000, the efficiency score for NET, NMU, and NTMU increase

---

[14]https://community.ibm.com/community/user/ai-datascience/blogs/
sepideh-seifzadeh1/2021/10/05/ai-for-code-predict-code-complexity-using-ibms-cod

Table 24: Evaluation result of GPT-3.5-turbo-0301 with the different number of tests for EFFIBENCH. "10" means the evaluation results are obtained with 10 tests.

| number of tests | max NET | NET | NET>5 | ET (s) | max NMU | NMU | NMU>5 | MU (Mb) | max NTMU | NTMU | NTMU>5 | TMU (Mb*s) |
|---|---|---|---|---|---|---|---|---|---|---|---|---|
| 10 | 4.13 | 2.36 | 0.0 | 0.27 | 2.01 | 1.83 | 0.0 | 49.00 | 8.84 | 4.75 | 41.9 | 8.84 |
| 100 | 27.70 | 3.18 | 1.4 | 0.39 | 2.05 | 1.91 | 0.0 | 60.53 | 70.62 | 6.50 | 89.1 | 19.06 |
| 1000 | 66.68 | 3.95 | 4.6 | 0.56 | 11.91 | 2.84 | 5.0 | 162.11 | 436.11 | 10.08 | 66.6 | 340.51 |

Table 25: Evaluation result of GPT-3.5-turbo-0301 with five different executions. The mean and standard deviation (std) values are reported to two decimal places.

| number of tests | max NET | NET | NET>5 | ET (s) | max NMU | NMU | NMU>5 | MU (Mb) | max NTMU | NTMU | NTMU>5 | TMU (Mb*s) |
|---|---|---|---|---|---|---|---|---|---|---|---|---|
| 0 | 27.70 | 3.18 | 1.4 | 0.39 | 2.05 | 1.91 | 0.0 | 60.53 | 70.62 | 6.50 | 89.1 | 19.06 |
| 1 | 27.70 | 3.17 | 1.4 | 0.39 | 2.06 | 1.91 | 0.0 | 60.55 | 70.50 | 6.48 | 89.1 | 19.07 |
| 2 | 27.76 | 3.17 | 1.4 | 0.38 | 2.06 | 1.91 | 0.0 | 60.55 | 70.41 | 6.52 | 89.1 | 19.21 |
| 3 | 27.42 | 3.18 | 1.4 | 0.39 | 2.05 | 1.91 | 0.0 | 60.54 | 70.70 | 6.70 | 89.2 | 18.95 |
| 4 | 27.78 | 3.18 | 1.4 | 0.39 | 2.05 | 1.91 | 0.0 | 60.53 | 70.48 | 6.41 | 89.1 | 19.05 |
| Mean | 27.67 | 3.18 | 1.4 | 0.39 | 2.05 | 1.91 | 0.0 | 60.54 | 70.54 | 6.52 | 89.1 | 19.07 |
| Std | 0.13 | 0.00 | 0.0 | 0.00 | 0.00 | 0.00 | 0.0 | 0.01 | 0.10 | 0.10 | 0.0 | 0.09 |

for GPT-3.5-turbo-0301. For example, the GPT-3.5-turbo-0301's NTMU increases from 4.75 to 10.08. We indicate that the key reason is once we increase the number of tests, more edge cases would be covered (e.g., more length, data distribution). However, since the tests for the efficiency experiments, the overhead such as memory usage increases largely. For example, when we increase the tests from 100 to 1,000, the TMU increases from 8.84 MB*s to 340.51 MB*s, which requires more computation resources for experiments. So in our experiments and Leaderboard, we focus on studying the LLM-generated code efficiency in 100 tests.

## A.22 Difficulty

We also provide the efficiency results of all open- and closed-source models in the different difficulty in Table 27-29. We can observe that the pass@1 of open-source LLMs is very low.

## A.23 Randomness

**Seed** We also evaluated the efficiency of the code generated by GPT-3.5-turbo-0301 five times in the same environments to ensure the reliability of our results. As shown in Table 25, we can observe that performance metrics such as ET, MU, and TMU show remarkable consistency across different executions. Specifically, the standard deviations (std) for these metrics are exceptionally low, demonstrating minimal variability and highlighting the stability of the code execution in our testing environment. This consistent performance underpins the robustness of our experimental approach, providing a solid foundation for further analysis of the model's operational characteristics.

**Environment** We also provide an analysis of the efficiency of the code generated by closed-source models in different local environments. The results are shown in Table 26, where we can observe that in different environments, the efficiency changed slightly, which pushes us to consider how can we avoid the bias for different users to use EffiBench to quantify the efficiency of their pre-trained code generation models. To avoid this problem, we provide two different solutions that can maintain the same code execution environment. First, we provide **Request efficiency evaluation form in our Leaderboard and Github**, by filling the request we will evaluate the efficiency of the request pre-trained code generation model and then report it to the user. Second, we also provide a server in the Hugging Face Space where users can directly upload the code generation JSON file and then the server will execute the code locally and then report the efficiency results. The testing time in the server only requires less than one minute for each model (See Appendix A.24).

## A.24 Overhead

We provide the overhead report for the closed-source models in Table 30. We can observe that the overhead required by each model for efficiency testing is lower than 1 minute.

Table 26: Evaluation result of closed-source models for different environments. Both the canonical solution and LLM-generated code were executed in the same environments.

| number of tests | max NET | NET | NET>5 | ET (s) | max NMU | NMU | NMU>5 | MU (Mb) | max NTMU | NTMU | NTMU>5 | TMU (Mb*s) |
|---|---|---|---|---|---|---|---|---|---|---|---|---|
| **8336C CPU\|Python 3.11.2** | | | | | | | | | | | | |
| gpt-3.5-turbo-0301 | 27.70 | 3.18 | 1.4 | 0.39 | 2.05 | 1.91 | 0.0 | 60.53 | 70.62 | 6.50 | 89.1 | 19.06 |
| gpt-3.5-turbo-0613 | 46.70 | 3.22 | 0.9 | 0.39 | 2.64 | 1.92 | 0.0 | 59.82 | 161.12 | 6.71 | 89.9 | 19.11 |
| gpt-3.5-turbo-1106 | 68.71 | 3.40 | 1.6 | 0.40 | 9.12 | 1.94 | 0.2 | 59.34 | 182.63 | 7.24 | 90.9 | 19.39 |
| gpt-4 | 13.89 | 3.12 | 1.0 | 0.37 | 2.25 | 1.92 | 0.0 | 58.85 | 43.92 | 6.36 | 91.1 | 17.69 |
| gpt-4-turbo-preview | 27.00 | 3.19 | 1.2 | 0.38 | 9.13 | 1.93 | 0.2 | 57.06 | 68.48 | 6.57 | 91.1 | 16.92 |
| claude-3-haiku | 28.75 | 3.28 | 0.7 | 0.39 | 2.05 | 1.91 | 0.0 | 59.15 | 72.87 | 6.71 | 90.0 | 17.99 |
| claude-3-sonnet | 17.43 | 3.22 | 0.9 | 0.40 | 2.06 | 1.91 | 0.0 | 60.22 | 50.78 | 6.57 | 90.5 | 23.29 |
| **8336C CPU\|Python 3.10.14** | | | | | | | | | | | | |
| GPT-3.5-turbo-0301 | 22.92 | 2.77 | 1.7 | 0.34 | 2.07 | 1.91 | 0.0 | 60.65 | 58.28 | 5.69 | 73.7 | 17.07 |
| gpt-3.5-turbo-0613 | 38.48 | 2.78 | 0.9 | 0.33 | 2.64 | 1.92 | 0.0 | 59.92 | 133.49 | 5.80 | 73.9 | 16.54 |
| gpt-3.5-turbo-1106 | 53.63 | 2.84 | 1.2 | 0.34 | 9.03 | 1.94 | 0.2 | 59.43 | 142.42 | 6.04 | 68.3 | 16.38 |
| gpt-4 | 10.01 | 2.71 | 1.6 | 0.32 | 2.33 | 1.92 | 0.0 | 58.96 | 31.21 | 5.57 | 70.1 | 15.05 |
| gpt-4-turbo-preview | 23.00 | 2.80 | 1.2 | 0.33 | 9.03 | 1.94 | 0.2 | 57.17 | 58.01 | 5.81 | 71.4 | 14.91 |
| claude-3-haiku | 22.38 | 2.76 | 0.7 | 0.33 | 2.06 | 1.91 | 0.0 | 59.22 | 56.46 | 5.66 | 75.5 | 15.15 |
| claude-3-sonnet | 14.97 | 2.70 | 0.7 | 0.33 | 2.06 | 1.92 | 0.0 | 60.30 | 43.86 | 5.51 | 73.3 | 19.29 |
| **8336C CPU\|Python 3.9.19** | | | | | | | | | | | | |
| GPT-3.5-turbo-0301 | 22.62 | 2.40 | 1.2 | 0.29 | 2.06 | 1.91 | 0.0 | 60.64 | 57.11 | 4.89 | 29.6 | 14.25 |
| gpt-3.5-turbo-0613 | 39.71 | 2.80 | 0.9 | 0.33 | 2.64 | 1.92 | 0.0 | 59.90 | 137.14 | 5.85 | 73.7 | 16.65 |
| gpt-3.5-turbo-1106 | 53.73 | 2.89 | 1.2 | 0.34 | 9.03 | 1.94 | 0.2 | 59.45 | 142.72 | 6.16 | 72.8 | 16.69 |
| gpt-4 | 10.04 | 2.67 | 0.6 | 0.32 | 2.33 | 1.92 | 0.0 | 58.96 | 31.11 | 5.44 | 72.6 | 15.20 |
| gpt-4-turbo-preview | 22.25 | 2.78 | 1.4 | 0.33 | 9.04 | 1.94 | 0.2 | 57.16 | 56.62 | 5.73 | 72.9 | 14.84 |
| claude-3-haiku | 21.55 | 2.79 | 1.4 | 0.33 | 2.06 | 1.92 | 0.0 | 59.25 | 54.78 | 5.74 | 75.3 | 15.30 |
| claude-3-sonnet | 14.31 | 2.48 | 0.9 | 0.31 | 2.06 | 1.92 | 0.0 | 60.32 | 40.41 | 5.03 | 41.1 | 18.37 |
| **8336C CPU\|Python 3.8.19** | | | | | | | | | | | | |
| GPT-3.5-turbo-0301 | 19.04 | 2.36 | 1.2 | 0.29 | 2.08 | 1.92 | 0.0 | 60.94 | 48.50 | 4.84 | 24.6 | 14.21 |
| gpt-3.5-turbo-0613 | 36.77 | 2.29 | 0.6 | 0.27 | 2.64 | 1.92 | 0.0 | 59.92 | 126.82 | 4.76 | 10.8 | 13.53 |
| gpt-3.5-turbo-1106 | 53.30 | 2.93 | 1.4 | 0.35 | 9.04 | 1.94 | 0.2 | 59.43 | 141.23 | 6.23 | 74.4 | 16.88 |
| gpt-4 | 9.04 | 2.69 | 0.8 | 0.32 | 2.33 | 1.92 | 0.0 | 58.96 | 27.48 | 5.49 | 74.6 | 15.33 |
| gpt-4-turbo-preview | 22.08 | 2.71 | 0.8 | 0.32 | 9.03 | 1.94 | 0.2 | 57.17 | 56.10 | 5.59 | 71.2 | 14.60 |
| claude-3-haiku | 22.10 | 2.77 | 0.7 | 0.33 | 2.06 | 1.92 | 0.0 | 59.25 | 55.93 | 5.73 | 72.3 | 15.39 |
| claude-3-sonnet | 15.91 | 2.76 | 0.7 | 0.34 | 2.06 | 1.92 | 0.0 | 60.29 | 46.86 | 5.68 | 74.2 | 20.17 |
| **4216 CPU \|Python 3.11.2** | | | | | | | | | | | | |
| GPT-3.5-turbo-0301 | 19.42 | 2.32 | 1.2 | 0.28 | 2.07 | 1.92 | 0.0 | 60.94 | 49.48 | 4.74 | 16.1 | 13.95 |
| gpt-3.5-turbo-0613 | 39.10 | 2.44 | 0.6 | 0.29 | 2.65 | 1.92 | 0.0 | 59.94 | 134.30 | 5.08 | 26.7 | 14.10 |
| gpt-3.5-turbo-1106 | 53.92 | 2.91 | 1.4 | 0.35 | 9.04 | 1.94 | 0.2 | 59.44 | 143.13 | 6.20 | 74.0 | 16.63 |
| gpt-4 | 9.62 | 2.70 | 0.8 | 0.32 | 2.33 | 1.92 | 0.0 | 58.94 | 30.08 | 5.52 | 72.6 | 15.25 |
| gpt-4-turbo-preview | 22.71 | 2.76 | 1.1 | 0.33 | 9.03 | 1.94 | 0.2 | 57.17 | 57.82 | 5.68 | 72.0 | 14.87 |
| claude-3-haiku | 23.36 | 2.71 | 0.7 | 0.32 | 2.06 | 1.92 | 0.0 | 59.26 | 58.57 | 5.56 | 69.5 | 15.09 |
| claude-3-sonnet | 15.35 | 2.68 | 0.9 | 0.33 | 2.06 | 1.92 | 0.0 | 60.31 | 44.58 | 5.50 | 68.9 | 19.25 |
| **4116 CPU \|Python 3.11.2** | | | | | | | | | | | | |
| GPT-3.5-turbo-0301 | 19.08 | 2.27 | 1.2 | 0.28 | 2.06 | 1.91 | 0.0 | 60.65 | 48.41 | 4.63 | 11.8 | 13.82 |
| gpt-3.5-turbo-0613 | 38.45 | 2.82 | 1.7 | 0.34 | 2.64 | 1.92 | 0.0 | 59.92 | 132.92 | 5.91 | 70.0 | 16.70 |
| gpt-3.5-turbo-1106 | 54.70 | 2.97 | 1.6 | 0.35 | 9.04 | 1.94 | 0.2 | 59.41 | 145.30 | 6.32 | 75.0 | 16.82 |
| gpt-4 | 9.68 | 2.70 | 1.2 | 0.32 | 2.33 | 1.92 | 0.0 | 58.94 | 29.62 | 5.47 | 72.8 | 15.20 |
| gpt-4-turbo-preview | 22.23 | 2.76 | 0.9 | 0.33 | 9.04 | 1.94 | 0.2 | 57.17 | 59.36 | 5.68 | 74.3 | 14.85 |
| claude-3-haiku | 23.12 | 2.70 | 0.5 | 0.32 | 2.06 | 1.92 | 0.0 | 59.27 | 58.72 | 5.54 | 70.2 | 15.02 |
| claude-3-sonnet | 16.98 | 2.66 | 0.9 | 0.33 | 2.06 | 1.92 | 0.0 | 60.29 | 48.34 | 5.45 | 65.0 | 19.28 |

## A.25 Discussion on Time and Space Complexity

In our experiment, we aim to quantify the efficiency of code generated by code generation models with our efficiency metrics. While time and space complexity are conventional metrics in software development for assessing code efficiency, we opted not to rely solely on these for several reasons. Firstly, identical time and space complexity annotations do not guarantee equivalent performance across different implementations. For instance, two algorithms with time complexities expressed as $T(2n)$ and $T(n)$ might both be classified under the same complexity order $O(n)$. However, their practical execution times and resource utilization can vary significantly, underscoring the limitations of using complexity classes as the sole measure of efficiency. Secondly, accurately determining the time and space complexity of a given piece of code typically requires manual analysis and labeling. This process is inherently subjective and prone to human error, making it less suitable for automated, large-scale evaluation of code generation models. The necessity for manual intervention contradicts our goal of automating the efficiency evaluation process as much as possible. Thirdly, although there are models designed to predict the time and space complexity of code, these predictions are often sub-optimal and can be inaccurate[15]. Relying on such models for critical evaluations might introduce significant errors, leading to misleading conclusions about a code generation model's efficiency. Given these considerations, we chose to focus on direct measurements of execution time and memory usage through our specified metrics. These measurements provide a more accurate, objective, and practical assessment of the generated code's efficiency, reflecting real-world performance more closely than

---

[15]https://community.ibm.com/community/user/ai-datascience/blogs/
sepideh-seifzadeh1/2021/10/05/ai-for-code-predict-code-complexity-using-ibms-cod

Table 27: Code efficiency of widely-studied LLMs reported by EFFIBENCH (Easy).

| Model | max NET | NET | NET>5 | ET (s) | max NMU | NMU | NMU>5 | MU (Mb) | max NTMU | NTMU | NTMU>5 | TMU (Mb*s) | Pass@1 |
|---|---|---|---|---|---|---|---|---|---|---|---|---|---|
| **Open-source models** | | | | | | | | | | | | | |
| CodeLlama-7b-hf | 3.09 | 2.93 | 0.0 | 0.30 | 2.05 | 2.00 | 0.0 | 48.10 | 6.39 | 5.97 | 100.0 | 9.81 | 0.7 |
| CodeLlama-13b-hf | 3.21 | 2.91 | 0.0 | 0.31 | 2.05 | 1.93 | 0.0 | 50.40 | 6.53 | 5.81 | 88.9 | 10.28 | 0.9 |
| CodeLlama-34b-hf | 3.34 | 3.00 | 0.0 | 0.32 | 2.06 | 1.95 | 0.0 | 50.13 | 6.93 | 6.09 | 94.6 | 10.47 | 3.7 |
| CodeLlama-70b-hf | 7.56 | 3.02 | 2.3 | 0.36 | 2.06 | 1.92 | 0.0 | 62.08 | 15.82 | 6.14 | 90.7 | 18.22 | 4.3 |
| CodeLlama-7b-Instruct-hf | 15.89 | 3.45 | 4.0 | 0.40 | 2.05 | 1.92 | 0.0 | 68.21 | 46.14 | 7.51 | 88.0 | 21.95 | 2.5 |
| CodeLlama-13b-Instruct-hf | 4.46 | 2.92 | 0.0 | 0.36 | 2.06 | 1.89 | 0.0 | 75.56 | 10.22 | 5.90 | 89.5 | 23.28 | 3.8 |
| CodeLlama-34b-Instruct-hf | 3.50 | 2.91 | 0.0 | 0.35 | 2.05 | 1.92 | 0.0 | 69.94 | 7.37 | 5.88 | 89.1 | 20.21 | 4.6 |
| CodeLlama-70b-Instruct-hf | 3.13 | 2.92 | 0.0 | 0.31 | 2.06 | 1.96 | 0.0 | 49.55 | 6.68 | 5.97 | 95.5 | 10.13 | 2.2 |
| deepseek-coder-1.3b-instruct | 3.11 | 2.89 | 0.0 | 0.31 | 2.03 | 1.92 | 0.0 | 50.52 | 6.41 | 5.85 | 90.0 | 10.22 | 1.0 |
| deepseek-coder-6.7b-instruct | 5.59 | 3.02 | 4.3 | 0.35 | 2.05 | 1.94 | 0.0 | 68.72 | 13.81 | 6.23 | 95.7 | 20.11 | 2.3 |
| deepseek-coder-6.7b-base | 12.25 | 3.10 | 2.0 | 0.40 | 2.14 | 1.90 | 0.0 | 61.01 | 23.39 | 6.23 | 88.2 | 19.92 | 5.1 |
| deepseek-coder-33b-base | 19.54 | 3.25 | 1.4 | 0.36 | 37.39 | 2.42 | 1.4 | 68.90 | 604.11 | 14.33 | 91.8 | 28.06 | 7.3 |
| OpenCodeInterpreter-DS-1.3B | 3.36 | 2.88 | 0.0 | 0.35 | 2.05 | 1.91 | 0.0 | 71.55 | 7.09 | 5.79 | 91.3 | 21.27 | 2.3 |
| OpenCodeInterpreter-DS-6.7B | 5.70 | 2.94 | 2.3 | 0.35 | 2.04 | 1.88 | 0.0 | 63.79 | 14.14 | 5.91 | 84.1 | 17.08 | 4.4 |
| OpenCodeInterpreter-DS-33B | 3.67 | 2.98 | 0.0 | 0.33 | 2.05 | 1.91 | 0.0 | 57.64 | 7.68 | 6.08 | 90.1 | 13.91 | 7.1 |
| Phind-CodeLlama-34B-v1 | 3.51 | 2.96 | 0.0 | 0.34 | 2.06 | 1.93 | 0.0 | 62.33 | 7.76 | 6.01 | 91.7 | 16.52 | 3.6 |
| Phind-CodeLlama-34B-v2 | 4.80 | 3.00 | 0.0 | 0.35 | 2.04 | 1.90 | 0.0 | 64.35 | 11.30 | 6.08 | 87.1 | 17.84 | 7.0 |
| starcoder | 3.22 | 2.77 | 0.0 | 0.37 | 2.06 | 1.87 | 0.0 | 88.70 | 6.57 | 5.47 | 84.6 | 29.48 | 1.3 |
| starcoder2-3b | 3.08 | 2.87 | 0.0 | 0.32 | 2.01 | 1.91 | 0.0 | 53.76 | 6.61 | 5.76 | 87.5 | 10.85 | 0.8 |
| starcoder2-7b | 5.19 | 3.18 | 14.3 | 0.34 | 2.06 | 2.00 | 0.0 | 48.15 | 12.69 | 6.78 | 100.0 | 11.47 | 0.7 |
| starcoder2-15b | 3.01 | 2.41 | 0.0 | 0.47 | 1.99 | 1.59 | 0.0 | 152.41 | 6.14 | 4.24 | 40.0 | 62.31 | 0.5 |
| starcoderbase | 3.34 | 2.77 | 0.0 | 0.38 | 2.04 | 1.81 | 0.0 | 91.97 | 7.09 | 5.44 | 75.0 | 29.75 | 1.2 |
| WizardCoder-13B | 3.20 | 2.81 | 0.0 | 0.35 | 2.05 | 1.87 | 0.0 | 77.99 | 6.51 | 5.58 | 82.4 | 23.33 | 1.7 |
| WizardCoder-15B | 3.17 | 2.78 | 0.0 | 0.36 | 2.04 | 1.89 | 0.0 | 82.49 | 6.61 | 5.55 | 85.7 | 25.89 | 1.4 |
| XwinCoder-13B | 3.25 | 2.91 | 0.0 | 0.33 | 2.05 | 1.94 | 0.0 | 61.49 | 6.71 | 5.92 | 92.3 | 16.25 | 3.9 |
| XwinCoder-34B | 4.45 | 2.97 | 0.0 | 0.34 | 2.05 | 1.91 | 0.0 | 62.77 | 10.42 | 6.01 | 88.2 | 16.80 | 7.6 |
| Yi-34B-200K | 3.17 | 2.89 | 0.0 | 0.31 | 2.04 | 1.94 | 0.0 | 50.86 | 6.78 | 5.87 | 85.7 | 10.36 | 2.1 |
| Yi-34B-Chat | 3.15 | 2.74 | 0.0 | 0.36 | 2.03 | 1.87 | 0.0 | 82.08 | 6.69 | 5.45 | 86.7 | 26.12 | 1.5 |
| Yi-34B | 3.17 | 2.77 | 0.0 | 0.38 | 2.04 | 1.84 | 0.0 | 92.73 | 6.77 | 5.43 | 83.3 | 30.39 | 1.2 |
| Artigenz-Coder-DS-6.7B | 15.96 | 3.26 | 1.9 | 0.37 | 2.21 | 1.91 | 0.0 | 59.91 | 46.16 | 6.80 | 91.5 | 16.75 | 10.6 |
| CodeFuse-DeepSeek-33B | 6.10 | 3.16 | 1.1 | 0.34 | 2.05 | 1.93 | 0.0 | 50.94 | 15.19 | 6.47 | 91.5 | 11.47 | 9.4 |
| codegemma-7b | 8.09 | 3.13 | 2.0 | 0.35 | 2.05 | 1.94 | 0.0 | 58.52 | 20.96 | 6.46 | 94.0 | 16.07 | 5.0 |
| Magicoder-S-DS-6.7B | 5.15 | 3.01 | 0.9 | 0.34 | 2.06 | 1.92 | 0.0 | 58.97 | 12.56 | 6.09 | 88.9 | 14.98 | 11.7 |
| Mistral-7B-codealpaca-lora | 3.11 | 2.81 | 0.0 | 0.31 | 2.04 | 1.92 | 0.0 | 52.51 | 6.45 | 5.68 | 90.9 | 10.47 | 1.1 |
| octocoder | 2.99 | 2.99 | 0.0 | 0.30 | 2.02 | 2.01 | 0.0 | 47.93 | 6.20 | 6.12 | 100.0 | 9.66 | 0.2 |
| **Closed-source models** | | | | | | | | | | | | | |
| gpt-3.5-turbo-0301 | 16.24 | 3.14 | 0.9 | 0.35 | 2.05 | 1.92 | 0.0 | 58.65 | 46.95 | 6.47 | 90.2 | 15.49 | 11.2 |
| gpt-3.5-turbo-0613 | 5.58 | 3.07 | 0.8 | 0.34 | 2.05 | 1.93 | 0.0 | 57.70 | 13.79 | 6.28 | 93.3 | 15.00 | 12.0 |
| gpt-3.5-turbo-1106 | 4.78 | 3.10 | 0.0 | 0.35 | 2.05 | 1.93 | 0.0 | 57.82 | 11.23 | 6.32 | 91.7 | 14.97 | 12.1 |
| gpt-4 | 8.46 | 3.12 | 0.7 | 0.36 | 2.25 | 1.92 | 0.0 | 57.52 | 17.85 | 6.34 | 92.0 | 15.93 | 13.7 |
| gpt-4-turbo-preview | 10.40 | 3.18 | 1.2 | 0.35 | 2.05 | 1.92 | 0.0 | 56.58 | 19.63 | 6.49 | 92.5 | 16.17 | 16.1 |
| claude-3-haiku | 11.06 | 3.31 | 0.9 | 0.37 | 2.05 | 1.92 | 0.0 | 58.59 | 29.68 | 6.81 | 92.2 | 16.68 | 11.6 |
| claude-3-sonnet | 4.98 | 3.16 | 0.0 | 0.35 | 2.06 | 1.93 | 0.0 | 57.35 | 11.81 | 6.45 | 92.4 | 15.13 | 13.1 |

Table 28: Code efficiency of widely-studied LLMs reported by EFFIBENCH (Medium).

| Model | max NET | NET | NET>5 | ET (s) | max NMU | NMU | NMU>5 | MU (Mb) | max NTMU | NTMU | NTMU>5 | TMU (Mb*s) | Pass@1 |
|---|---|---|---|---|---|---|---|---|---|---|---|---|---|
| **Open-source models** | | | | | | | | | | | | | |
| CodeLlama-7b-hf | 3.25 | 2.50 | 0.0 | 0.31 | 1.99 | 1.71 | 0.0 | 49.34 | 6.80 | 4.87 | 75.0 | 10.23 | 0.4 |
| CodeLlama-13b-hf | 2.60 | 1.80 | 0.0 | 0.78 | 1.93 | 1.46 | 0.0 | 347.49 | 5.32 | 3.16 | 50.0 | 194.83 | 0.2 |
| CodeLlama-34b-hf | 4.46 | 2.98 | 0.0 | 0.35 | 2.06 | 1.91 | 0.0 | 59.86 | 9.17 | 6.00 | 93.0 | 16.15 | 4.3 |
| CodeLlama-70b-hf | 13.92 | 3.40 | 7.5 | 0.52 | 2.05 | 1.86 | 0.0 | 65.11 | 32.04 | 6.84 | 82.5 | 28.69 | 4.0 |
| CodeLlama-7b-Instruct-hf | 3.36 | 2.75 | 0.0 | 0.40 | 2.03 | 1.86 | 0.0 | 94.89 | 6.72 | 5.49 | 84.2 | 37.34 | 1.9 |
| CodeLlama-13b-Instruct-hf | 3.95 | 2.90 | 0.0 | 0.34 | 2.48 | 1.93 | 0.0 | 59.30 | 10.07 | 5.89 | 94.7 | 15.75 | 3.8 |
| CodeLlama-34b-Instruct-hf | 13.66 | 3.19 | 1.9 | 0.40 | 2.06 | 1.93 | 0.0 | 55.59 | 31.46 | 6.48 | 88.7 | 18.68 | 5.3 |
| CodeLlama-70b-Instruct-hf | 14.60 | 3.20 | 2.5 | 0.44 | 2.06 | 1.92 | 0.0 | 57.22 | 33.69 | 6.52 | 87.5 | 24.52 | 4.0 |
| deepseek-coder-1.3b-instruct | 3.63 | 2.80 | 0.0 | 0.34 | 2.03 | 1.91 | 0.0 | 61.17 | 8.13 | 5.61 | 90.3 | 14.45 | 3.1 |
| deepseek-coder-6.7b-instruct | 4.53 | 2.79 | 0.0 | 0.41 | 2.57 | 1.86 | 0.0 | 83.79 | 10.72 | 5.58 | 83.3 | 35.78 | 3.6 |
| deepseek-coder-6.7b-base | 6.63 | 2.94 | 1.2 | 0.37 | 2.06 | 1.92 | 0.0 | 67.61 | 14.18 | 5.93 | 91.9 | 21.92 | 8.6 |
| deepseek-coder-33b-base | 14.85 | 3.14 | 1.7 | 0.42 | 2.07 | 1.93 | 0.0 | 59.01 | 26.99 | 6.36 | 92.4 | 22.62 | 11.8 |
| OpenCodeInterpreter-DS-1.3B | 3.35 | 2.79 | 0.0 | 0.35 | 2.05 | 1.88 | 0.0 | 66.99 | 7.02 | 5.55 | 79.3 | 23.06 | 2.9 |
| OpenCodeInterpreter-DS-6.7B | 6.03 | 2.96 | 1.4 | 0.39 | 2.37 | 1.91 | 0.0 | 65.96 | 12.85 | 5.98 | 87.8 | 21.98 | 7.4 |
| OpenCodeInterpreter-DS-33B | 17.39 | 3.11 | 2.3 | 0.40 | 2.14 | 1.91 | 0.0 | 62.60 | 49.99 | 6.37 | 87.5 | 21.25 | 12.8 |
| Phind-CodeLlama-34B-v1 | 3.57 | 2.87 | 0.0 | 0.39 | 2.06 | 1.87 | 0.0 | 74.39 | 7.13 | 5.67 | 85.1 | 28.43 | 6.7 |
| Phind-CodeLlama-34B-v2 | 53.08 | 3.53 | 2.1 | 0.49 | 2.60 | 1.88 | 0.0 | 79.87 | 139.88 | 7.48 | 84.5 | 35.83 | 9.7 |
| starcoder | 3.34 | 2.91 | 0.0 | 0.31 | 2.04 | 1.96 | 0.0 | 49.06 | 6.88 | 5.94 | 88.2 | 10.17 | 1.7 |
| starcoder2-3b | 3.13 | 2.93 | 0.0 | 0.31 | 2.04 | 1.99 | 0.0 | 48.02 | 6.49 | 5.99 | 100.0 | 10.02 | 0.4 |
| starcoder2-7b | 3.01 | 2.86 | 0.0 | 0.30 | 2.03 | 1.97 | 0.0 | 48.94 | 6.17 | 5.82 | 100.0 | 9.89 | 0.7 |
| starcoder2-15b | 2.88 | 2.88 | 0.0 | 0.31 | 2.01 | 2.01 | 0.0 | 47.64 | 6.04 | 6.04 | 100.0 | 10.14 | 0.1 |
| starcoderbase | 2.95 | 2.87 | 0.0 | 0.31 | 2.05 | 1.94 | 0.0 | 49.43 | 5.84 | 5.77 | 100.0 | 10.04 | 0.4 |
| WizardCoder-Python-13B-V1.0-GPTQ | 3.16 | 2.55 | 0.0 | 0.40 | 2.03 | 1.76 | 0.0 | 98.77 | 6.45 | 4.85 | 72.7 | 32.50 | 1.1 |
| WizardCoder-15B-V1.0 | 4.07 | 2.88 | 0.0 | 0.35 | 2.06 | 1.91 | 0.0 | 67.82 | 9.51 | 5.86 | 84.6 | 17.41 | 1.3 |
| XwinCoder-13B | 3.39 | 2.92 | 0.0 | 0.32 | 2.04 | 1.93 | 0.0 | 54.69 | 6.93 | 5.91 | 95.0 | 12.55 | 4.0 |
| XwinCoder-34B | 6.32 | 2.96 | 1.1 | 0.34 | 2.42 | 1.92 | 0.0 | 55.50 | 17.70 | 6.00 | 87.5 | 12.77 | 8.8 |
| Yi-34B-200K | 3.16 | 2.99 | 0.0 | 0.31 | 2.06 | 2.00 | 0.0 | 47.90 | 6.49 | 6.16 | 100.0 | 9.90 | 1.1 |
| Yi-34B-Chat | 2.98 | 2.76 | 0.0 | 0.32 | 2.05 | 1.89 | 0.0 | 55.22 | 6.16 | 5.44 | 90.0 | 11.22 | 1.0 |
| Yi-34B | 3.38 | 2.74 | 0.0 | 0.37 | 2.05 | 1.91 | 0.0 | 86.38 | 7.13 | 5.57 | 90.0 | 28.88 | 1.0 |
| Artigenz-Coder-DS-6.7B | 9.69 | 3.11 | 1.5 | 0.39 | 2.48 | 1.91 | 0.0 | 66.56 | 26.21 | 6.35 | 90.0 | 22.60 | 20.1 |
| CodeFuse-DeepSeek-33B | 3.80 | 2.99 | 0.0 | 0.37 | 2.06 | 1.89 | 0.0 | 64.54 | 7.75 | 6.00 | 84.7 | 20.30 | 16.3 |
| codegemma-7b | 3.40 | 2.94 | 0.0 | 0.34 | 2.06 | 1.92 | 0.0 | 54.46 | 7.14 | 5.93 | 90.5 | 12.60 | 6.3 |
| Magicoder-S-DS-6.7B | 6.73 | 2.98 | 0.5 | 0.37 | 2.33 | 1.90 | 0.0 | 63.72 | 14.24 | 5.99 | 88.0 | 19.55 | 19.2 |
| Mistral-7B-codealpaca-lora | 3.82 | 2.92 | 0.0 | 0.32 | 2.36 | 1.97 | 0.0 | 51.26 | 9.20 | 6.01 | 92.3 | 10.64 | 1.3 |
| octocoder | 2.78 | 2.35 | 0.0 | 0.34 | 1.99 | 1.66 | 0.0 | 70.03 | 5.60 | 4.02 | 50.0 | 13.37 | 0.2 |
| **Closed-source models** | | | | | | | | | | | | | |
| gpt-3.5-turbo-0301 | 17.25 | 3.12 | 1.7 | 0.40 | 2.05 | 1.89 | 0.0 | 64.74 | 43.96 | 6.34 | 88.5 | 22.27 | 23.5 |
| gpt-3.5-turbo-0613 | 46.70 | 3.23 | 0.8 | 0.40 | 2.64 | 1.91 | 0.0 | 63.86 | 161.12 | 6.90 | 88.5 | 22.50 | 26.0 |
| gpt-3.5-turbo-1106 | 68.71 | 3.45 | 2.1 | 0.42 | 9.12 | 1.94 | 0.4 | 63.09 | 182.63 | 7.46 | 89.7 | 22.55 | 28.2 |
| gpt-4 | 13.89 | 3.14 | 1.4 | 0.30 | 2.16 | 1.91 | 0.0 | 62.30 | 43.92 | 6.43 | 90.3 | 20.56 | 27.9 |
| gpt-4-turbo-preview | 22.41 | 3.14 | 1.1 | 0.38 | 9.13 | 1.93 | 0.3 | 59.86 | 65.33 | 6.47 | 90.0 | 18.45 | 36.0 |
| claude-3-haiku | 6.69 | 3.17 | 0.4 | 0.38 | 2.05 | 1.90 | 0.0 | 62.02 | 14.20 | 6.42 | 88.4 | 19.45 | 24.2 |
| claude-3-sonnet | 17.43 | 3.26 | 1.7 | 0.44 | 2.05 | 1.89 | 0.0 | 64.31 | 50.78 | 6.64 | 89.1 | 30.72 | 23.9 |

Table 29: Code efficiency of widely-studied LLMs reported by EFFIBENCH (Hard).

| Model | max NET | NET | NET>5 | ET (s) | max NMU | NMU | NMU>5 | MU (Mb) | max NTMU | NTMU | NTMU>5 | TMU (Mb*s) | Pass@1 |
|---|---|---|---|---|---|---|---|---|---|---|---|---|---|
| **Open-source models** | | | | | | | | | | | | | |
| CodeLlama-7b-hf | | | | | | | | | | | | | |
| CodeLlama-13b-hf | | | | | | | | | | | | | |
| CodeLlama-34b-hf | 3.10 | 2.77 | 0.0 | 0.32 | 2.00 | 1.84 | 0.0 | 55.74 | 6.51 | 5.37 | 75.0 | 11.15 | 0.4 |
| CodeLlama-70b-hf | 3.26 | 3.08 | 0.0 | 0.32 | 2.03 | 1.97 | 0.0 | 49.01 | 6.90 | 6.37 | 100.0 | 10.47 | 0.7 |
| CodeLlama-7b-Instruct-hf | 17.26 | 6.60 | 25.0 | 1.19 | 3.59 | 2.40 | 0.0 | 57.44 | 56.61 | 18.86 | 100.0 | 71.07 | 0.4 |
| CodeLlama-13b-Instruct-hf | 3.11 | 2.83 | 0.0 | 0.31 | 2.04 | 1.92 | 0.0 | 50.02 | 6.46 | 5.68 | 71.4 | 10.31 | 0.7 |
| CodeLlama-34b-Instruct-hf | 4.16 | 2.87 | 0.0 | 0.33 | 2.56 | 1.96 | 0.0 | 53.45 | 10.32 | 5.83 | 75.0 | 11.44 | 1.2 |
| CodeLlama-70b-Instruct-hf | 3.10 | 2.92 | 0.0 | 0.32 | 2.03 | 1.94 | 0.0 | 51.16 | 6.43 | 5.93 | 90.0 | 11.15 | 1.0 |
| deepseek-coder-1.3b-instruct | 3.07 | 2.87 | 0.0 | 0.30 | 2.01 | 1.96 | 0.0 | 49.12 | 6.40 | 5.91 | 75.0 | 10.00 | 0.4 |
| deepseek-coder-6.7b-instruct | 3.34 | 2.93 | 0.0 | 0.31 | 2.02 | 1.96 | 0.0 | 49.01 | 7.02 | 5.99 | 90.0 | 10.16 | 1.0 |
| deepseek-coder-6.7b-base | 3.50 | 2.92 | 0.0 | 0.34 | 2.04 | 1.90 | 0.0 | 51.17 | 7.44 | 5.85 | 85.7 | 11.60 | 2.8 |
| deepseek-coder-33b-base | 3.60 | 2.94 | 0.0 | 0.32 | 2.02 | 1.93 | 0.0 | 49.52 | 7.64 | 5.93 | 90.7 | 10.54 | 4.3 |
| OpenCodeInterpreter-DS-1.3B | 3.93 | 3.34 | 0.0 | 0.34 | 2.00 | 1.99 | 0.0 | 48.37 | 8.44 | 7.05 | 100.0 | 11.14 | 0.3 |
| OpenCodeInterpreter-DS-6.7B | 3.20 | 2.96 | 0.0 | 0.33 | 2.02 | 1.98 | 0.0 | 48.71 | 6.81 | 6.02 | 100.0 | 10.86 | 1.4 |
| OpenCodeInterpreter-DS-33B | 26.06 | 3.64 | 2.6 | 0.44 | 2.43 | 1.92 | 0.0 | 51.74 | 66.25 | 7.72 | 86.8 | 16.81 | 3.8 |
| Phind-CodeLlama-34B-v1 | 3.35 | 3.00 | 0.0 | 0.32 | 2.02 | 1.97 | 0.0 | 48.88 | 6.95 | 6.04 | 92.9 | 10.40 | 1.4 |
| Phind-CodeLlama-34B-v2 | 4.12 | 3.04 | 0.0 | 0.35 | 2.02 | 1.93 | 0.0 | 50.91 | 9.52 | 6.17 | 91.3 | 12.02 | 2.3 |
| starcoder | 2.92 | 2.67 | 0.0 | 0.32 | 2.00 | 1.78 | 0.0 | 60.88 | 6.01 | 4.98 | 66.7 | 11.53 | 0.3 |
| starcoder2-3b | 3.07 | 3.07 | 0.0 | 0.31 | 1.99 | 1.99 | 0.0 | 48.37 | 6.26 | 6.26 | 100.0 | 10.23 | 0.1 |
| starcoder2-7b | 3.01 | 3.01 | 0.0 | 0.30 | 1.98 | 1.98 | 0.0 | 48.59 | 6.13 | 6.13 | 100.0 | 9.83 | 0.1 |
| starcoder2-15b | 3.20 | 3.20 | 0.0 | 0.31 | 2.01 | 2.01 | 0.0 | 47.95 | 6.59 | 6.59 | 100.0 | 10.07 | 0.1 |
| starcoderbase | 3.01 | 2.78 | 0.0 | 0.31 | 2.00 | 1.94 | 0.0 | 49.68 | 6.17 | 5.57 | 66.7 | 10.22 | 0.3 |
| WizardCoder-Python-13B-V1.0-GPTQ | 16.48 | 5.06 | 16.7 | 0.87 | 3.57 | 2.21 | 0.0 | 55.67 | 53.63 | 13.64 | 66.7 | 48.55 | 0.6 |
| WizardCoder-15B-V1.0 | 3.21 | 2.90 | 0.0 | 0.31 | 2.00 | 1.99 | 0.0 | 48.33 | 6.79 | 6.01 | 66.7 | 10.06 | 0.3 |
| XwinCoder-13B | 4.16 | 3.06 | 0.0 | 0.39 | 2.01 | 1.90 | 0.0 | 50.44 | 8.95 | 6.17 | 60.0 | 13.95 | 0.5 |
| XwinCoder-34B | 4.25 | 3.04 | 0.0 | 0.34 | 2.01 | 1.94 | 0.0 | 50.24 | 9.15 | 6.19 | 84.2 | 11.71 | 1.9 |
| Yi-34B-200K | 3.02 | 2.84 | 0.0 | 0.31 | 1.99 | 1.92 | 0.0 | 50.15 | 6.19 | 5.70 | 100.0 | 10.44 | 0.4 |
| Yi-34B-Chat | 2.98 | 2.93 | 0.0 | 0.31 | 2.01 | 1.93 | 0.0 | 50.04 | 6.36 | 6.01 | 100.0 | 10.40 | 0.2 |
| Yi-34B | 3.07 | 3.04 | 0.0 | 0.31 | 2.04 | 2.00 | 0.0 | 48.08 | 6.44 | 6.26 | 100.0 | 10.22 | 0.3 |
| Artigenz-Coder-DS-6.7B | 27.78 | 3.55 | 1.8 | 0.41 | 2.04 | 1.91 | 0.0 | 50.67 | 70.28 | 7.45 | 92.9 | 15.16 | 5.6 |
| CodeFuse-DeepSeek-33B | 4.24 | 3.12 | 0.0 | 0.35 | 2.03 | 1.94 | 0.0 | 48.71 | 9.10 | 6.36 | 88.6 | 11.74 | 3.5 |
| codegemma-7b | 3.43 | 2.99 | 0.0 | 0.33 | 2.02 | 1.91 | 0.0 | 51.34 | 7.25 | 6.04 | 93.3 | 11.06 | 1.5 |
| Magicoder-S-DS-6.7B | 4.16 | 3.03 | 0.0 | 0.34 | 2.61 | 1.94 | 0.0 | 49.86 | 10.77 | 6.19 | 92.5 | 11.28 | 5.3 |
| Mistral-7B-codealpaca-lora | 2.87 | 2.56 | 0.0 | 0.30 | 2.00 | 2.00 | 0.0 | 47.58 | 5.81 | 5.16 | 50.0 | 9.73 | 0.2 |
| octocoder | | | | | | | | | | | | | |
| **Closed-source models** | | | | | | | | | | | | | |
| gpt-3.5-turbo-0301 | 27.70 | 3.36 | 1.3 | 0.40 | 2.03 | 1.93 | 0.0 | 50.19 | 70.62 | 6.98 | 88.0 | 14.33 | 7.5 |
| gpt-3.5-turbo-0613 | 27.10 | 3.39 | 1.2 | 0.40 | 2.04 | 1.93 | 0.0 | 50.38 | 68.94 | 7.06 | 89.2 | 14.51 | 8.3 |
| gpt-3.5-turbo-1106 | 27.12 | 3.63 | 2.2 | 0.42 | 2.04 | 1.94 | 0.0 | 49.54 | 68.84 | 7.70 | 92.1 | 15.39 | 8.9 |
| gpt-4 | 4.53 | 3.06 | 0.0 | 0.34 | 2.04 | 1.92 | 0.0 | 50.40 | 10.16 | 6.18 | 92.3 | 11.64 | 9.1 |
| gpt-4-turbo-preview | 27.00 | 3.33 | 1.5 | 0.38 | 2.03 | 1.93 | 0.0 | 50.02 | 68.48 | 6.89 | 91.7 | 13.65 | 13.2 |
| claude-3-haiku | 28.75 | 3.61 | 1.4 | 0.42 | 2.02 | 1.92 | 0.0 | 50.34 | 72.87 | 7.57 | 91.4 | 15.20 | 7.0 |
| claude-3-sonnet | 3.75 | 3.17 | 0.0 | 0.35 | 2.03 | 1.93 | 0.0 | 50.35 | 8.20 | 6.45 | 90.2 | 11.71 | 6.1 |

Table 30: Overhead result of closed-source models efficiency testing time.

| model | time |
|---|---|
| gpt-3.5-turbo-0301 | 32s |
| gpt-3.5-turbo-0613 | 34s |
| gpt-3.5-turbo-1106 | 35s |
| gpt-4 | 37s |
| gpt-4-turbo-preview | 34s |
| claude-3-haiku | 17s |
| claude-3-sonnet | 24s |

theoretical complexity classes. This approach allows for a nuanced analysis of the models' output, enabling a comprehensive evaluation of their practical utility in software development scenarios.

## A.26 Discussion Automatically-generated Test Cases

As discussed in Section 3.3, EFFIBENCH generated test cases by first developing a test case generator for each coding problem, where we modify the test case generator to make sure the test cases generated by the generator are correct. Then, we use the test case generator to generate test cases for

Table 31: Evaluation results of test case accuracy for canonical solutions. For each test case generated by LLMs, we analyze whether the test case is accurate for the canonical solution. Then, we calculate the accuracy based on the total correct test cases/total generated test cases.

| Model | accuracy |
|---|---|
| GPT-3.5-turbo-0301 | 5.9 |
| GPT-3.5-turbo-0613 | 8.2 |
| GPT-4-turbo | 14.3 |
| GPT-4 | 13.7 |

each task. In this section, we discuss why do we not directly require LLM (e.g., GPT-3.5-turbo) to generate test cases for each task. Specifically, we provide the experiment results of four closed-source LLMs generated test cases' accuracy. The evaluation results are demonstrated in Table 31, where we can observe that the accuracy of the test cases generated by four LLMs is lower than 15%, which explains why do we not use LLM to generate test cases for each task, i.e., the accuracy of test cases are very low.

### A.27 Case illustration of test case generator

We provide a case example to illustrate that how does test case generator generate test cases for EFFIBENCH. Specifically, as shown in Figure 9, we can observe that the script is used to generate 100 tests for the function *lengthOfLongestSubstring*, where the test case generator randomly generates input and then feeds into the canonical solution. Then, the canonical solution returns the output for the given input.

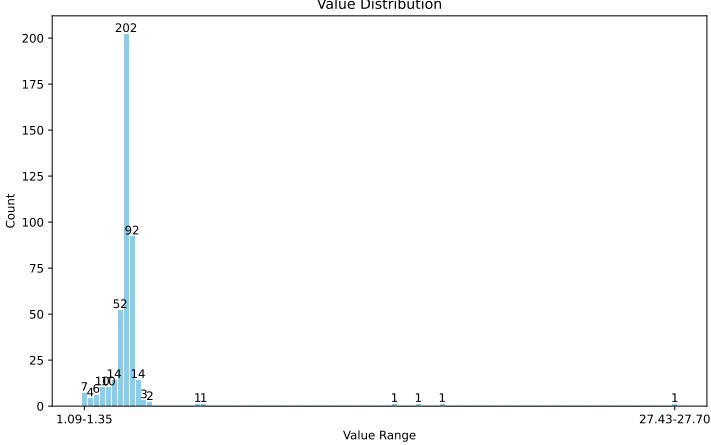

(a) Normalized execution time distribution.

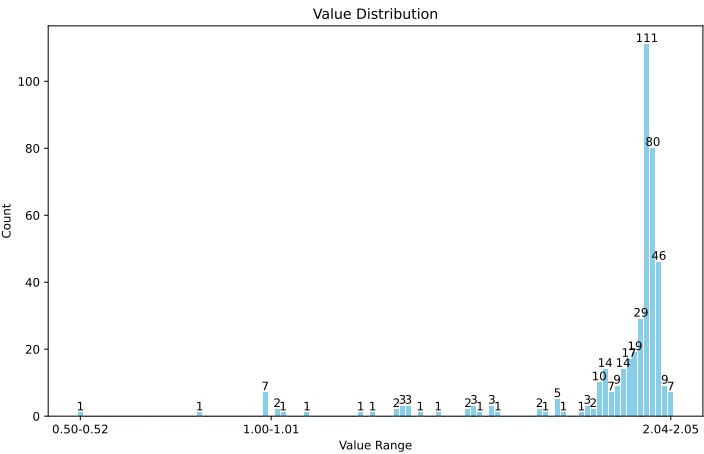

(b) Normalized maximum memory usage distribution.

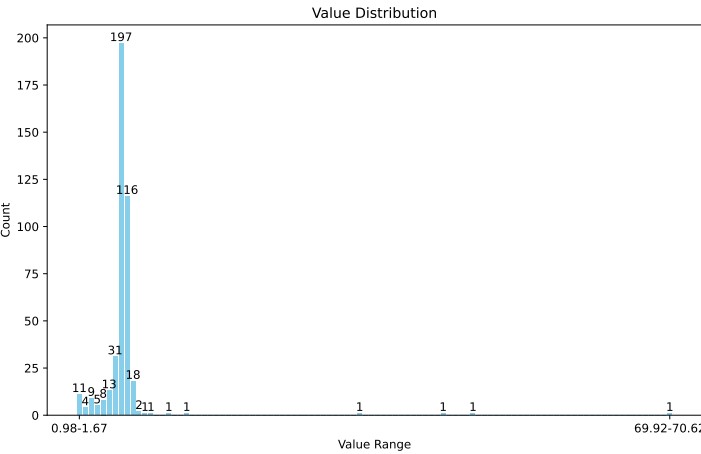

(c) Normalized memory usage distribution.

Figure 7: Various distributions of computational resources used by GPT-3.5 Turbo 0301 version. We divided the metric value range into ten columns based on the minimum and maximum values for each metric.

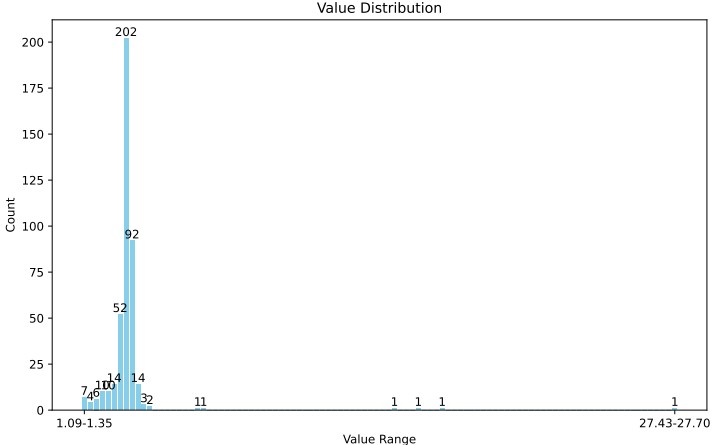

(a) Normalized execution time distribution.

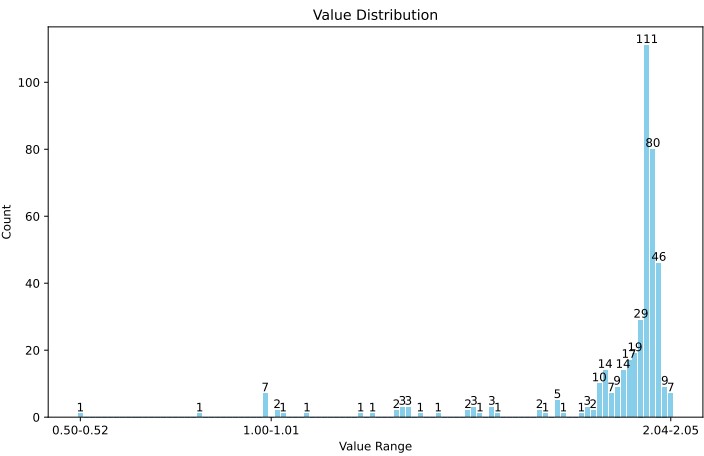

(b) Normalized maximum memory usage distribution.

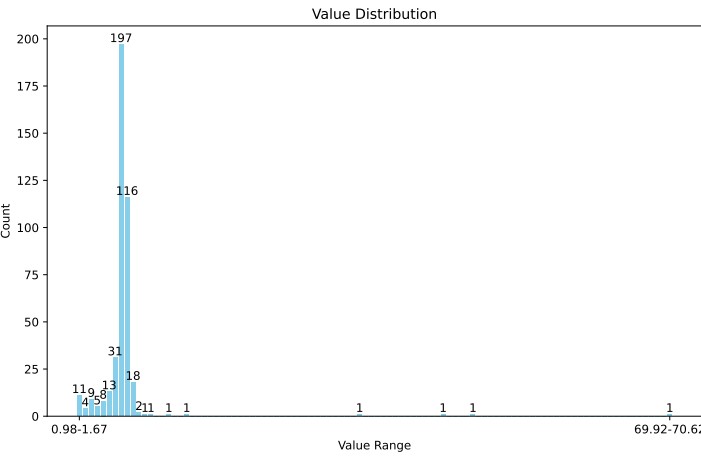

(c) Normalized memory usage distribution.

Figure 8: Various distributions of computational resources used by GPT-3.5 Turbo 0301 version. We divided the metric value range into ten columns based on the minimum and maximum values for each metric.

## Test Case Generation

```python
import random

class Solution:
    def lengthOfLongestSubstring(self, s: str) -> int:
        ss = set()
        i = ans = 0
        for j, c in enumerate(s):
            while c in ss:
                ss.remove(s[i])
                i += 1
            ss.add(c)
            ans = max(ans, j - i + 1)
        return ans

def generate_test_case():
    solution = Solution()

    # Generate a random string
    s = ''.join(random.choices('abcdefghijklmnopqrstuvwxyzABCDEFGHIJKLMNOPQRSTUVWXYZ0123456789',
    ↪ k=random.randint(0, 10)))

    # Calculate the expected result using the provided Solution class
    expected_result = solution.lengthOfLongestSubstring(s)

    return (s, ), expected_result

def test_generated_test_cases(num_tests):
    test_case_generator_results = []
    for i in range(num_tests):
        inputs, expected_result = generate_test_case()
        solution = Solution()
        assert solution.lengthOfLongestSubstring(*inputs) == expected_result

        test_case_generator_results.append(f"assert solution.lengthOfLongestSubstring({',
        ↪ '.join(map(repr, inputs))}) == {expected_result}")
    return test_case_generator_results

if __name__ == '__main__':
    num_tests = 100
    test_case_generator_results = test_generated_test_cases(num_tests)

    with open("./full_tmp/0.txt", "w") as f:
        f.write("\n".join(test_case_generator_results))
    print(len(test_case_generator_results))
```

Figure 9: A case illustration of the test case generation process for the LeetCode task. The test case generator (function generate_test_case) generate 100 tests for the solution.

