# OpenReview forum: "EffiBench: Benchmarking the Efficiency of Automatically Generated Code"
_NeurIPS.cc/2024/Datasets_and_Benchmarks_Track — NeurIPS 2024 Track Datasets and Benchmarks Poster_

### Official Review · Reviewer_CNU4 · 2024-07-16
**Important contribution to measuring generated code efficiency in an exec aware manner**

**Rating:** 7
**Confidence:** 4
**Correctness:** Yes
**Clarity:** Yes

**Review:**

Overall, this dataset is a valuable contribution as most evals for code gen models so far focus on exec-based correctness. The authors uncover that many SOTA models might generate correct but inefficient code a lot of the time. Additionally, they openly release their dataset. Overall, I think this paper is a clear accept.

**Strengths:**

1. The authors release a code efficiency profiling dataset which is novel.
2. The authors conduct a thorough analysis of the efficiency profile of model-generated code.

**Additional Feedback:**

N/A

**Documentation:**

Yes

**Ethics:**

Yes

**Limitations:**

Yes

**Opportunities For Improvement:**

1. The problem distribution for the code is somewhat limited by being sourced only from a few types of leetcode questions
2. The authors might consider using something like the Gem5 simulator (https://www.gem5.org/) for more accurate profiling of runtimes as there can be large variations in measured runtimes from run to run even on the same system.
3. The authors might consider releasing a containerized env for easily repeatable runs.

**Relation To Prior Work:**

Yes

**Summary And Contributions:**

The authors collect leetcode problems that test proficiency in using specific methods and synthetically generate test-cases for them. The resultant dataset can be used for exec based eval but also for measuring efficiency using the measured runtimes against canonical solution (most starred discussion solution on leetcode).

---

> ### Author Rebuttal · Authors · 2024-08-14
>
> Dear Reviewer CNU4,
>
> Thanks for your appreciation for EffiBench and believe that it is an important contribution to measuring generated code efficiency. For the suggestions and questions,  we provided detailed clarifications and additional experiments to address your concerns below. Since EffiBench is currently in the borderline scenario, if you feel that all of your concerns have been adequately addressed, we would deeply appreciate it if you could consider increasing your overall assessment of EffiBench.
>
> **The problem distribution for the code is somewhat limited by being sourced only from a few types of leetcode questions**
>
> Great points. First, we use and focus on LeetCode questions is due to we need efficiency-critical tasks, where in these tasks different solutions may have a different overhead. In addition, we also enhance our benchmark by providing an easy-to-use framework that can also apply datasets such as HumanEval and MBPP to measure the efficiency of LLM-generated code, which is also demonstrated in our GitHub repo.
>
>
>
> **The authors might consider using something like the Gem5 simulator (https://www.gem5.org/) for more accurate profiling of runtimes as there can be large variations in measured runtimes from run to run even on the same system.**
>
> Thanks for your suggestion. We provide the efficiency results of six different LLMs with 20 times execution below with the PIE-provided Gem5 simulator [1]. The results format is **mean (std)**. We can observe that although the values are a little different from the evaluation results in our machine setup. However, we can also observe that StarCoder2-15B also achieves SOTA efficiency for the generated code in open-source models. For example, the NTMU of StarCoder2-15B is 1.18x compared with canonical solutions, while other LLMs such as deepseek-coder-1.3b-instruct will require 5.49x.
>
> Next, we can also observe that the std of 20 times execution is very small. For example, The ET std of StarCode2-15B is 0.02, while the ET mean is 0.49 (s). We believe that the variation does not impact the overall efficiency measurement.
>
>
> | Steps                        | ET (s)      | NET         | MU (Mb)      | NMU         | TMU (Mb*s)   | NTMU        |
> |------------------------------|-------------|-------------|--------------|-------------|--------------|-------------|
> | OpenCodeInterpreter-1.3B     | 0.80 (0.03) | 5.74 (0.03) | 62.60 (0.02) | 1.61 (0.00) | 45.18 (0.23) | 5.63 (0.03) |
> | deepseek-coder-1.3b-instruct | 0.63 (0.02) | 4.57 (0.02) | 59.32 (0.02) | 1.64 (0.00) | 27.57 (0.14) | 5.49 (0.03) |
> | CodeLlama-7b-Instruct-hf     | 3.98 (0.04) | 23.93 (0.04)| 103.78 (0.35)| 2.19 (0.01) | 370.74 (1.09)| 28.06 (0.08)|
> | starcoder2-15b               | 0.49 (0.02) | 1.88 (0.02) | 132.27 (0.04)| 1.17 (0.00) | 53.54 (0.92) | 1.18 (0.02) |
> | gpt-3.5-turbo-0301           | 0.68 (0.03) | 4.83 (0.03) | 102.39 (0.00)| 2.82 (0.00) | 302.04 (0.17)| 39.98 (0.02)|
> | claude-3-sonnet              | 0.74 (0.03) | 5.20 (0.03) | 64.02 (0.01) | 1.76 (0.00) | 56.85 (0.24) | 7.29 (0.03) |
>
>
> *Evaluation results of different LLMs in PIE provided Gem5 simulator.*
>
> [1] Shypula, Alexander, et al. "Learning performance-improving code edits." arXiv preprint arXiv:2302.07867 (2023).
>
>
> **The authors might consider releasing a containerized environment for easily repeatable runs.**
>
> Thanks for your suggestion. We have provided an environment to make sure researchers can conduct experiments in the same environment in our HuggingFace Leaderboard, researchers can submit their completed code and then the efficiency results would be returned. Next, we also pushed a docker into hdong1999/effibench. Researchers now can run *docker pull hdong1999/effibench* to achieve the same environments as our experiments.

---

> > ### Author Response · Authors · 2024-08-26
> > **Kindly Reminder**
> >
> > Dear Reviewer CNU4,
> >
> > Thank you for your comment. To address your concerns about `Opportunities For Improvement`, we have provided detailed clarification about EffiBench limitations, and we also provided the new experiments and also uploaded our execution environments in docker.
> >
> > We hope that these updates adequately address your concerns. In light of these changes, we kindly request that you reconsider your overall assessment.

---

> ### Author Response · Authors · 2024-09-01
> **Thank you so much for your positive review!**
>
> Dear Reviewer CNU4,
>
> Thank you so much for your positive review!
>
> We sincerely appreciate your thorough review and valuable suggestions. We will carefully incorporate your comments into the final paper.
>
> As the Author-Review discussion period is nearing its end, we hope that our explanation adequately addresses your concerns. Thank you very much for your time and consideration!
>
> Best regards,
>
> The Authors of EffiBench

---

### Official Review · Reviewer_ufVm · 2024-07-22
**Benchmarking the Efficiency of Automatically Generated Code**

**Rating:** 6
**Confidence:** 4
**Correctness:** Yes
**Clarity:** Yes

**Review:**

+ The paper demonstrates its quality by providing several results of baselines.
  - The paper conducts various experiments with an extensive set of LLMs to justify their claims.

+ The dataset can be useful in the field of automatic code generation by focusing on code efficiency.

+ The paper lacks clarity due to its insufficient explanation of experimental results.
  - The analysis in Section 5.1 merely rephrases the results from Table 3. Therefore, an in-depth explanation of the numerical analysis is lacking. Instead of simply examining the time and space efficiency of code generated by LLMs, it would be beneficial to supplement the analysis based on the fundamental differences between the models. Having more parameters does not necessarily lead to better performance, as seen in the case of CodeLlama. This aspect should be further explained.
  - There is no analysis to determine whether the test cases generated by the generator are truly reliable. There should be a way of verifying whether inputs and outputs are valid since even the solution codes of the problems may work improperly outside the scope of the problem. Alongside this problem, the authors should consider the coverage of the test cases.
  - There is also no analysis to determine whether the considered metric is reliable, especially in measuring memory usage. Are there any previous works addressing code efficiency that use the same metrics as yours?

+ There is no sophisticated consideration of code efficiency.
  - Despite claiming to evaluate code efficiency, the benchmark construction process does not differ from other benchmarks.

**Strengths:**

The paper evaluates 42 LLMs (both open and closed source) and provides rich results of baselines. These evaluations provide the performance of the baselines on the proposed benchmark through the various metrics.

**Additional Feedback:**

1. There are no algorithm tags or difficulty levels provided. It would be beneficial to add this information to the benchmark.
2. Overall, the text size in the tables is small and not very readable.
3. It would improve readability to add arrows (e.g., \downarrow) to the metrics in Tables 3, 4, and 5 to indicate that smaller scores are better.
4. It could be beneficial to describe the process used to create the test case generator. For example, disclosing the prompts that are used would be a good approach.

**Documentation:**

Yes

**Limitations:**

Yes

**Opportunities For Improvement:**

The paper focuses on numerical comparison, but it lacks in-depth analysis.
1. In particular, there is a limited discussion regarding the reasons behind the varying performance of different models. A more thorough analysis is needed to understand the performance differences that arise due to the specific characteristics and architectures of the models.
2. The metrics in the paper only use the overall execution time and the memory usage compared to canonical solutions. The metrics should consider the time and space complexity since the research problem originated from the efficiency of codes. For example, there can be variations in execution time depending on the input length. An analysis of the relationship between input length and execution time would be beneficial.

**Relation To Prior Work:**

Yes

**Summary And Contributions:**

The paper introduces "EFFIBENCH," a benchmark specifically designed to evaluate the efficiency of automatically generated codes. Recent studies showed that LLM-generated codes can be less efficient than standard solutions in terms of execution time and memory usage. To bridge this gap, the authors present EFFIBENCH as a benchmark focused on evaluating the efficiency of code generation.
 EFFIBENCH is constructed as follows:
 1. Collect all 2,605 problems tagged with “LeetCode” on the HuggingFace, resulting in the selection of 1,146 typical algorithm problems.
 2. Collect the top-starred solutions for each problem from the LeetCode Discussion Forum for canonical solution construction.
3. Leverage LLM for generating test cases of code problems.

---

> ### Author Rebuttal · Authors · 2024-08-14
>
> Dear Reviewer ufVm,
>
> Thank you for your valuable comments on EffiBench. We appreciate your suggestions and have provided detailed clarifications and additional experiments to address your concerns below. Since EffiBench is currently in the borderline scenario, if you feel that all of your concerns have been adequately addressed, we would deeply appreciate it if you could consider increasing your overall assessment of EffiBench.
>
> **The analysis in Section 5.1 merely rephrases the results from Table 3. Therefore, an in-depth explanation of the numerical analysis is lacking. Instead of simply examining the time and space efficiency of code generated by LLMs, it would be beneficial to supplement the analysis based on the fundamental differences between the models. Having more parameters does not necessarily lead to better performance, as seen in the case of CodeLlama. This aspect should be further explained.**
>
>
> Thanks for your suggestion. To examine the correlation between model parameters and efficiency performance, we provide two figures. The first figure, which includes [all open-source models](https://github.com/huangd1999/EffiBench/blob/main/relationship_codellama_same_tasks.pdf), illustrates the efficiency values presented in Table 3 of our paper. From this figure, we can observe that there is no strong correlation between model parameters and efficiency metrics. The second figure focuses on the CodeLlama-Instruct family of models, detailing their parameters and performance on three tasks successfully addressed by the CodeLlama-Instruct family. This figure can be found in the [CodeLlama Figures](https://github.com/huangd1999/EffiBench/blob/main/relationship_codellama_same_tasks.pdf). Here too, we observe a lack of correlation.
>
> Next, in our final version, we will expand this analysis to discuss potential reasons for the performance differences between models, such as variations in model architectures, training datasets, and sizes. We will note non-intuitive findings like CodeLlama's performance not scaling with increased parameters. While fully explaining these differences is challenging without knowing all the model details, we will aim to provide well-reasoned hypotheses and insights to the extent possible.
>
>
>
> **There is no analysis to determine whether the test cases generated by the generator are truly reliable. There should be a way of verifying whether inputs and outputs are valid since even the solution codes of the problems may work improperly outside the scope of the problem. Alongside this problem, the authors should consider the coverage of the test cases.**
>
>
> Following the instruction of Evalpuls [1], we have checked the test case effectiveness by analyzing the code line coverage of test cases for canonical solutions with the Coverage.py package [https://coverage.readthedocs.io/en/7.4.4/]. Our evaluation results illustrate that the code line coverage of each canonical solution obtains 99.24% and 99.70% for public test cases and private test cases. In addition, when we utilize 1000 tests in our constructed benchmark, the code line coverage further increases to 99.95%, which provides strong evidence for the effectiveness of our tests. The test case generator of EffiBench can further increase the code line coverage. We will add the coverage analysis in our final version. By the way, the long-string random inputs are more suitable for worst-case code efficiency analysis, e.g., revealing the worst-case fast-sorting functions' time complexity.
>
>
>
>
> **There is also no analysis to determine whether the considered metric is reliable, especially in measuring memory usage. Are there any previous works addressing code efficiency that use the same metrics as yours?**
>
> Traditional works [1-6] have the same pipeline to measure the efficiency of the programs. For example, to measure the max memory usage, existing works also utilize a memory profiler to profile the memory usage of the program.
>
> [1] https://www.pluralsight.com/resources/blog/guides/profiling-memory-usage-in-python
>
> [2] Leelaprute, Pattara, et al. "Does coding in pythonic zen peak performance? preliminary experiments of nine pythonic idioms at scale." Proceedings of the 30th IEEE/ACM International Conference on Program Comprehension. 2022.
>
> [3] Doglio, Fernando. Mastering Python High Performance. Packt Publishing Ltd, 2015.
>
> [4] Gorelick, Micha, and Ian Ozsvald. High Performance Python: Practical Performant Programming for Humans. O'Reilly Media, 2020.
>
> [5] Lanaro, Gabriele. Python high performance. Packt Publishing Ltd, 2017.
>
> [6] Lanaro, Gabriele, Quan Nguyen, and Sakis Kasampalis. Advanced Python Programming: Build high performance, concurrent, and multi-threaded apps with Python using proven design patterns. Packt Publishing Ltd, 2019.
>
> **There is no sophisticated consideration of code efficiency. Despite claiming to evaluate code efficiency, the benchmark construction process does not differ from other benchmarks.**
>
> We clarify that the tasks collected by EffiBench are mainly efficiency-critical, which is different from existing datasets such as HumanEval and MBPP, and EffiBench also constructs a test case generator to generate massive test cases (default 100 and also provide 1000 tests in our experiments),  which also different with existing datasets such as APPS. Second, the test case generator constructed by EffiBench is also a contribution compared with existing works that directly constructed a few test cases (e.g., lower than 10 for MBPP and HumanEval), our test case generator can generate massive test cases, e.g., 100 to 1000, for efficiency experiments, where the efficiency measurements need massive test cases to evaluate the efficiency of code. Second, another contribution of EffiBench is our efficiency testing framework, which allowed existing researchers to conduct efficiency experiments on simple datasets such as HumanEval.
>
>
> **Continue in the next thread**

---

> > ### Author Rebuttal · Authors · 2024-08-14
> >
> > **The metrics in the paper only use the overall execution time and the memory usage compared to canonical solutions. The metrics should consider the time and space complexity since the research problem originated from the efficiency of codes. For example, there can be variations in execution time depending on the input length. An analysis of the relationship between input length and execution time would be beneficial.**
> >
> > Thanks for your suggestion. **We provide the detailed reasons for EffiBench does not consider Time/Space complexity in EffiBench Appendix A.16.** We agree that time and space complexity also can be used to measure the efficiency of the codes. However, for two solutions even have the time complexity (e.g., O(n)), the actual calculation times may be T (2n) and T(n), and then the NET of two solutions would be 2 (theoretically), and 1 for the canonical solution if the canonical solution is same as the second solutions generated by LLMs.
> >
> > In addition, as discussed above, we theoretically treat each operation of the two solutions as identical. However, in real-world applications, different operations may have varying execution times and memory usage. For example, some operations might be more computationally expensive or require more memory than others. These differences in execution time and memory usage can further lead to inaccurate conclusions about the overall efficiency of the solutions if we utilize time/space complexity for efficiency measurement.
> >
> > In addition, time/space complexity requires developers to check manually, which then poses two potential challenges. First, manual analysis and labeling are subjective and prone to human error. Different developers may have varying interpretations of the code's complexity, leading to inconsistent or inaccurate assessments. Second, manual analysis is difficult to apply at the dataset level for efficiency measurements. For example, EffiBench contains 1,000 tasks, and manually analyzing the time/space complexity of each task would be an extremely time-consuming and labor-intensive process. It would require significant effort from developers to carefully examine each task, understand its algorithmic structure, and determine the corresponding time and space complexities. This manual effort would be impractical and would hinder the scalability of efficiency evaluation across large datasets. Therefore, while time/space complexity can provide insights into algorithmic efficiency, it may not be the most practical or reliable approach for assessing efficiency in large-scale benchmarks like EffiBench.
> >
> >
> > Finally, for the relationship between input length and execution time, we provide a case study in a task (problem idx 2206) of the evaluation results of the CodeLlama-ins family for the distribution of input length and overhead results in [Link](https://github.com/huangd1999/EffiBench/blob/main/model_metrics_comparison.pdf). We can observe that the execution time (ET) and total memory usage (TMU), do not have a clear correlation between these two metrics and input length. However, we can observe that for the peak memory usage, the MU would be increased once the input length increases. The key reason is that the solution generated by four LLMs (CodeLlama-Ins-7B, 13B, 34B, and 70B in [Link](https://github.com/huangd1999/EffiBench/tree/main/codellama-family-solutions)) tends to allocate memory in a manner that scales with the size of the input. Specifically, as the input length increases, the algorithms may need to store more intermediate data structures, buffers, or auxiliary variables, leading to higher peak memory usage (MU).
> >
> >
> > **There are no algorithm tags or difficulty levels provided. It would be beneficial to add this information to the benchmark.**
> >
> > Thanks for your suggestion. We have uploaded it in our GitHub source code data/dataset_with_difficulty_and_algorithm.json. We will also add some discussion for these two attributes in our final version.
> >
> >
> > **Overall, the text size in the tables is small and not very readable.**
> >
> > Thanks for your suggestion. We will revise our table in our final version to make it easy to read.
> >
> >
> > **It would improve readability to add arrows (e.g., \downarrow) to the metrics in Tables 3, 4, and 5 to indicate that smaller scores are better.**
> >
> > Thanks for your suggestion. We will add it in our final version.
> >
> >
> > **It could be beneficial to describe the process used to create the test case generator. For example, disclosing the prompts that are used would be a good approach.**
> >
> > Thanks for your suggestion. We have uploaded the test case generator file and prompts in our GitHub source code (in src/gpt_generate_test_case_generator.py).

---

> > > ### Comment · Reviewer_ufVm · 2024-08-22
> > >
> > > Thank you for addressing my concerns with your experimental analysis and explanations. Your response has alleviated some of my worries, and as a result, I have decided to slightly increase my overall evaluation. However, there are still some shortcomings that need to be addressed. The contribution of the test case generator seems to primarily rely on ChatGPT, which, in my opinion, is insufficient to make a substantial impact in the field. While the large quantity of test cases is nice, their quality is not well-justified. Although the authors have demonstrated the coverage of test cases using a Python package, the fact that the generator predominantly relies on ChatGPT may limit its value as a significant advancement in the field. While ChatGPT can generate a large number of test cases, its utility as the sole engine behind the generator raises concerns. It would be better if the authors could provide an analysis that substantiates the reliability and robustness of the test cases.

---

> > > > ### Author Response · Authors · 2024-08-23
> > > >
> > > > Dear Reviewer vfVm,
> > > >
> > > > Thank you for your appreciation of our rebuttal and for increasing the overall score. We would like to address the concern regarding the test cases with the following points.
> > > >
> > > >
> > > > **The contribution of the test case generator seems to primarily rely on ChatGPT, which, in my opinion, is insufficient to make a substantial impact in the field.**
> > > >
> > > > Response:
> > > >
> > > > Thank you for your feedback. We would like to clarify that once we generated a test case generator for each canonical solution, we also manually verified and rectified all test case generators to make sure that they could generate valid test cases (see more explanation in our answer to your second point).
> > > >
> > > > Our primary and more impactful contribution lies in our comprehensive efficiency benchmarking for existing LLMs in code generation tasks, and the large-scale empirical study encompassing 42 LLMs (i.e., 35 open-source and 7 closed-source models). This extensive analysis offers valuable insights and benchmarks that significantly advance the understanding and evaluation of LLM efficiency in the field.
> > > >
> > > >
> > > > **While the large quantity of test cases is nice, their quality could be better justified. Although the authors have demonstrated the coverage of test cases using a Python package, the fact that the generator predominantly relies on ChatGPT may limit its value as a significant advancement in the field. While ChatGPT can generate a large number of test cases, its utility as the sole engine behind the generator raises concerns. It would be better if the authors could provide an analysis that substantiates the reliability and robustness of the test cases.**
> > > >
> > > > Response:
> > > >
> > > > Thank you for acknowledging the extensive set of test cases provided by EffiBench. We would like to clarify a few points to address your concerns:
> > > > - **Test Generation Process:** We did not directly use the tests generated by ChatGPT. Instead, we used an LLM to create a test case generator. Each generator was manually verified and corrected to ensure it could produce valid test cases. This involved checking that the input parameter ranges aligned with the task description and running the generator to ensure the canonical solution could process the generated inputs. If the canonical solution encountered errors or invalid inputs, we revised the test case generator accordingly. In cases where invalid inputs were generated, we skipped those and continued with valid ones.
> > > > - **Input-Output Validation:** After confirming the correctness of all test case generators, we used them to generate 100 or 1000 test inputs for our experiments. These inputs were processed by the canonical solution, which provided the corresponding outputs. We then used these input-output pairs to construct assertion oracles for our experiments. Because the outputs were generated by the canonical solution, we can ensure that the assertions are accurate, establishing a reliable ground truth for evaluating the correctness of LLM-generated code.
> > > > - **Coverage and diversity of generated tests:** To maximize code coverage, we instructed the test case generator to incorporate random and diverse test cases. This approach allowed the generator to achieve 99.70% and 99.95% code line coverage with 100 and 1000 tests, respectively, ensuring that most edge cases were addressed.
> > > > - **Robustness of generated tests:** We provide two experiments to demonstrate the robustness of generated tests. The first is flaky results for five/ten times execution in the canonical solution, where as shown in Table 1, we can observe that the flaky test results are zero for both five and ten executions. Next, we also report the pass@1 of five-time executions for the GPT-3.5-turbo generated code, where we can also observe that the std is zero for five-time executions.
> > > > - **Purpose of Test Generation:** Our primary goal was to generate a large and diverse set of test cases to reveal efficiency differences between different programs during execution. While producing highly reliable tests was not our main focus, the rigorous process we followed ensured the tests were indeed reliable.
> > > >
> > > > We believe these points demonstrate that our generated tests are robust enough to support our key contributions: providing a code efficiency benchmark and conducting an extensive empirical study on the efficiency of LLM-generated code.
> > > >
> > > >
> > > > *Table 1. Flaky result for different execution times.*
> > > >
> > > > |Model|5|10|
> > > > |-----|-----|----|
> > > > |EffiBench|0|0|
> > > >
> > > > Finally, to further discuss the robustness of the test cases for LLM-generated code, we provide the pass@1 metrics for GPT-3.5-turbo-0301. The results show consistent pass@1 metrics across five executions, with a standard deviation of 0, confirming the robustness of the test cases.
> > > >
> > > >
> > > > *Table 2. The evaluation result of GPT-3.5-turbo-0301 (pass@1) for five times execution.*
> > > >
> > > > | number of tests | pass@1 |
> > > > |-----|--------|
> > > > | 0| 42.3|
> > > > | 1| 42.3|
> > > > | 2| 42.3|
> > > > | 3| 42.3|
> > > > | 4| 42.3|
> > > > | **Mean**  | 42.3|
> > > > | **Std**| 0|

---

> > > > ### Author Response · Authors · 2024-08-26
> > > > **Kindly Reminder**
> > > >
> > > > Dear Reviewer ufVm,
> > > >
> > > > Thank you for taking the time to review our work and provide valuable feedback. We have carefully considered your concerns regarding the reliability and robustness of the test cases.
> > > >
> > > > In response to your comments, we have conducted additional experiments and provided detailed clarifications to address these issues. We have included new results that demonstrate the improved reliability and robustness of our test cases.
> > > >
> > > > We hope that these updates adequately address your concerns and provide a clearer picture of the strengths of our work. In light of these changes, we kindly request that you reconsider your overall assessment.
> > > >
> > > > We would greatly appreciate a brief response acknowledging that you have received our message and had a chance to review the updated materials.
> > > >
> > > > Thank you once again for your time and consideration.

---

> > > > > ### Comment · Reviewer_ufVm · 2024-08-28
> > > > >
> > > > > The authors have sufficiently addressed the concerns raised, and as a result, most of my concerns are now resolved well. Accordingly, I have adjusted my score once more. Thank you for the careful response.

---

### Official Review · Reviewer_s8Hy · 2024-07-29
**The contributions of this paper are substantial. Further enhancement could be achieved by exploring the interrelationships among multiple metrics in conjunction with real-world scenarios.**

**Rating:** 7
**Confidence:** 3
**Correctness:** Yes.
**Clarity:** Yes.

**Review:**

- This paper is well-written and easy to follow.
- This paper provides detailed data collection process and data statistics.
- This paper release an efficiency testing framework, which enables evaluating the efficiency across various code generation benchmarks.
- It will be better to discuss the contribution of different metrics to efficiency.

**Strengths:**

- It is meaningful to further consider the execution efficiency of the generated code, which can promote the next development of the code generation model.
- The paper is well-written and easy to follow.
- The discoveries presented in this paper hold significant implications for the future trajectory of research in code generation models.

**Additional Feedback:**

N/A

**Documentation:**

Yes

**Limitations:**

The author has discussed the limitation in the Appendix.

**Opportunities For Improvement:**

- Efficiency-critical Problems are collected from LeetCode. These problems may have appeared in the training set of the code generation model evaluated in this paper. However the paper does not explain or discuss the data leakage problem.
- Experimental findings reveal that while certain models produce highly accurate code, their efficiency remains suboptimal. In practical scenarios, the trade-off between correctness and efficiency becomes pivotal. Further exploration of this topic could provide valuable insights for practitioners.
- This paper introduces several metrics designed to assess the efficiency of generated code. However, in algorithmic problems, trade-offs between time complexity and space complexity are inherent. Therefore, a thorough analysis of their interplay with efficiency and their relative importance is of considerable value.

**Relation To Prior Work:**

Yes.

**Summary And Contributions:**

The authors propose the first benchmark specifically designed to assess the efficiency of code generated by LLMs. In addition, they conduct an extensive evaluation of 42 LLMs on the benchmark.

---

> ### Author Rebuttal · Authors · 2024-08-14
>
> Thanks for your appreciation of EffiBench and believe that the contribution of EffiBench is substantial. In our rebuttal, we hereby address your concerns below:
>
> **Opportunities For Improvement 1**
>
>
> Thanks for your reminder about whether the problems are already in the LLM training data.
>
> Firstly, it's important to note that the primary objective of EffiBench is not to evaluate the correctness of the generated code but rather to benchmark the efficiency of LLM-generated code on the provided tasks. This focus on efficiency means that even if the generated code resembles the original code, it does not impact the core aim of EffiBench.
>
> Next, to address the reviewer’s concern, we follow the instructions of SWT-Bench [1] to analyze whether the tasks are already in the LLM training dataset. Specifically, we conduct experiments for seven different LLMs and analyze the pass@1 before the knowledge cutoff and after the knowledge cutoff to demonstrate that the leetcode source code may not exist in the LLM training dataset as the researchers already filter these data from the training dataset. As shown in the below. We can observe that the pass@1 of LLMs for the before KC and after KC have similar pass@1 (the difference is lower than 0.3%). We will add these analyses in our final version.
>
> Furthermore, as shown in the EffiBench source code, EffiBench also provides an easy-to-use framework for other datasets to measure the efficiency of LLM-generated code. If in the future EffiBench’s tasks are leaked in the LLMs, researchers can directly use the EffiBench framework to measure LLM’s efficiency in other datasets that are not leaked to LLMs.
>
> *We analyze LLMs pass@1 for the tasks before the knowledge cutoff (Before KC) and after the knowledge cutoff (After KC).*
>
> |Model|Before KC|After KC|
> |-----|-----|----|
> |deepseek-coder-1.3b-instruct|4.80|4.51|
> |CodeLlama-7b-Instruct-hf|7.00|6.82|
> |CodeLlama-70b-Instruct-hf|7.80|7.74|
> |starcoder2-3b|1.70|1.56|
> |gpt-3.5-turbo-0301|44.00|43.77|
> |gpt-4|52.50|52.22|
> |claude-3-sonnet|45.50|45.35|
>
> [1] Mündler, Niels, et al. "Code Agents are State of the Art Software Testers." arXiv preprint arXiv:2406.12952 (2024).
>
> **Opportunities For Improvement 2**
>
>
> Thank you for this suggestion. We agree that the trade-off between code correctness and efficiency in practical scenarios is an important consideration that warrants further exploration.
>
> To address your concern, we provide evaluation results to demonstrate the correlations/trade-offs between correctness and efficiency in [Link](https://github.com/huangd1999/EffiBench/blob/main/correaltion_pass@1_efficiency.pdf), where the x-axis represents the pass@1 of LLM-generated code, while the y-axis represents the efficiency of LLM-generated code for the efficiency metrics. We can observe that for most of the metrics, the overhead increases when LLMs with higher pass@1 are used for most of the efficiency metrics. The key reason may be that LLMs with higher pass@1 may address some tasks with higher overhead compared to other tasks, which then increases average efficiency compared to only addressing easy tasks that would also be addressed by LLMs with lower pass@1.
>
> To further discuss the correlation between pass@1 and the efficiency of LLM-generated code, we use three tasks addressed by the CodeLlama family [See Link](https://github.com/huangd1999/EffiBench/blob/main/correaltion_pass@1_efficiency_codellama.pdf) and 210 tasks for closed-source LLMs [See Link](https://github.com/huangd1999/EffiBench/blob/main/correaltion_pass@1_efficiency_closed_llms.pdf) and then analyze the correlation between efficiency and pass@1 of the CodeLlama family and closed-source LLMs, where the CodeLlama family and closed-source LLMs are evaluated individually as no task is addressed by all of these models. We can observe that in most of the metrics, LLMs with lower pass@1 have lower overhead, i.e., in most of the metrics, efficiency and correctness have a positive correlation.
>
> In future work, we plan to dig deeper into analyzing and characterizing these trade-offs across different types of models and problems. This will provide additional useful insights for LLM training researchers needing to balance these factors.
>
>
> **Opportunities For Improvement 3**
>
>
> Thank you for your suggestion. We have provided the correlation between ET and MU for seven LLMs at both the task and model levels. The task-level correlation, presented in [Link](https://github.com/huangd1999/EffiBench/blob/main/rebuttal/correlation_metrics_task.pdf) analyzes the relationship between ET and MU for different LLM-generated codes in four specific tasks addressed by seven closed-source LLMs. We can observe that it does not have a strong trade-off between metrics in EffiBench. The key reason may be that the LLM-generated code is less efficient, which means that the code may both less efficient in ET and MU.
>
>
> Furthermore, the model-level correlation, shown in[Link](https://github.com/huangd1999/EffiBench/blob/main/rebuttal/correlation_metrics_model.pdf), is conducted by analyzing the ET and MU of all code generated by each model. We can also observe that different from the trade-off insights, the trade-off between ET and MU still does not exist. We believe that the key reason is the same as task-level correlation.
>
>
> Finally, we appreciate your suggestions and have provided detailed clarifications and additional experiments to address your concerns below. Since EffiBench is currently in the borderline scenario, if you feel that all of your concerns have been adequately addressed, we would deeply appreciate it if you could consider increasing your overall assessment of EffiBench.

---

> > ### Comment · Reviewer_s8Hy · 2024-08-18
> >
> > The author has well addressed most of my concerns.  I have increased my score.

---

> > > ### Author Response · Authors · 2024-08-18
> > >
> > > Dear Reviewer s8Hy:
> > >
> > > Thank you for acknowledging that we have addressed your concerns and increasing your score from 6 to 7.

---

### Decision · Program_Chairs · 2024-09-26

**Decision:**

Accept (Poster)

**Comment:**

This paper has undergone a good rebuttal discussion and eventually received overall positive review scores: 7, 6, 7. Reviewers overall recognized the merit of this work: a well-motivated benchmark, clear writing, and rich results of baselines. And they also raised some concerns: potential data leakage and limited data distribution using LeetCode problems, trade-off between correctness and efficiency, unclear experimental result explanation, more thorough analysis and discussion, etc. The rebuttal was persuasive and addressed most concerns. The final ratings unanimously recommend acceptance. The AC checked the paper, rebuttal, and review comments, and recommends accepting the paper.